# A statistical fracture model for Antarctic ice shelves and glaciers

Veronika Emetc[1], Paul Tregoning[1], Mathieu Morlighem[2], Chris Borstad[3], and Malcolm Sambridge[1]

[1]Research School of Earth Science, Australian National University, Canberra, Australia
[2]Department of Earth System Science, University of California, Irvine, USA
[3]Department of Arctic Geophysics, The University Centre in Svalbard

*Correspondence to:* Veronika Emetc (Veronika.Emetc@anu.edu.au)

**Abstract.** Antarctica and Greenland hold enough ice to raise sea level by more than 65 m if both ice sheets were to melt completely. Predicting future ice sheet mass balance depends on our ability to model these ice sheets, which is limited by our current understanding of several key physical processes, such as iceberg calving. Large scale ice flow models either ignore this process or represent it crudely. To model fractured zones, an important component of many calving models, Continuum Damage Mechanics as well as Linear Fracture Mechanics are commonly used. However, these methods have a large number of uncertainties when applied across the entire Antarctic continent because the models were typically tuned to match processes seen on particular ice shelves. Here we present an alternative, statistics-based method to model the most probable zones of the location of fractures and demonstrate our approach on all main ice shelf regions in Antarctica, including the Antarctic Peninsula. We can predict the location of observed fractures with an average success rate of 84% for grounded ice and 61% for floating ice and mean over-estimation error of 26% and 20%, respectively. We found that Antarctic ice shelves can be classified into groups based on the factors that control fracture location.

**Keywords.** Antarctic, ice shelves, glaciers, probability, calving, fracture nucleation, crevasse, logistic regression, bayesian

## 1 Introduction

In recent years, increased positive temperature anomalies have been observed in Antarctica (Jansen et al., 2007; Vaughan et al., 2003; Johanson and Fu, 2007; Steig et al., 2009) and future climate change in this area may be even more pronounced (Vaughan et al., 2003). This may cause the state of the Antarctic Ice Sheet to change significantly and could lead to a release of fresh water currently stored in the ice sheet; West Antarctica alone could contribute up to 4.3 metres to global sea level (Fretwell et al., 2013). Thus, understanding the factors that control the mass balance of the Antarctic ice sheet is crucial if we want to better understand the future impact of climate change and contribution of Antarctic ice mass loss to global sea level rise (SLR). Increased calving from the major ice shelves between 1998 and 2003 led to growing concern about the ice sheet stability (Shepherd et al., 2012).

Overall, research on crevasse propagation started as early as 1955 and calving parameterization has been under development for the last 20 years. It has been shown that the increased ice mass loss from Antarctica is caused primarily by an increased number of calving events in the last two decades which has led to significant ice front retreat (e.g., the collapse of Larsen B ice shelf and break off a part of Larsen C ice shelf (Mercer, 1978; Jacobs et al., 1992; Katz and Worster, 2010; Gudmundsson,

2013; Borstad et al., 2013; Hogg and Gudmundsson, 2017)). Studies by Jezek (1984); De Angelis and Skvarca (2003); Dupont and Alley (2005); Goldberg et al. (2009); Katz and Worster (2010); Gudmundsson (2013); Borstad et al. (2013) showed that increased calving can lead to destabilization of ice shelves and thus to a loss of the supporting mechanism (known as the "buttressing effect" or "back stress") they provide to inland ice in Antarctica. This support can be crucial for the overall stability

of the West Antarctic ice sheet as strong basal melting and reduced ice shelf buttressing can make the ice sheet unstable (Miles et al., 2013).

Developing a reliable calving law requires knowledge of where these fractures are located and how they evolve. Fractures in Antarctic ice shelves and the ice sheet are visible in satellite imagery, and can occur more often than every 50 m. Because of the size of Antarctica, the only feasible means of creating a data base of fractured zones is through the analysis of satellite imagery

and altimetry. However, fractures can be covered by snow and/or not be visible because of poor resolution of the available imagery. It is for these reasons that inverse methods are often used (Borstad et al., 2013, 2016). Because of the incomplete knowledge of the location of fractured zones in Antarctica, models to represent fracturing in ice shelves/glaciers remain poorly constrained. The modelling of initiation and propagation of fractures is an active area of research, however no models to date have been able to successfully model this process on an ice-sheet scale. Existing studies have focused on Greenland (e.g. Nick

et al. (2010); Cook et al. (2014); Krug et al. (2014); Sugiyama et al. (2015)), Svalbard (Chapuis and Tetzlaff, 2014) and the Antarctic Peninsula (e.g Otero et al. (2010); Bassis and Walker (2012)) and a method that can universally describe calving at any ice shelf in Antarctica has not yet been developed.

The aim of this study is to construct an empirical model that can predict the locations of fractures. We focus on modelling of crevasses (surface fractures less than 200 metres wide) on the surface of the Antarctic ice sheet and surrounding ice shelves. Our

model to predict fractured zones is based on a probabilistic approach, where we utilize the logistic regression algorithm (LRA) to find a relationship that enables the prediction of fracture locations. Our approach accounts for many potential parameters that include geometry, mechanical properties and flow regime (predictor parameters) and is based on a combination of modeling and remote sensing. We use a dataset of fracture observations, build by careful manual selection of the locations of visible fractures in satellite images, to built a model that can identify fractured regions on most of the Antarctic ice shelves as well

as grounded ice regions around ice shelves in Antarctica, including the Antarctic Peninsula. We compare the ability of our model to match observations of fractures from satellite imagery versus the predictive ability of the damage-based method of Borstad et al. (2013). From the modelling of 45 ice shelves/glaciers, we found that we can predict the location of fractures that match the observations with a success rate ranging from ∼45% to 99% (Figure 2a) for grounded ice and ∼30% to 90% for floating ice (Figure 2a). The average success rate when applying damage method to floating ice was equal to 34% in contrast

to 61% achieved when applying our LRA method. A mean over-estimation error of 26% and 20% for floating and grounded ice, respectively, occurs where we predict fractures in locations where no fractures are visible, but this may be affected by poor resolution of some of the satellite images and/or the presence of snow covering the surface of the ice. Our best results were achieved combining LRA with Bayesian as well as Jensen-Shannon Divergence theory described in section 4.

## 2 Background

### 2.1 Current state of calving computations in ice sheet models

A number of calving parameterisations have been developed and implemented in some software packages, but none of them includes the propagation of fractures both vertically and horizontally. Most of the available parameterisations are specific to a particular case and set of predictors (e.g (Pralong and Funk, 2005)), or calibrated for a particular location (e.g (Krug et al., 2014)), and therefore cannot be applied generically to any ice shelf.

Discrete Element Models are used to predict short-term calving events (HiDEM). Some other ice sheet models, such as ELMER/Ice, the Parallel Ice Sheet Model (PISM), the Community Ice Sheet Model (CISM) as well as the Ice Sheet System Model (ISSM), have attempted to include calving built on simplified physics (Alley et al., 2008): ISSM has a calving algorithm for marine terminating glaciers derived from a tensile Von Mises yield criterion (Morlighem et al., 2016). PISM uses a calving algorithm based on along- and across-flow strain rate (Levermann et al., 2012), which is based on the correlation between the calving rate and the first order approximation of local ice-flow spreading rate (it includes spreading rates of the first-order and assigns all higher order terms to zero). This idea is based on the observations of the increase of calving rate with along-flow ice shelf spreading rates and the spreading rates perpendicular to the calving front. However, it considers only large-scale behaviour and does not take into account the formation and propagation of crevasses. A second method implemented in PISM involves calculating a calving rate based on the critical ice thickness, which is mainly used to model calving of marine-terminating glaciers rather than floating ice shelves (due to different physics governing calving between the grounded ice and floating ice). Most of the experiments with ELMER/Ice calving were performed for Greenland glaciers, which have a different calving mechanism from the floating ice shelves in Antarctica (Van der Veen, 2002). The Community Ice Sheet Model (CISM) assumes that calving takes place when the water depth reaches a certain value (Price et al., 2014). This water-depth calving model uses flotation criteria to estimate the location of the glacier terminus. It allows calving to be linked to glacier dynamics as well as surface melting when applied to marine-terminating glaciers in Greenland (Nick et al., 2010). However, it cannot describe calving at floating ice shelves in Antarctica since the floating part is simply removed from the CISM model (the water-depth relationship requires a glacier to calve once it floats).

A number of other approaches have advanced our understanding of calving and the main existing studies are presented in Table 1. To date, Continuum Damage Mechanics (CDM) (Kachanov, 1958) and Linear Elastic Fracture Mechanics (Van der Veen, 1998a) are the most common methods used to model fractured zones, which is important information for modelling calving. The Linear Elastic Fracture Mechanics (LEFM) approach used by Krug et al. (2014) consists of calculating a stress intensity factor around fractures and assuming that they propagate until the factor falls below a certain critical value. To apply this method to any ice shelf, the modelled fractured zones need to be in good agreement with the observed surface fractures. Therefore, modelling the location of the fractured zones is an important basis for the subsequent estimation of fracture depth as well as calving and it must be described in the ice sheet models accurately. This method proposed Borstad et al. (2016) can describe both the formation and evolution of fractures in a fully viscous continuum damage model, although coupling with an elastic damage model might be more appropriate for representing short-timescale evolution of fractures. (Borstad et al., 2016).

Krug et al. (2014) built an alternative scheme by combining LEFM and CDM and found that they could match the observed evolution of a tidewater glacier in Greenland. This method is more complex compared to earlier approaches as it allows for both viscous and elastic behaviour and is able to reproduce development of small crevasses over a long period of time. The ELMER/Ice model (Gagliardini et al., 2013) combines CDM and LEFM to model calving, but this method has only been applied to Greenland (Krug et al., 2014). There are a number of studies that have proposed other calving laws (Pralong and Funk, 2005; Duddu and Waisman, 2012). These methods can include hydrofracture and other modes of failure, but have largely been applied to grounded calving margins, in contrast to the methods by Borstad et al. (2016) that are calibrated to remote sensing data and have been applied to ice shelves, but not grounded calving margins. Moreover, most of the mentioned methods might be not applicable in a generalised large-scale case.

## 3 Observational datasets

### 3.1 Ice Sheet System Model (ISSM) setup

Our statistical model is built upon knowledge of the velocities, stresses, strain rates, back stresses as well as friction coefficient and viscosity of the ice. We use ISSM (Larour et al., 2012) to derive estimates of these predictors for our statistical model. ISSM is a fully dynamic model that includes both two dimensional (2-D) and 3-D stress balance approximations. Our experiments rely on the shelfy stream approximation (SSA) as it is computationally cheap and suitable for modelling floating ice shelves and grounded ice streams undergoing widespread basal sliding.

All our experiments were performed for 45 ice shelf regions in Antarctica (see Figure 3a), each including both ice shelves and the grounded ice around 100 kilometres upstream from the grounding line (hereafter referred to as ice shelf regions or ice shelf/glacier). We ran one simulation to create a stress balance solution per region (ice shelf/glacier), which allowed us to obtain the predictor parameters required for the calculation of the probability of fracturing. We used SeaRISE air temperature, snow accumulation and geothermal heat flux (Le Brocq et al., 2010) as climate forcing data. The information about the ice temperature for grounded ice is calculate as the depth-average temperature from Liefferinge and Pattyn (2013); Pattyn (2010) (21 vertical levels). The steady-state depth averaged ice temperature on floating ice shelves (mainly used for the calculation of damage) was calculated using surface, basal temperatures and basal melting rate according to Holland and Jenkins (1999). To calculated ice temperature we corrected the surface temperature with a lapse rate and imposed it on the ice surface. Basal melting rates on ice shelves were taken from (Depoorter et al., 2013).

The data for the geometry of the ice shelves and surrounded grounded ice (bedrock topography, ice thickness and glacier surface) were interpolated from Bedmap2 data (Fretwell et al., 2013) at 1 km spatial resolution. Basal friction under grounded ice and rheology for floating ice were calculated from an inversion of velocities (Khazendar et al., 2007), where the observations of the horizontal ice velocities were taken from InSAR (450-metres resolution) (Rignot et al., 2011b, a) and the sliding law is the Budd sliding law (Budd et al., 1979). In the inversions we used regularisation to penalise sharp gradients of the cost function, calibrated using an L-curve analysis (Morlighem et al., 2013). We set boundary conditions as follows: the upper surface is considered stress-free and friction is applied at the ice-bedrock interface. At the inflow boundary we applied Dirichlet

conditions. The position of the grounding line position is calculated using a flotation criterion. We used the outputs of the simulations as input for our modelling using ISSM and as the predictor factors in our probabilistic method (see Section 4.1).

For all the simulations, we used a multi-resolution mesh approach for the chosen domains in East and West Antarctica as well as the Antarctic Peninsula. This method was chosen due to the fact that on the one hand using a 50- or 100-metre mesh resolution created a significant increase in the computational time of the model, but on the other hand it was important to have a fine resolution mesh in order to model surface fractures, as the distance between them is normally around 50-100 metres. In order to have a fine resolution together with smaller computational time, we first calculated all the main predictors on a 200-metre resolution mesh (to achieve a faster computational speed) and then interpolated the values to nodes on a 100-metre resolution mesh (to use in our fracture model resolved at 100-metres). All further computations and analyses were performed on this finer mesh.

## 3.2 Observing fractures using satellite images

We focus only on predicting the location of surface crevasses without modelling rifts, since the processes that cause rift opening might differ from processes that allow surface crevasses to stay open. In fact rifts might be formed due to presence of basal fractures, tidal deformation (Bromirski et al., 2010), ocean swell (Bassis and Walker, 2012) and stay open due to presence of melange (mix of snow and sea ice) (Rignot and MacAyeal, 1998; Larour et al., 2004; Bassis et al., 2005; Fricker et al., 2005). To model this we would need to include information about ocean temperature, sea ice and seismic activity, all of which is outside the scope of this paper. Moreover, rifts form when cracks propagate through the entire ice thickness and, therefore, the ice becomes effectively discontinuous. We, therefore, do not include rifts and focus only on surface crevasses.

In order to obtain the observations of the location of fractures on the ice sheet surface we used satellite images taken from Google Earth-Pro, where images of the Antarctic ice sheet were available at different spatial resolutions. However, to be able to see surface fractures we limited our choice to only images with a horizontal resolution smaller than 10 metres for the period between 2011 and 2015. We included only regions with at least one high resolution satellite image and where it was relatively easy to identify surface fractures.

The visual images of the ice surface include many features and it is important to distinguish the surface fractures from other patterns such as surface troughs (formed as a result of presence of bottom crevasses or subglacial channels). It has been suggested by Luckman et al. (2012) that wide features on the images with a large spacing between them (e.g > 1 km) are more likely to be troughs associated with basal crevasses. Modelling of basal fractures is outside the scope of this papers. Thus, such large linear features that are visible on the satellite images are surface troughs and should not be interpreted as surface fractures that our model fails to predict.

To construct a set of observed fractures, we manually selected fractured locations as well as non-fractured zones that we could identify in the satellite images. Most of the identified non-fractured regions are located in blue ice regions, which are areas with low snow accumulation or where the snow has been removed by the wind. In such areas we can clearly see where the ice is not damaged. It is important to note that in some locations the resolution of images was not always sufficiently high

to clearly see every fracture. Moreover, some surface fractures may be covered in snow and, therefore, are not identified by our analysis.

We constructed two different types of data sets: 'calibration' and 'evaluation', for building the statistical model and for studying the output of the model, respectively.

In the calibration data set we select a subset of observed fractures, being a representative sample of locations where fractures are found on 35 ice shelves/glaciers. The statistical approach requires training on a large number of ice shelf regions with different characteristics and a variety of observations. Therefore, we use our calibration dataset to train our LRA. This improves the reliability of the model, as the diversity in sampling provides a better estimation of correlation coefficients (called $\beta$ coefficients in LRA). We assign a value of 1 to fractured nodes and 0 to non-fractured nodes (due to the fact that Logistic

Regression Analysis (LRA) that we use in our approach, described later in section 4.1 uses a categorical input).

We form the evaluation data set to test how well our new approach predicts fractures for each ice shelf region individually. Although we did not need to select all the fractures on the ice sheet surface to build the calibration data set, to construct the evaluation data set we made a concerted effort to select the majority of the visible surface fractures on each of the ice shelf regions. It is possible that some fractures were missed due to the large spatial extent of the experiments. Moreover, we do

not present every fracture on the figures in this paper in order to make the figures legible. In addition, we perform validation experiments with another 10 ice shelf regions to test how well LRA works for a randomly selected ice shelf/glacier that was not a part of the construction of the model.

It is important to note that the evaluation data sets are not just discrete values (0 and 1), but are a continuous field representing the probability of observing a fracture in a location. In a node where we could see a fracture we assigned the probability of

20 observing a fracture to 1. Nodes around the observed fracture are more likely than not to be fractured. It is important to mention that the spacing of crevasses is often linked to their depth. A single crevasse can penetrate much deeper than a crevasse in a set of closely spaced crevasses. However, in this study we do not focus on estimating either depth or spacing of crevasses. Therefore, we then set the probability of observing a fracture to simply decrease from 1 to 0.55, decreasing with increased distance from the observed fracture (within 500 metres radius). On the other hand, if a non-fractured node was found within

25 a region with high resolution imagery, we assigned the probability of fracturing in this node to 0.05. Within a 500 metres radius of the non-fractured node we allowed the probability to increase linearly from 0.05 to 0.4. In all other nodes we set the probability of observing a fracture to 0.5. The last assumption is due to the fact if there are no fractures visible in the area of poor resolution of the image it is equally likely for the node to be fractured or non-fractured. This allows us to account for uncertainties of the observations, since it is not always possible to determine whether there are no fractures or whether fractures

are just not visible. We do not include any information about the depth and spacing of the crevasses.

## 4   Methods

We used statistics-based methods as an alternative to physics-based approaches in order to gain insights into the location of fractured zones in ice shelves and glaciers. In the well-known damage-based approach, the damage variable varies from 0 to 1

representing the fraction of a volume that is fractured, with 0 being not fractured and 1 being fully fractured. Instead of using damage-based method we use the Logistic Regression Algorithm (LRA), which provides us with the probability of fracturing (also varying from 0 to 1). We then apply this method to derive fracture likelihood functions for both floating ice shelves and the grounded ice for any ice shelf region. To construct the likelihood function we need to find coefficients that describe the relationships between predictor variables and what we want to predict (in our case it is surface fractures not including rifts). Thus, in order to create a statistical model we use our calibration data set of observations of surface fractures and non-fractures as well as information about the flow regime at the locations of each observation (predictor parameters).

Our main goal is to determine the most likely location of surface fractures. We do not focus on identifying the location of their initiation, since it is not possible to know whether the observed fractures were formed where observed or have advected to that position having formed upstream. We tried to select observed fractures where there were no other fractures visible upstream, meaning that the observed fractures would identify the initiation zones, but this may not always be possible. The model will, therefore, predict the locations of initiation of fractures but also some zones to which fractures have advected. For this reason, we do not distinguish between the high-advection (advection from upstream) and low-advection (because of local stresses) (Colgan et al., 2016) cycles. Although we do not directly model advection, the statistical model predicts the presence of both initiated and advected fractures without distinguishing one from another. The question that arises then is how do we know that the flow regime conditions that caused opening of the fracture are the conditions at the point where the fracture is observed and not the conditions upstream from the observed fracture (in case of advection)? However, even if an observed fracture was not formed at a particular location, but was advected with the ice flow, it is still visible on the satellite image. The fact that fractures can be seen indicates that there are factors at that location that act to permit the fractures to exist, whether they formed in that particular location or remained open after being advected from upstream (since another combination of factors could close the fracture).

This section is structured in the following order. First, we present our method (logistic regression algorithm) used for predicting the formation of fractures. Second, we describe the predictor factors (predictors) we include in this method. Then, two methods used for optimising a set of predictor factors are presented (Bayesian based algorithm and Jensen-Shannon Divergence). Finally, we present the damage calculation used for a qualitative comparison with our results.

## 4.1 The logistic regression algorithm (LRA)

Logistic regression is a statistical technique generally used to classify data based on values of input fields. The method is similar to linear regression but takes a categorical target field (in our case nodes which are fractured or non-fractured) instead of a numerical series. The logistic function allows us to calculate the likelihood of an event as a function of different predictor factors (see Table 2 for the predictors used in our model). Taking any range of data, it produces values from 0 to 1 and thus it can be used to represent the probability of fracturing (Hosmer Jr and Lemeshow, 2004).

To apply the logistic regression algorithm, we constructed a logistic function $P_j$ (Eq. 1) that describes the probability of a certain node to be fractured as a function of the predictor factors $x_i$. This function is not designed to provide any information about the depth of a fracture, just its spatial location.

$$P_j = Prob(X = 1|x_i) = \frac{\exp(\beta_0 + \beta_1 \cdot x_{1j} + \beta_2 \cdot x_{2j} + \beta_3 \cdot x_{3j} + ...)}{1 + \exp(\beta_0 + \beta_1 \cdot x_{1j} + \beta_2 \cdot x_{2j} + \beta_3 \cdot x_{3j} + ...)}, \tag{1}$$

where for $j$ is the node number, $x_{ij}$ is the value of predictor $x_i$ for node $j$ and $\beta$ are correlation coefficients.

The unknown coefficients $\beta_i$ are found by maximising the likelihood function $L$ (Eq. 2).

$$L(\beta_j) = \prod_{j=1}^{n} P_j^{\delta_j} (1 - P_j)^{1-\delta_j}, \tag{2}$$

where $n$ is the number of observations and $\delta$ is the Kronecker symbol:

$$\delta = \begin{cases} 1, & \text{when there is a surface fracture visible on a satellite image} \\ 0, & \text{otherwise.} \end{cases} \tag{3}$$

Once the values of $\beta$ are found we can find the probability of a node to be fractured by substituting a chosen set of predictors into Equation 1.

## 4.2   Predictor parameters:

We started with a set of 19 predictors, $x_i$. Some of them are known to influence fracturing (stresses, strain rates, ice rheology),
while others we considered to be potentially important (various geometrical properties, proximity to the ice front and the grounding line, etc). Temperature and accumulation were not included in the list of predictors due to the incompatibility of their spatial resolution with the relatively fine 100-metre mesh we used to model fractures. They might be important for the formation and propagation of fractures, as warmer temperatures can increase the number of fractures due to the effect of melt water (Weertman, 1973; Van der Veen, 1998b; Mobasher et al., 2016), but a better resolution climate dataset would be needed
to assess this.

All the experiments and sets of parameters used in LRA were constructed separately for floating and grounded ice. This is due to the fact that some parameters that were used for prediction of fractures on grounded ice are not applicable for predicting fractures on floating ice and vice versa (for example, friction and bed slope are irrelevant on floating ice, whereas back stress cannot be applied to grounded ice).
The calculation of some predictors was performed using methods already implemented in ISSM (e.g. stresses, strain rates, friction coefficient). Other predictors (e.g. calculation of curvature, distances to ice front, grounding line, proximity to glacier edges and nunataks) are not produced by ISSM and were calculated independently. Here we describe the methods we used to

calculate each predictor parameter as well as a brief description as to why each parameter may have an impact on the location of fractures:

(i) Principal values of the deviatoric stress and effective stress:

Following the Shallow ice approximation, the devatoric stress is:

$$\sigma' = \begin{bmatrix} \sigma'_{xx} & \sigma'_{xy} & 0 \\ \sigma'_{xy} & \sigma'_{yy} & 0 \\ 0 & 0 & -\sigma'_{xx} - \sigma'_{yy} \end{bmatrix} \tag{4}$$

The devatoric stress values have a direct effect on the opening and closing of crevasses; the sign of the first principal stress component determines whether it is compressive (negative) or tensile (positive). Effective deviatoric stress is calculated as:

$$\sigma'_e = \sqrt{\sigma'^2_{xx} + \sigma'^2_{yy} + \sigma'^2_{xy} + \sigma'_{xx}\sigma'_{yy}}, \tag{5}$$

where $\sigma'_{ij}$ are the deviatoric stress components.

Von Mises stress is calculated as:

$$\sigma_{vm} = \sqrt{\frac{3}{2}\sum_{i,j}\sigma'_{ij}\sigma'_{ij}} = \sqrt{3}B\dot{\varepsilon}_e^{1/n} \tag{6}$$

where $B$ and $n$ are the creep parameter and the creep exponent, respectively.

(ii) Effective strain rate:

The effective strain rate $\dot{\varepsilon}_e$ is included in our analysis because it is known that crevasse initiation is linked to strain rates (Campbell et al., 2013). If the strain rate in the horizontal plane is sufficiently high, crevasses can propagate to greater depth (Benn and Evans, 2010). In addition, stresses can trigger brittle fracturing but, to take into account a gradual viscoelastic effect that can lead to fracture formation, strain rates are included in our model.

The principal strain rates are calculated as eigenvalues of the matrix:

$$\dot{\varepsilon} = \begin{bmatrix} \frac{\partial u}{\partial x} & \frac{1}{2}\left(\frac{\partial u}{\partial y} + \frac{\partial v}{\partial x}\right) & 0 \\ \frac{1}{2}\left(\frac{\partial u}{\partial y} + \frac{\partial v}{\partial x}\right) & \frac{\partial v}{\partial y} & 0 \\ 0 & 0 & -\frac{\partial u}{\partial x} - \frac{\partial v}{\partial y} \end{bmatrix} \tag{7}$$

where $u$ and $v$ are horizontal components of the surface velocity.

Using again the shallow ice approximation, vertical shear is neglected and the effective pressure is approximated as:

$$\dot{\varepsilon}_e = \sqrt{\dot{\varepsilon}_{xx}^2 + \dot{\varepsilon}_{yy}^2 + \dot{\varepsilon}_{xy}^2 + \dot{\varepsilon}_{xx}\dot{\varepsilon}_{yy}}, \tag{8}$$

where $\dot{\varepsilon}_{ij}$ are the strain rate components (since in 2D we neglect $\dot{\varepsilon}_{xz}$ and $\dot{\varepsilon}_{yz}$ and using incompressibility $\dot{\varepsilon}_{zz} = -\dot{\varepsilon}_{yy} - \dot{\varepsilon}_{xx}$).

(iii) Horizontal strain rate gradient:

The change in strain rate sometimes is not the cause but the consequence of the presence of fractures. However, the aim of our study is to identify where fractures are present without attempting to fully describe the process by which they are formed. Thus, we use the change in strain rate as a predictor precisely because it tells us where we can expect to find fractures. This predictor allows us to discover new regions where crevasses are present even if they were not seen in the imagery. A lack of observed fractures but high strain rate means that fractures may not be visible but should still be present (e.g. if fractures are covered in snow or not visible due to bad resolution of the satellite images). Furthermore, changing geometry or boundary conditions can cause changes in strain rate, and also cause fractures (e.g. a glacier flowing over a convex slope or icefall: the change in bed slope causes a change in strain rate and also causes fractures, and it's not the fractures that cause the change in strain rate in this case.

(iv) Friction:

Low friction at the base of glaciers will lead to a higher sensitivity to membrane stresses, which can lead to more crevassing in tensile mode. We obtain this parameter from the inversion of surface velocities in ISSM.

(v) Stiffness of ice and ice thickness:

In addition, we include the viscosity parameter B in Glen's flow law as well as ice thickness due to their physical relation to fracture mechanics. When ice stiffness increases and ice crystals cannot creep fast enough, fracture may occur. Therefore, this parameter (obtained from the inversion of velocities implemented in ISSM) is added as a predictor. Adding temperature directly into the analysis did not improve the prediction results, which might be due to the uncertainties in the temperature estimation.

(vi) Proximity to glacier edges:

Generally the lateral friction along the glacier boundary is not considered in ice sheet models when stress is calculated. The stress field alone can predict transverse, longitudinal and radial splaying crevasses. They are all formed due to opening stress and are normally considered in existing damage modelling methods. However, the prediction of crevasses near the edges of glaciers requires a parameterisation of the lateral drag. Thus, we include the proximity to edges of glaciers and to nunataks as a predictor in our model.

(vii) Distance to the ice front and the grounding line:

We can see in the satellite images that more fractures are present at a certain distance from the ice front as well as near the grounding line. We found that the relation between the presence of fractures and distance to the ice front as well as the distance to the grounding line is non-linear (Figure 3b). For most ice shelves/glaciers we can see more fractures 3-5 km as well as 10-13 km away from the front and a slightly smaller number of fractures closer than 3 km to the front or between 5 and 10 km. Therefore, instead of using $d_{IF}$ and $d_{GL}$ (distance to the ice front and the grounding line in km) as predictor variables, we construct dummy variables: $DM_{IF}$ and $DM_{GL}$, respectively, which represent two-column arrays in the following form:

$$DM_{IF} = \begin{cases} (1,1), & \text{when } 3\text{km} \leq d_{IF} < 5\text{km} \\ & \text{or} \quad 10\text{km} \leq d_{IF} < 13\text{km} \\ (1,0), & \text{when } 5\text{km} \leq d_{IF} < 10\text{km} \\ (0,1), & \text{when } d_{IF} < 3\text{km} \\ (0,0), & \text{else} \end{cases} \tag{9}$$

$$DM_{GL} = \begin{cases} (1,1), & \text{when } 5\text{km} \leq d_{GL} < 15\text{km} \\ (1,0), & \text{when } d_{GL} < 5\text{km} \\ (0,1), & \text{when } 15\text{km} \leq d_{GL} < 20\text{km} \\ (0,0), & \text{else} \end{cases} \tag{10}$$

(viii) Bed and surface slopes:

There are a number of parameters such as surface velocity, surface slope and a curvature of a glacier channel that are included by other studies in the calculation of the stress field (Larour et al., 2012), but for our method we look at each component separately:

Thus, bed and surface slopes are included in the model since shear stress increases on a steeper slope and can lead to fracturing (e.g. ice fall is an extreme case).

(ix) Surface gradient change:

We include this predictor in the analysis due to the fact that fracturing can be caused by an increase in stress due to an abrupt change in surface elevation.

(x) Curvature:

It is clearly visible in satellite images that more fractures occur around horizontal bends in glaciers. Therefore, the curvature of the glacier channel was included as a predictor, calculated as:

$$\alpha = \arccos \left( \frac{\boldsymbol{v}(P) \cdot \boldsymbol{v}(E)}{|\boldsymbol{v}(P)| \cdot |\boldsymbol{v}(E)|} \right), \tag{11}$$

where $\boldsymbol{v}(P)$ is the ice velocity at the point of observations and $\boldsymbol{v}(E)$ is the velocity $D$ metres away from the point. The distance $D$ is based on the velocity magnitude $\boldsymbol{v}(P)$, because if the velocity is high we need to increase $D$ so that two subsequent points capture the geometry of the bend of a glacier. Thus, if $\boldsymbol{v}(P)$ is greater than 2000 m/yr we assign $D = 3\boldsymbol{v}(P)$, otherwise $D = 6\boldsymbol{v}(P)$. These values are not arbitrary: this assignment is used in the model only to have enough data points to see the local curvature of a glacier and it does not affect the calculation of the curvature itself.

Finally, due to the fact that all predictor parameters have different units, as well as significantly different magnitudes, we normalise each predictor as following:

$$x_i^* = \frac{x_i - \mu_i}{\sigma_i}, \text{ where } \mu_i \text{ and } \sigma_i \text{ are mean and standard deviation of the predictor variables, respectively.} \tag{12}$$

### 4.2.1 Test run with a small set of parameters

Including stress variables to predict fractures is intuitive as they are one of the major indicator of ice been fractured or non-fractured. Other variables such as geometry correlate to stress variables, but we found that it is important to include them in the model because the results are inferior if the parameters are not included. This might be caused by limitations of the predictor parameter values produced by the ice sheet model.

In order to show that our fracture model works better when including both physics-based and geometry-based predictors, we ran three additional experiments. In the first test run we included only effective deviatoric stress as a predictor and found that, although it produces reasonable results matching the observations, the success of identifying fractures is about 20 per cent lower that the results of the model with the chosen optimal set of the predictors (Figure 1 (a,b)). The second set of experiments contained only principal stresses and produced results that did not agree well with the observations (Figure 1 (c,d)). Similarly, the results of the third test, that included only von Mises stress as the predictor, did not agree with the observed fractures (success rate not exceeding 50 per cent, Figure 1 (e,f)). This shows that stress measures alone are not sufficient to model fractured zones and a more complex set of predictor parameters is required.

Moreover, including both friction and strain rate might be ambiguous since lower friction can lead to a larger strain rate. By looking at the predictor data sets we found that the optimal choice of parameters for each group includes either friction or strain rate, never both at the same time. We ran an experiment replacing strain rates by friction and found that the prediction success for some glacier decreases by only about 3%. We, therefore kept only strain rate as a predictor parameter and discarded friction.

### 4.3 Optimisation the choice of predictors

To construct the probability function for each glacier we sought a set (or subset) of the predictor factors required to include in the LRA. We started with a first guess (calculated using LRA and a potential choice of the predictor parameters) and then improved it based on three methods: random walk, Bayesian and Jensen-Shannon Divergence algorithm.

### 4.3.1 Random walk

For each of the 45 ice shelf regions we performed a 100000-step run with random sets of predictors used at each step (the number and selection of predictors were chosen at random at each step). We defined a potentially good model to be the one with a success of identifying fractures larger than 70% and the error of over-estimation not exceeding 15% (however, if a good-fit model was not found after 2000 steps we looked for a model with a 65% success and 20% error). Once a good fit was found, we saved it as a potentially good set and continued running the model with different sets of predictor factors for the remaining number of steps to search for a better model. At the end of each run the algorithm provided us with a mean set of factors for a best-fitting model.

### 4.3.2 Bayesian

To test the behaviour of the models with different sets of parameters and, thus, to choose with more precision an optimal set of predictors from the full set we performed a non-linear Bayesian inversion.

To find the likelihood function for a Bayesian inversion we need to add the probabilities of fracturing for all nodes. The area of each ice shelf region is $\sim 10^8 km^2$, which, when adding for all nodes, leads to a very large sum of all modelled probabilities (using LRA) for non-fractured nodes and therefore extremely large likelihoods (note that these values should not be confused with 0 and 1 values set for observed fractures only). In order to achieve a more realistic magnitude of the likelihood function, and account for the fact that in general there are more observed non-fractured nodes, we re-calculated the estimated LRA probabilities by scaling them between 0.55 and 1. To do this, we assigned all probabilities below 0.55 to the value of non-fractured nodes (value of zero) and scaled the remaining values to the range from 0 to 1.

In addition to defining a likelihood function, Bayesian inversion requires an input of prior model and prior scores. For a prior model, we took a calculated fracture probability $p^*$ from the best-fitting model obtained earlier using the random walk search. We assumed that the prior probability density function (PDF) was uniformly distributed between 1 and 17 (U[1,17], because the maximum number of predictor factors in a set is equal to 17).

Finding an expression for a likelihood function, $L_i$ for our model was problematic. We tested a number of different commonly used expressions, such as:

$$
\begin{cases}
L_i & = \sum_i \log(f_i), \text{ where} \\
f_i & = (1 - p_i)^{1 - d_i} + p_i^{d_i},
\end{cases}
\tag{13}
$$

$$\begin{cases} L_i & = \sum_i \log(f_i), \text{ where} \\ f_i & = (1 - p_i) \cdot (1 - d_i) + p_i \cdot d_i, \end{cases} \tag{14}$$

where $d_i$ and $p_i$ represent observed and modelled fractures on an ice shelf/glacier, respectively, and $f_i$ is the probability of agreement between two predictions in a cell $i$.

5    However, all of them produced very large likelihoods which increased dramatically with a small percentage change in the probability density function. The value of the likelihood function increased up to an order of $10^5$, which was unrealistic and made the inversions unstable. Therefore, it was crucial to choose a better representative likelihood function.

In order to construct the function, first we assumed that the measure $R$ of the total agreement between two models (the sum of all probabilities) followed a Gaussian distribution with a mean $E$ (Eq. 15) and a standard deviation $\sigma$ (Eq. 16):

$$10 \quad E(f_i^{pred}) = \sum_{i=1}^{N} f_i \tag{15}$$

$$\sigma = |E(f_i^{obs}) - E(f_i^{best})|, \tag{16}$$

15   where the two expected values for both data and the chosen model $E(f_i^{obs})$ and $E(f_i^{best})$, respectively, are defined as:

$$E(f_i^{obs}) = \sum_{i=1}^{N} d_i^2 + (1 - d_i)^2 \tag{17}$$

$$E(f_i^{best}) = \sum_{i=1}^{N} p^* \cdot d_i + (1 - p^*) \cdot (1 - d_i), \tag{18}$$

where $p^*$ is the best-fit probability.

20   Second, our idea was to calculate the likelihood $L_i$ as an exponential function of the misfit $\phi_d(m)$ between the data and the model, assuming that either the data (observed fractures) or the analysed model contain an error (Eq. 19):

$$L_i = e^{-\frac{1}{2}\phi_d(m)}, \text{ where} \tag{19}$$

$$\phi_d(m) = \frac{(E(f_i^{obs}) - E(f_d^{pred}))^2}{\sigma^2}. \tag{20}$$

We performed a Bayesian analysis for 500 steps, then narrowed down the selection and accepted only those models that had likelihoods greater than 90% of the best likelihood. Each step included two criteria: if a new likelihood was greater than the prior likelihood or was greater than a certain percentage (taken at random at each step) of the old likelihood we accepted the model. This allowed us to identify the most commonly chosen sets of parameters.

## 4.4 Glaciers classification and Jensen-Shannon Divergence (JSD)

In order to select a set of predictors for a general case and to find whether it is possible to identify a set that can be used for any ice shelf region, we started with the construction of a binary array for each ice shelf/glacier, where the number of rows represents the number of good-fitting models for an ice shelf/glacier and the number of columns represents each of the predictor factors.

We then found the average occurrence of each predictor:

$$A_i = \frac{1}{N} \sum_{j=1}^{N} k_j, \tag{21}$$

where $i \in [1, 17]$ is the predictor index, $N$ is the number of good-fitting models. $k_j = 1$ when the predictor is included in the good-fitting model $j$ and 0 otherwise.

We could then determine how often a certain predictor was included in the good-fit models. If a predictor was selected more than 50% of the time then it was assigned as a potential for the best-fitting model. Thus, we obtained a $45 \times 17$ array (45 glaciers vs. 17 predictors) that consisted of 1 when the predictor was included in the best-fit model and 0 otherwise.

Then, we classified the glaciers in groups. There were a large number of possible combinations to select such groups. Therefore, we constructed a test that assessed every possible combination and calculated a percentage of similarity between glaciers in a group (Eq. 22).

$$S = \frac{M}{17} \times 100, \tag{22}$$

where $M$ is the number of matches between sets of predictors for two glaciers and $S$ is a group number.

Finally, we found that we could categorised all 45 glaciers in 4 different groups, with Group 1 having glaciers/ice shelves that can be more easily combined and Group 4 being a more narrow group of specific glaciers/ice shelves that can not be placed in any of the other three groups.

The Jensen-Shannon Divergence method (JSD) (Dagan et al., 1997) can be used as a tool to measure the distance between two distributions and can provide a value that we can use to assign a particular glacier/ice shelf to one of the groups. The JSD formula is widely used in statistics to measure a divergence of one probability distribution from another. We applied JSD to identify the similarity between the best probability for each glacier and a probability calculated by placing the glacier in a certain group.

The Kullback-Leibler divergence (Kullback and Leibler, 1951) is defined as:

$$JSD(P_1||P_2) = \frac{1}{2} D(P_1||M) + \frac{1}{2} D(P_2||M), \tag{23}$$

where two conditional PDFs $D(P_1||M)$ and $D(P_2||M)$ are defined as:

$$D(P_1||M) = \sum \left( P_1 \log \left( \frac{P_1}{M} \right) \right) \tag{24}$$

and

$$D(P_2||M) = \sum \left( P_2 \log \left( \frac{P_2}{M} \right) \right), \tag{25}$$

where $M = \frac{P_1 + P_2}{2}$ and $P_1$ and $P_2$ are the new probability (when assigning a glacier/ice shelf to a new group) and the old probability (the best-fit model), respectively. Both probabilities $P_1$ and $P_2$ have to be normalised before applying Equation 23.

### 4.5 Calculation of Damage

Here we utilise the damage-based model as an independent method in order to compare it with our statistics-based method. We do not compare our probability-based model with the damage model directly; rather, we evaluate their respective ability to predict the formation of fractures in ice. For this we compare calculated damage with the observations of fractures and identify areas where it can and cannot accurately predict the presence of fractures.

The Continuum Damage Mechanics approach, based on the method suggested by Kachanov (1958), includes estimation of damaged zones where the ice is softened due to the presence of fractures. Continuum Damage Mechanics has been successfully applied to a wide range of engineering problems as well as to model damage at individually selected ice shelves such as the Ross, Filchner-Ronne, Amery (Bassis and Ma, 2015), Larsen C (Borstad et al., 2013) and Larsen B (Borstad et al., 2016) ice shelves.

Damage is a scalar variable used to determine failure of ice and the nucleation of fractures, usually when the damage predictor reaches a certain value. There are two different methods for inverting for damage: methods applied to invert for damage and methods used to model damage propagation in ice sheet models. Borstad et al. (2012) suggested a direct inversion

for damage using a cost function. Later, Borstad et al. (2013) proposed a method to calculate damage as a post-processing routine after inverting for the ice viscosity. In the first stage, inversion of ice rheology is performed following Larour et al. (2005), then damage is calculated from the inversion of velocities at the ice shelves, which is based on minimising the cost function that quantifies the misfit between the observed and modelled surface velocities.

The second stage includes the propagation of damage. (Krug et al., 2014; Albrecht and Levermann, 2014) proposed to model the propagation of damage using an advection scheme and a source function. The main limitations of this method are the choice of what should be used as a source function as well as the number of decisive parameters that define the damage evolution (damage threshold, initiation threshold and the enhancement factor). The source function is the controlling factor in the damage propagation and Pralong et al. (2003) as well as Krug et al. (2014) proposed a source function definition. Both

of the approaches work well for particular regions, but control predictors in this model have been derived using data from only one specific glacier in Greenland (Duddu and Waisman, 2012) and through the use of small-scale laboratory experiments (Pralong and Funk, 2005) and have not yet been generalised to be applicable to all ice shelves/glaciers. These models do not yet account for factors such as ice fabric and impurities (Borstad et al., 2012).

     Recently, Borstad et al. (2016) proposed a framework where, instead of computing a damage source term as is usually done,

damage is part of a generalized constitutive relationship. This approach has a number of advantages as it allows the calculation of mechanical ice weakening and the prediction of the degradation of ice shelves. This can significantly improve the accuracy of identifying zones where the ice is weakened by illuminating the uncertainties related to the source function. The main weakness of the approach lies in determining the constant parameters that define damage, because the validity of the parameter values can only be tested when an ice shelf undergoes pronounced mechanical changes, as did the Larsen B Ice Shelf in 2002.

The parameters suggested by Borstad et al. (2016) have not been tested for other ice shelves apart from Larsen B, and so it is unknown whether the approach is valid for other locations and settings.

     In this study we use the damage inversion method proposed by Borstad et al. (2013) to identify regions where fractures are initiated. Damage in this context has no vertical coordinate, but comes from a linear mapping of the depth-integrated shallow-shelf equations. This approach is performed in two steps. First, the inversion of rheology based on the misfit between observed

and modelled velocity is performed on floating ice. Then, damage is calculated as:

$$D = 1 - \frac{B_I}{B_T}, \tag{26}$$

where $B_I$ and $B_T$ are viscosity parameters calculated from an inversion and initialisation of viscosity based on temperature analysis, respectively.

     It is important to keep in mind that the inversion only infers damage in areas where fractures (crevasses or rifts) are being

actively formed and, thus, creating a jump in strain rate/velocity. Many rifts are formed at one point in time and then only intermittently propagate. If the velocity observations do not show a discrete jump across a fracture, then there is nothing for the inversion to pick up in terms of damage. It only finds fractures that are actively enhancing the flow and it is not meant to locate every fracture.

Estimation of $B_T$ is the source of the main uncertainty in damage calculations due to the lack of ice temperature data, which can be crucial in affecting the accuracy of the viscosity parameter (Bassis and Ma, 2015). Thus, the errors in assumed temperature may affect the inferred value for damage.

Fractures that have been advected can be identified by damage but this is not always the case, due to the fact that the inverse method for calculating damage will only find damage where there are fractures that give rise to velocity gradients. Damage will capture some fractures that were formed upstream and advected to a region with different stress conditions only if the fracture enhances the flow and creates a local velocity gradient. Thus, we first calculate flow lines for each observed fracture. If upstream from the fracture the damage is larger than 50% we assume that the observed fracture was formed upstream, that the damage calculation may be correct and that the observed fracture was formed upstream. If there is no damage initiated at the point or damage upstream from the observed fracture we assign the observation point as not captured by the damage method and consider this as a failure of damage to identify the fracture (which can be due to the fact that the fracture in observation point does not cause a local gradient in strain rate) .

Physics-based methods, such as Linear Elastic Fracture Mechanics (LEFM) and Continuum Damage Mechanics (CDM), are necessary when modelling fractures in Antarctica. We do not intend to substitute these methods; rather, we seek a method that can improve on some aspects and cases when physics-based models do not predict well the formation of fractures. In particular it is possible that some fractures are initiated upstream from the grounding line rather than on floating ice. It is therefore important to be able to predict the formation of fractures in both cases. Damage is calculated only on floating ice based on model inversions using the Ice Sheet System Model (ISSM) (Larour et al., 2012) because it is not possible to distinguish between basal friction and damage on grounded ice, as they have similar effects on the ice velocity. Thus, the main motivation of this study is not to replace the damage approach, which in fact provides a strong physical background for ice sheet modelling, but to find an alternative method that can be applied to both ice shelves and grounded ice, can work for a large set of glaciers/ice shelves and does not depend on temperature observations and threshold parameters.

## 5   Results

We applied the LRA method combined with the random walk method to 45 ice shelf regions that include both ice shelves and surrounding grounded ice (the corresponding names and locations can be found in Table 6 and Figure 3a, respectively) and found a best-fitting model for 44 of them. The fracturing of the remaining ice shelf cannot be described using the predictors we have, producing unacceptably large or small probabilities.

In total, for each ice shelf/glacier the random walk analysis gave a number of possible sets of predictors that can produce a good-fitting model. We combined all of these possible sets for each glacier to see which predictors are always present in the good-fitting model and which ones are never included. The results of the random walk and the Bayesian inversion agreed well. Most of the essential predictors for each particular glacier selected in the Bayesian approach were also chosen when performing the random walk. In most cases, the Bayesian analysis showed equal importance of most of the predictors although effective strain rate and velocity had a slightly higher rate of selection. There was no universal set of factors that could be used to model

all ice shelf regions. However, subsets of glaciers had some similarities in terms of the predictors that had to be included in order to achieve a good-fitting model.

To estimate how well our probability model and the damage model identify observed fractures we calculated the percentage of success and error for each ice shelf/glacier model. First, we found the number of cases when there is a modelled fracture in the vicinity of an observed fracture (within 100-metres radius). Then, we divide this number by the total number of observed fractures to find the percentage of success. To find the percentage of failure we calculated how many times there is a modelled fracture when there are no observed fractures within a 100-metres radius. We divide this number by the total number of non-fractured nodes to find the failure percentage.

Thus, we categorised the 45 glaciers into 4 groups, requiring that the deviation from the best-fit models did not exceed 5%. Next, we performed a test to assess whether these selected sets were the optimal choice, by estimating the deviation from the best solution using the Jensen-Shannon Divergence algorithm. We assigned each glacier to a particular group based on its minimum value of the deviation from the best-fitting model in JSD analysis. In so doing, we slightly modified the members of each group that we had previously created. For example, Glacier 27 belonged to Group 1 previously and it fit well with only a slight change of the best-fit score. However the JSD showed that if we move this glacier to Group 2 the deviation from the best-fit decreased from 0.01 to 0.003. However, we had to take into account the fact that the JSD algorithm measures the total distance to the best-fit probability and, thus, can decrease the over-estimation error while, at the same time, significantly decrease the success rate (Glaciers 10, 13, 15, 11, 30, 32). Therefore, since these six glaciers were of a similar type to the glaciers in Group 1 and their JSD was similar for Group1 and Group 2 (e.g JSD=0.02 in Group 2 and JSD=0.0205 in Group 1) they were assigned to Group 1 to avoid a decrease in the success rate of identifying fractures.

Finally, to reach an optimal agreement between our model and the observations of fractures, we assigned each glacier to a particular group and the set of factors for each group are presented in Table 3 and Table 4, 5, respectively. We found that the sets of predictors for each group varied significantly, however surface velocity was included in the grounded ice set for all groups. For the floating ice the analysis showed that surface velocity was not a determining factor in predicting fractures, instead effective strain rate as well as deviatoric principal stress values were present in each predictor set.

While the success of identifying fractures on floating ice was lower than for grounded ice, we were still able to identify the main fracture patterns and the success rate was high for the majority of ice shelves (see Figure 2a). Our method is able to identify up to 99% of the exact location of fractures on grounded ice with an average of $84\%$ (Figure 2a) and 61% for floating ice (Figure 2b. The mean over-estimation error for grounded ice and floating ice was $26\%$ and $20\%$, respectively. There are many cases where our method agrees with the results produced by the damage-based approach. However, in almost all cases the success of LRA on floating ice was higher than of the damage-based method with the exception of two glaciers. Overall, for all four groups, where we could not achieve a high score using LRA, the damage-based method did not produce a high success score either (see Figure 2b).

## 5.1 Group 1

This was the largest group of glaciers and the best-fit model includes as many as 10 predictors for grounded ice and seven predictors for floating ice. The analysis of the estimated coefficients in LRA showed that predictors with the highest weights in our model for this group of glaciers were: effective strain rate, proximity to glacier edges and nunataks as well as the surface elevation gradient. We present the modelled probability of fractures in Figures 5b and 4c as well as comparison with the damage-based results in Figures 7a and 10c.

The main pattern of surface fractures is well represented for this group. On grounded ice the success of identifying fractures is larger than 88% with a quarter of glaciers at almost 100%. The failure related to over-estimation of fractures is 27%. On floating ice the success amounted to 55% and the failure was equal to 15% on average. For Vanderford IS (see Figure 5b) the overall pattern is well represented, even though high resolution images were not available for this glacier. The over-estimation error is mainly related to the region that is far from the ice front and has a relatively high accumulation rate, possibly obscuring the fractures in the imagery. On floating ice the probability of fracturing is relatively smaller, mainly showing a higher chance of fracturing closer to the groundling line. Conversely, Drygalski Ice Shelf has a larger number of high resolution areas and, as a possible result, less over-estimation of fracturing (see Figure 4c). We can see that the "definitely non-fractured nodes" (selected in blue ice areas) are successfully represented in our model. For this glacier, none of the observed non-fractured nodes was assigned to have a high probability of fracturing, with the modelled probability being as low as 0.1. Moreover, in the regions with a large number of observed fractures, the probability is as high as 0.9 and it is slightly lower in the areas with a smaller number of observed fractures (between 0.6 and 0.8). Observed fractures not captured by our model were not captured by the damage-based model either.

The modelling results for the Cook ice shelf are shown in Figure 7b. There are distinct fractures visible towards the front and in the central part of the ice shelf that are not captured by either approach, which we interpret as showing that most of the fractures are formed further upstream near the groundling line. In general, the probability and the damage-based models show good agreement on the floating ice near the grounding line. However, damage does not reach 50% in the majority of the locations. Moreover, in many locations where rifts are visible it shows 0 damage.

The modelling results for Larsen B IS are illustrated on Figure 10c. It is clear that the nodes where damage is high have a high probability of fracturing due to the fact that we added damage as one of the predictor parameters to this glacier. It can be also seen that there are two lines of high probability of fracturing that coincide with the location of the large rifts that can be seen on the satellite images.

The results for Nansen IS (Figure 8c) as well as for Pine Island (Figure 6a) agree well with observations even though the data from these two glaciers were not included in the calibration data set used to construct the LRA model. For Pine Island, we observe fractures in the central part of the shelf that were not captured by the model, but our model predicted high probabilities of fracture upstream where the ice is grounded. Thus, these fractures are likely advected from the grounding line out onto the floating ice shelf.

## 5.2 Group 2

The model for the second group of glaciers has the best-fit when the bed slope is excluded. Effective strain rate and surface slopes were found to be the most important predictors in the model for this group.

For this group the LRA method predicts fractures with a 70%-90% success on grounded ice and finds about 67% of observed fractures on floating ice with an over-estimation of 25% and 27%, respectively. In most cases the model represents the non-fractured nodes with high precision, except for the slight over-estimation at the front of the ice shelf. Similar situations are observed for most glaciers in this group: the area of floating ice is relatively small, thus the main prediction is performed for grounded ice. For, example, for Edward VII IS and Rayner Thyner IS (Figure 4a and 4b, respectively) the modelled probability captures most of the fractured as well as non-fractured nodes on grounded ice with the exception of a few very small regions that are outside of the high resolution image areas.

Interesting results were found for Larsen A IS (see Figure 5a), showing a very good agreement between our model and the observations. We can see that the nodes observed to be non-fractured and located within the high-resolution regions have lower probabilities of fracturing predicted by the LRA method. Both the LRA and damage-based results for Shirase IS (Figure 11a) are similar on floating ice, although LRA method captures a slightly higher number of fractures. We infer that most of these fractures are formed further upstream on grounded ice.

## 5.3 Group 3

This group includes 4 ice shelf regions, namely Totten IS, Nivl IS, Dibble IS and Holmes IS. These glaciers were very sensitive to the choice of predictor factors and the JSD process could not assign them to either of the two aforementioned groups. The mean success rate for this group was around 93% with an over-estimation rate just above 23% on grounded ice, and 56% and 23% success and error on floating ice, respectively. Potentially, in the model for this group we could include the proximity to the ice front since it produces slightly better results for 3 of the 4 glaciers. However, it lowers the success rate for Nivl IS significantly. Thus, in order to achieve a set that would give a good-fitting model for all of the glacier we exclude back stress and the ice front proximity from the list of predictor factors for this group.

In terms of results for Holmes IS (Figure 8a), there was good agreement between damage and LRA models as they both were able to predict the main fracture pattern. However, the LRA predicted the observed fracture pattern slightly better, especially at the front of the ice shelf. Similarly, for Dibble IS (Figure 9c), both methods produced similar patterns, although the LRA method had a higher over-estimation error. Nevertheless, the LRA method was able to more precisely estimate fractures at the western part of the ice shelf.

Finally, we present the result for the Totten IS (see Figure 11c). The images we used for this glacier were very hard to interpret due to the presence of many features on the ice surface as well as the low resolution of the imagery. Interestingly, for the Totten IS, only a certain set of factors can produce a good fit to observed fractures. For this glacier, including back stresses in the model produces a slightly smaller number of modelled fractures. Both the LRA model and the Damage model capture most of the fractures we could observe on the floating ice, displaying similar distribution patterns. Although, the over-

estimation is relatively high on grounded ice, it is not possible to say whether there are no fractures there and that our model shows over-estimation for this glacier or fractures are not visible due to the low resolution images.

## 5.4 Group 4

This group includes Larsen C, Amery, George IV and Borchgrevnik IS. The average success and error for this group amounted to 66 and 15% and 56 and 20% for floating and grounded ice, respectively. The most important predictor factors in this group for floating ice are effective strain rate, surface gradient change and ice thickness, while for prediction of fractures on grounded ice curvature and surface velocity need to be included in the model. For all of the ice shelves/glaciers in this group we found that including the ice front and grounding line proximity distorts the model, increasing significantly the error due to over-estimation of fractures. For Borchgrevnik IS it also led to a drop of the success rate of fracture prediction. In addition, Larsen C and Amery Ice Shelves can be grouped together but cannot be included in any of the groups mentioned above. For these glaciers only a small number of predictors needed to be included in the model. The Bayesian analysis also confirmed the sensitivity of the Amery fracture model to this set of predictor factors. The LRA model for the Amery IS (Figure 6b) was able to capture most of the fracture pattern on the grounded ice and near the edges of the floating ice and demonstrated a similar pattern to the damage model. For George IV (Figure 9a) we observe a very good agreement with observations both on floating and grounded ice. There is a high probability of fracturing in most locations where the high damage zones are predicted. Overall, for both ice shelves we can see that the majority of fractures are formed further upstream of the grounding line.

## 6 Glacier Characteristics

### 6.1 Group 1

The ice shelves/glaciers in this group have a number of characteristics that distinguish them from other ice shelves/glaciers. Most of them are relatively wide with a large floating area. The floating part is not restricted by any channel walls and the width of the shelf is similar to its length. All glaciers in this groups are relatively static, with less curvature or significant surface elevation changes. However, there were two exceptions: Abbot IS and Drygalski IS have slightly different characteristics. First, Abbot is a wide glacier that exhibits most of the properties of Group 1. However it has a large number of glaciers that restrict its outflow towards the ocean and, therefore, it has similarities with the glaciers from Group 4. This observation is in good agreement with the JSD results that showed that Abbot IS could be assigned also to Group 4 as the change in JSD distance in this case would be very small. Second, the JSD results showed that Drygalski IS could be as well placed in Group 2 or Group 3. This ice shelf has some characteristics similar to Group 2 (large number of nunataks) and Group 3 (a very long floating tongue). Therefore, we suggest that some glaciers have mixed features of Group 1-Group 2 (such as Vanderford) or Group 1-Group 3 (Ekström, Tracy-Tremenchus, Rennik), however they still have more characteristics of Group 1 and produce better-fitting results when assigned to this group.

## 6.2   Group 2

This group includes a relatively smaller number of ice shelves/glaciers. All of the ice shelves/glaciers have a large number of nunataks and smaller ice thickness as well as many small narrow channels and fast ice streams. They are mostly located on the Antarctic Peninsula or near the Trans-Antarctic Mountains. All of the ice streams are relatively steep which may explain why it is necessary to include surface slopes in order to achieve a good-fitting model.

## 6.3   Group 3

Group 3 glaciers were found to have many similar features. Most of the ice shelf regions in this group have one relatively long glacier that flows inside an embayment. For most of them the ice shelf is much longer than it is wide, and they all have a very low glacier channel curvature. The surface velocities of these ice shelves/glaciers are relatively high, which explains why changes in strain rate and surface velocity are the most important predictors for this group.

Interestingly, although the average back stress for Totten IS is one of the highest out of all 45 ice shelf regions, including it in the model does not significantly change the fracture probabilities. Thus, apparently, even though predictors may have large magnitudes, they can make just minor contributions to the constructed probability and other predictors dominate the fracturing process for the Totten IS. The effective strain rate is also one of the highest for Totten, but we found that it is not this predictor that mostly contributes to fracturing, rather it is the effective strain rate gradient. Thus, sudden changes in the flow regime of the glacier would be the most likely cause to promote an increase of the number of fractures.

## 6.4   Group 4

The JSD analysis has shown that Borchgrevnik IS could also be assigned to Group 1, but it produced slightly better results being placed in Group 4. On the other hand, Amery and Larsen C need to be strictly assigned to Group 4 only. George IV IS and Amery IS have similar characteristics as they are both narrow and long (in fact much longer than any other ice shelves of this type in Antarctica) and are located inside an embayment. Although Larsen C IS is not inside an embayment, it is a significantly long and narrow ice shelf stretching around the coast. Borchgrevnik IS also has similar features to the Amery and George IV ice shelves as it is of a similar shape and is located inside a narrow channel. However, it does not have exactly the same characteristics as the other ice shelf regions in this group as it is much shorter, which could be why JSD showed that it could also be placed in Group 1.

On the Amery IS (see Figure 6b) most of the fracturing occurs upstream of the grounding line. There were a number of fractures (right hand side of Figure 6b, close the the ice front of the glacier) that could not be represented by our models. However, the uncertainty of the fracture observations in this area is high due to the difficulty distinguishing them from the surface features caused by basal crevasses.

Adding the proximity to the grounding line and the ice front as predictors did not produce a good fit for the Borchgrevnik IS because of the specific shape of this region. The distance between the ice front and the grounding line is very small relatively to other glaciers in our analysis.

## 6.5  Discussion

We found that, in general, the most important predictor factors to model surface fractures on grounded ice for all analysed glaciers were the surface velocity and the change of the surface gradient, which is in agreement with the theory of possible mechanism of fracture formation (Colgan et al., 2016). Interestingly, the required parameters on floating ice were different from grounded ice, with effective strain rate and principal stress being the most important. Previous analysis based on damage accounts for effective stresses, thickness and viscosity, but does not include such predictors as proximity to glacier edges, nunataks and the grounding line as well as the curvature of a channel, which helped to improve the modelling of fractures on most ice shelves in our analysis. Our results can be used to identify potential regions with snow covered crevasses that may pose hazards for navigation in Antarctica. Many researchers use ground penetrating radar to find hidden crevasses, but it is a real time assessment method that requires both financial and human resources and, therefore, can not cover all the areas in Antarctica. Our approach can be done remotely and at low cost, in advance of field campaigns.

We do not claim that all the predictors that were chosen in the final set for each group represent the exact fracture mechanisms for each glacier. For some ice shelf regions, sets containing different predictors can lead to results close to the best-fitting model. However, for some cases, such as Amery and Totten Ice Shelves the number of good-fitting models is very limited. For example, including the effective strain rate and proximity to the ice front in the analysis we can achieve a better fit to the observations. Therefore, we conclude that some factors have a very strong effect on fracturing, while others are only minor for some glaciers. Ultimately, we seek only to be able to develop a model that can identify correctly the geographical location of fractures, not necessarily explain why they are there.

The main uncertainty of our method is related to the over-estimation of the number of fractures. It could be argued that we predict fractures in the locations where no observations of fractures are detected due to the fact that they are not visible due to snow accumulation or coarse resolution. A possible solution to this could be to supplement satellite images with radar and seismic measurements (Navarro et al., 2005; Delaney and Arcone, 2005) or to acquire higher resolution satellite images that can help to identify location of fractures even if they are hidden under the snow surface. However, for our method it would require a large set of observational data. In our paper we assume that 20-25% rate of over-estimation is reasonable, since most of our results show over-estimation when predicting fractures in the areas around observed fractures. Our assumption is based on the fact that the ice regime conditions are similar within a 500 metres radius, not implying any direct influence of the old fractures to the new fractures (Colgan et al., 2016). In addition, the area of high probability of a fracture is larger than the number of observed fractures mainly due to the fact that it is not possible to select all of the fractures on the satellite images manually. The observed fractures in or evaluation data set capture the main areas of fracturing assuming that surrounded nodes are likely to be fractured as well.

A significant over-estimation of predicted fracturing can be seed at the front of Vanderford IS (see Figure 5b), suggesting that there is a very high chance of developing surface fractures in that area. It may be that fractures exist there that are just not visible on the satellite images. In fact, that region has a relatively high snow accumulation rate reaching 1 m/yr. After closer inspection of the satellite image areas where we see the over-estimation error, we can recognise the presence of surface

fractures and we may have under-identified existing fractures in that area. Due to a very large number of fractures around Antarctica, the manual selection of all the fractured data points is a time intensive process. Sampling with a higher spatial density would require an automated algorithm. However, the low spatial density sampling does not influence the results of the fracture probability (as mentioned previously only diversity in sampling is important), it affects only the observations data set
that we use to estimate the success of our model. Thus, under-identification of fractures can lead to an apparent over-estimation of the error.

The damage-based approach sometimes produces areas of high damage downstream of the observed fractures (Figure 9d). If we assume that there are some fractures that are not visible on the images it is still unclear why the modelled ice is not damaged upstream where observed fractures are present. If we could see damaged ice upstream from the observed fractures
we could assume that the ice was damaged and transported downstream where we can see fractures. However, in many results based on the damage approach we could not identify this type of behaviour. Even after correcting the observations of fractures by back integrating the flow of ice, we still can find fractures that do not have zones of high damage upstream of the fractures. In addition, the method based on damage inversion predicts only damage on floating ice, whereas fractures are often formed upstream of the grounding line. In our probability-based model this type of behaviour is accounted for and the model in
most cases overestimates fractures only in the vicinity of or upstream of observed fractures. In Figure 8a we show modelled probability of fracturing for Holmes IS. The over-estimation rate is quite small and it all occurs upstream of the observed fractures, which might be due to fractures that were formed further upstream being transported downstream where they are visible on the satellite images.

We looked at various properties of Ronne IS for which we could not find a good approximation using any of the 17 predictors.
We found that the Ronne IS has the lowest elevation change as well as the principal stresses components. We do not have enough samples to cover values that are non-typical for the majority of glaciers, which may explain why we could not find a good-fit model for this ice shelf, neither with LRA nor using the Bayesian analysis. Thus, we conclude that our probabilistic model is not appropriate in this case.

## 7   Conclusions

Most previous large-scale modelling of surface fractures has focused on applying zero stress, Linear Elastic Fracture Mechanics, Continuum Damage Mechanics. We have shown that, using the suggested nominal parameters, damage-based approach does not fully reproduce the location of fracturing for any ice shelf region. In this study, we constructed a probability-based method to model surface fractures and generated improved predictions of fractures when physics-based models did not predict well the location of surface fractures. From this different perspective, we can construct an alternative method to predict the
location of fractured zones not only on floating ice but also on grounded ice.

We found that the Logistic Regression Analysis, combined with other statistical methods, can significantly improve the prediction of fractured zones for the Antarctic ice shelves/glaciers and can lead to the identification of up to 99% of observed surface crevasses for some ice shelf regions with an average of 70% for all ice shelf regions. Our approach has a number of

uncertainties and leads to some over-estimation of the number of fractures in comparison to the observations, but the rate is not significantly higher than the over-estimation error found when using the damage-based method. However, the damage-based method did not predict location of many fractures either upstream or downstream from the observed locations, suggesting an underestimation when applying the damage method. The probabilistic results suggest that our statistics-based methods are more reliable in identifying fractures and rifts in the locations were the damage method does not predict them (which is not related to a failure in the damage method, but the fact that it is not constructed to do so). There are also uncertainties in the damage-based method related to the surface temperatures in Antarctica, which may be poorly represented with available observations. It is possible that damage-based method needs to be tuned for every ice shelf separately, but this is beyond the scope of this study.

We classified the Antarctic ice shelf regions into 4 groups, where ice shelves/glaciers in each group have similar characteristics and each group has a set of predictors that can be used to predict the location of fractures. Although, there were ice shelves/glaciers of specific shapes and having specific regimes that are more difficult to describe applying the general set of factors suggested in this study, overall our method provides a tool that can be used in the analysis of fracturing for most of the ice shelves/glaciers in Antarctica.

This statistics-based method can help to expand our current knowledge of the crevasses as well as improve mapping of potential hazards. Our model is easy to implement and can be effectively used as a basis for modelling of fracture propagation and the first step in implementing a calving parameterization in ice sheet models.

# 8 Tables

**Table 1.** Development of calving parameterisations

| Year | Reference | Method |
|------|-----------|--------|
| 1955 | Crevasse penetration depth using tensile stress and overburden pressure | Nye (1955) |
| 1973 | Crevasse penetration depth of a single crevasse | Weertman (1973) |
| 1976 | Crevasse penetration depth estimation using LEFM | Smith (1976) |
| 1993 | Strain related fracture formation | Vaughan (1993) |
| 1997 | Sea level dependent calving | Motyka (1997) |
| 1998 | Linear Elastic Fracture Mechanics | Van der Veen (1998a, b) |
| 2003 | Damage mechanics for a single crevasse | Pralong et al. (2003) |
| 2005 | Damage mechanics for a single crevasse | Pralong and Funk (2005) |
| 2007 | Crevasse depth | Benn et al. (2007a, b) |
| 2010 | Crevasse depth | Nick et al. (2010) |
| 2010 | Crevasse depth | Otero et al. (2010) |
| 2012 | Damage mechanics applied to a crevasse field | Borstad et al. (2012) |
| 2012 | Kinetic 1st order calving | Levermann et al. (2012) |
| 2012 | CDM | Duddu and Waisman (2012) |
| 2013 | CDM | Duddu and Waisman (2013) |
| 2013 | Discrete element models | Bassis and Jacobs (2013) |
| 2013 | Particle-based simulation | Astrom et al. (2013) |
| 2013 | Crevasse depth criterion | Nick et al. (2013) |
| 2014 | Crevasse depth criterion | Cook et al. (2014) |
| 2014 | CDM | Albrecht and Levermann (2014) |
| 2014 | Combining CDM and LEFM | Krug et al. (2014) |
| 2016 | von Mises tensile stress | |

**Table 2.** Predictor factors (predictors)

| Type | Predictor | Description |
|---|---|---|
| **Geometry** | Ice thickness | Bedmap2 data for Antarctica at 1 km spatial resolution |
| | Maximum bed slope | Bedrock and ice surface slopes are calculated using a nodal function |
| | Maximum surface slope | |
| | Proximity to the ice front | $DM_{IF}$, calculated using Eq. 9 |
| | Proximity to grounding line | $DM_{GL}$, calculated using Eq. 10 |
| | Proximity to glacier edges and nunataks | |
| | Curvature | Curvature of the glacier channel $\alpha$, calculated in each node based on the direction and rate of the flow velocities (see Eq. 11) |
| **Flow parameters** | back stress | Buttressing effect on ice streams calculated in ISSM from inversion |
| | Velocity | InSAR ice flow velocity |
| | Rheology predictor (viscosity) | $B$, Glen's flow predictor, calculated from inversion of velocities (only for floating ice) |
| | Effective Strain rate | The effective strain rate is calculated using Eq. 8 with observed velocities as an input |
| | Principal stress (1 and 2) | Eigenvalues $\lambda$ (normal stresses) in Eq. 4 |
| | Principal strain rate (1 and 2) | Eigenvalues $\mu$ (see Eq. 7) |
| | Strain rate gradient | Maximum strain rate change in a 400-600 metres vicinity |

**Table 3.** Formed groups of ice shelf regions

| | |
|---|---|
| **Group1** | 9, 10, 11, 12, 13, 15, 18, 20, 21, 23, 25, 26, 30, 31, 32, 34, 4, 29 |
| **Group2** | 3, 6, 7, 8, 19, 27, 28, 33, 35 |
| **Group3** | 14, 17, 22, 24 |
| **Group4** | 1, 2, 5, 16 |

Glaciers/ice shelves for which we could not find a good-fitting probability are marked with red.

**Table 4.** Predictors for grounded ice regions in each formed group

| | Effective stress | Effective strain rate | Principal 1 strain rate | Principal 2 strain rate | Principal 1 stress | Principal 2 stress | Surface slope | Bed slope | Strain change | Curvature | Rheology B | Thickness | at the ice front | at the grounding line | near edges | surface change | Velocity |
|---|---|---|---|---|---|---|---|---|---|---|---|---|---|---|---|---|---|
| Group1: | | ✓ | ✓ | | | | ✓ | | | ✓ | ✓ | | ✓ | | ✓ | ✓ | ✓ |
| Group2: | ✓ | | | | ✓ | | | | ✓ | ✓ | ✓ | | | | | ✓ | ✓ |
| Group3: | | | | ✓ | | | | | | | ✓ | | | | ✓ | | ✓ |
| Group4: | | ✓ | | | | | | ✓ | ✓ | | | | | | | ✓ | ✓ |

Tick-mark stands for an addition of a predictor to the model for grounded ice.

**Table 5.** Predictors for floating ice in each formed group

| | Effective stress | Back stress | Effective strain rate | Principal 1 strain rate | Principal 2 strain rate | Principal 1 stress | Principal 2 stress | Surface slope | Strain rate change | Curvature | Rheology B | Thickness | at the ice front | at the grounding line | surface change | Velocity |
|---|---|---|---|---|---|---|---|---|---|---|---|---|---|---|---|---|
| Group1: | ✓ | | | | ✓ | ✓ | | ✓ | ✓ | | | | | | | ✓ |
| Group2: | | | ✓ | | | ✓ | | | | | ✓ | ✓ | | ✓ | ✓ | |
| Group3: | | | ✓ | | | | ✓ | | | | ✓ | ✓ | | | ✓ | |
| Group4: | | ✓ | ✓ | ✓ | | | | ✓ | ✓ | | | | | ✓ | ✓ | ✓ |

Tick-mark stands for an addition of a predictor to the model for floating ice.

**Table 6.** A list of analysed ice shelf regions

| Glacier | Group | Corresponding IS name | Region |
|---|---|---|---|
| 1 | 4 | George IV | Palmer land, AP |
| 2 | 4 | Larsen C | Fallieres Coast, AP |
| 3 | 2 | Larsen D | Black Coast, AP |
| 4 | 1 | Orville Coast side of the Ronne IS | WA |
| 5 | 4 | Amery | EA |
| 6 | 2 | Edward VII | Mawson Coast, EA |
| 7 | 2 | Rayner Thyner | EA |
| 8 | 2 | Shirase | Prince Harald Coast, EA |
| 9 | 1 | Stancomb-Brunt | Caird Coast, EA |
| 10 | 2 | Riiser-Larsen | Princess Martha Coast, EA |
| 11 | 3 | Fimbul IS | EA |
| 12 | 1 | Abbot | Eights Coast, WA |
| 13 | 2 | Baudoin | Princess Ragnhild Coast, EA |
| 14 | 3 | Nivl | Princess Astrid Coast, EA |
| 15 | 1 | Borchgrevnik and Lazarev | Princess Astrid Coast, EA |
| 16 | 4 | Borchgrevnik | Princess Raghild Coast, EA |
| 17 | 3 | Dibble IS | Clarie Coast, EA |
| 18 | 1 | Mertz IS | EA |
| 19 | 2 | Rennik | Pennell Coast, EA |
| 20 | 1 | Cook | George V Coast, EA |
| 21 | 1 | Ninnis | George V Coast, EA |
| 22 | 3 | Holmes | Banzare Coast, EA |
| 23 | 1 | Moscow University | Sabrina Coast, EA |
| 24 | 3 | Totten IS | EA |
| 25 | 2 | Vanderford IS | EA |
| 26 | 1 | West IS | Queen Mary Coast, EA |
| 27 | 2 | Larsen C | Oscar II Coast, AP |
| 28 | 2 | Larsen B | Nordenskjold Coast, AP |
| 29 | 2 | Larsen A | Davis Coast, AP |
| 30 | 3 | Tracy-Tremenchus | Knox Coast, EA |
| 31 | 1 | Drygalski | Scott Coast, EA |
| 32 | 2 | Mariner | Borchgrevnik Coast EA |
| 33 | 3 | Rennik | Lazarev Mountains, Oates Coast, EA |

AP - Antarctic Peninsula, EA - East Antarctica, WA - West Antarctica, IS -ice shelf     

**Table 7.** A list of analysed ice shelf regions

| Glacier | Group | Corresponding IS name | Region |
|---|---|---|---|
| 34 | 1 | Filchner | Coast Land, WA |
| 35 | 2 | Ross East | Hut Point Peninsula, EA |
| 36 | 1 | Wilkins and George VI | Rumill Coast, AP |
| 37 | 1 | Stange and Ferringo IS | Bryan Coast, AP |
| 38 | 1 | Pine Island and Thwaites | Walgreen Coast, WA |
| 39 | 2 | Getz | Hobbs and Bakutis Coast, WA |
| 40 | 1 | Nickerson and Sulzberger | Ruppers Coast, WA |
| 41 | 1 | West | Leopold and Astrid Coast, EA |
| 42 | 1 | Jelbart and Atka | Princess Martha Coast, WA |
| 43 | 1 | Nansen | Borchgrevnik Coast, EA |
| 44 | 1 | Prince Harald | Prince Harald Coast, EA |
| 45 | 1 | Larsen B | Oscar II Coast, AP |

AP - Antarctic Peninsula, EA - East Antarctica, WA - West Antarctica, IS -ice shelf

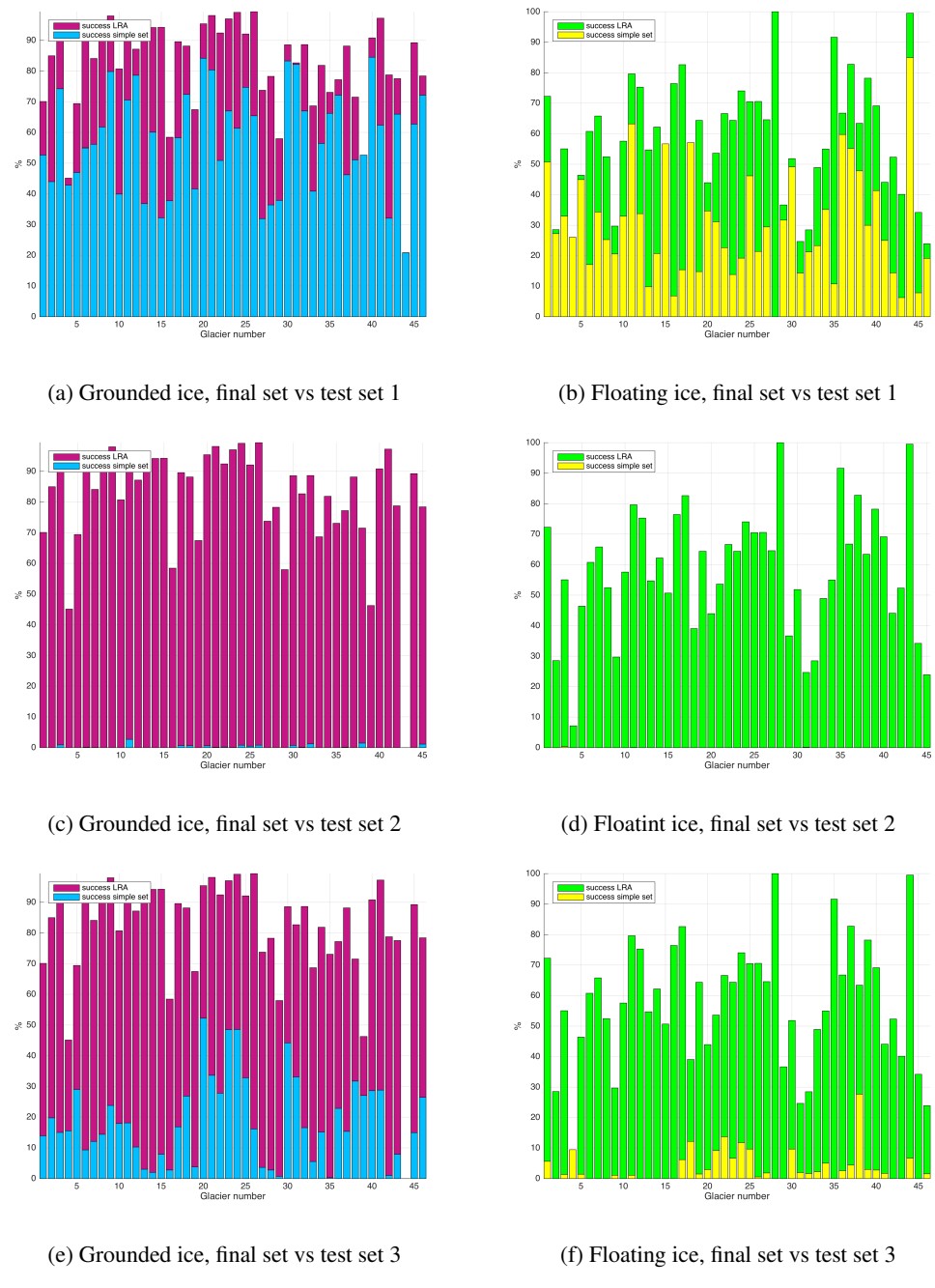

(a) Grounded ice, final set vs test set 1

(b) Floating ice, final set vs test set 1

(c) Grounded ice, final set vs test set 2

(d) Floatint ice, final set vs test set 2

(e) Grounded ice, final set vs test set 3

(f) Floating ice, final set vs test set 3

**Figure 1.** Comparison between the success of identifying fractures using LRA (purple for grounded ice, green for floating ice) and using Test set 1: effective deviatoric stress, Test set 2: principal deviatoric stress 1 and 2, Test set 3: von Mises stress (blue for grounded ice, yellow for floating ice). Left column represents grounded ice (a,c,e) and right show the results for floating ice (b,d,f).

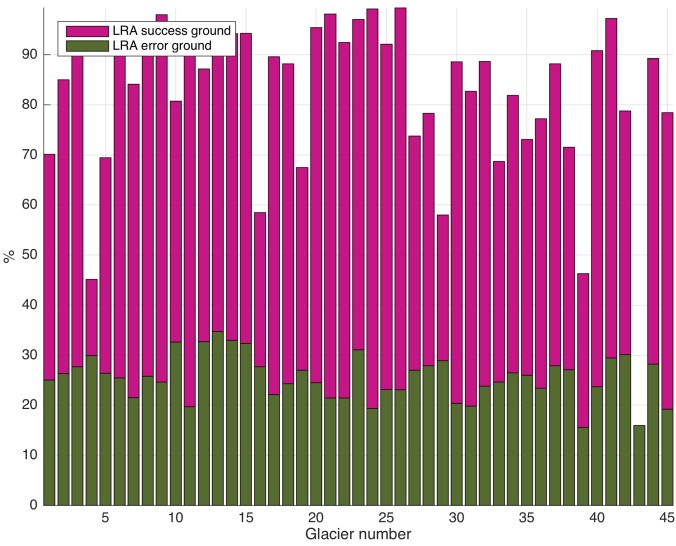

(a) Success and Error

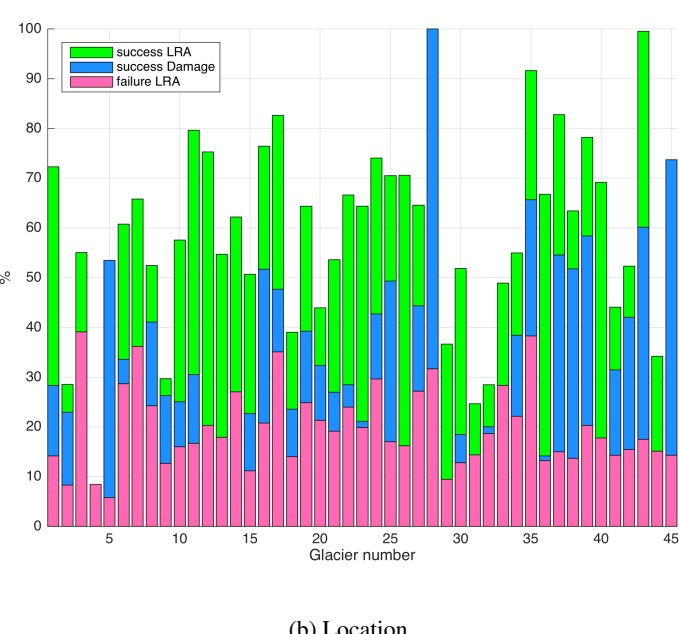

(b) Location

**Figure 2.** Success and error percentages for LRA for grounded ice is shown in panel a. Results for the floating ice applying LRA vs. Damage method is shown in panel b.

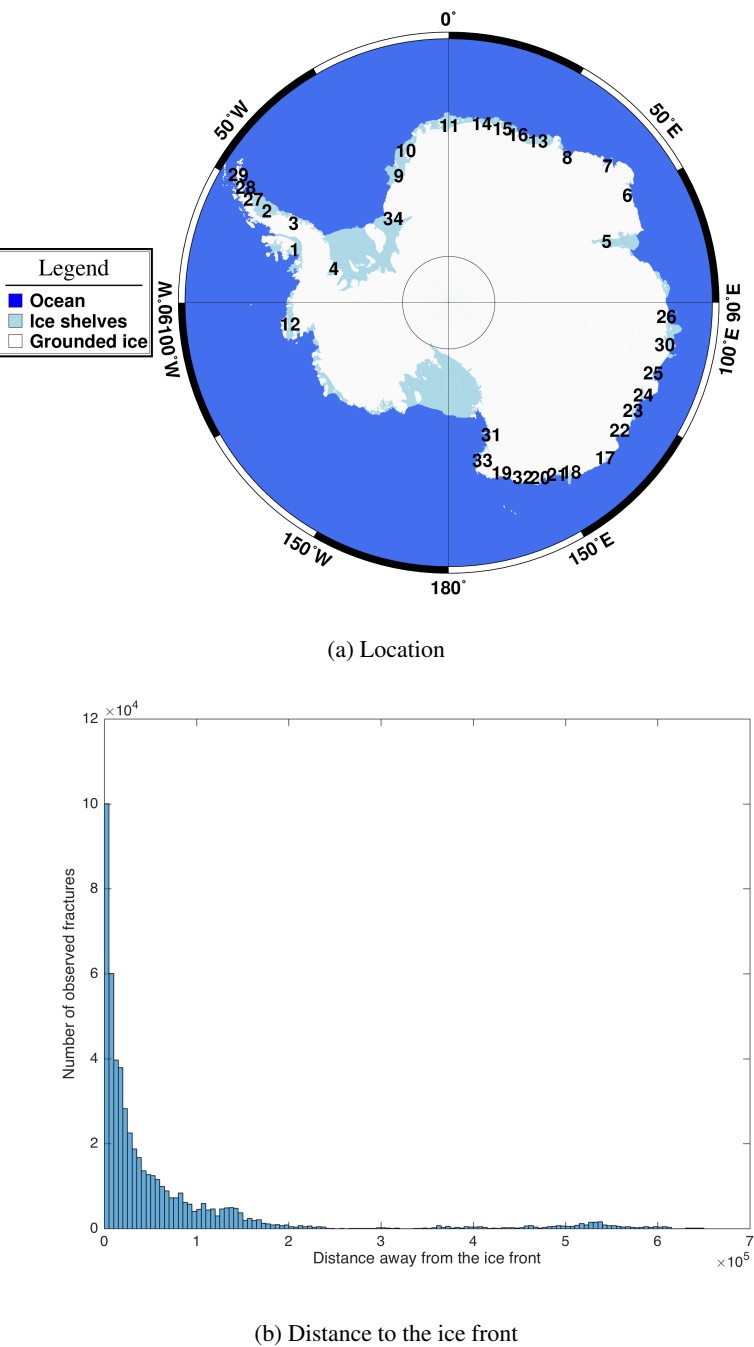

(a) Location

(b) Distance to the ice front

**Figure 3.** The location of each of the 45 ice shelf regions is shown in panel a. The number of observed fractures versus distance from the ice front (b).

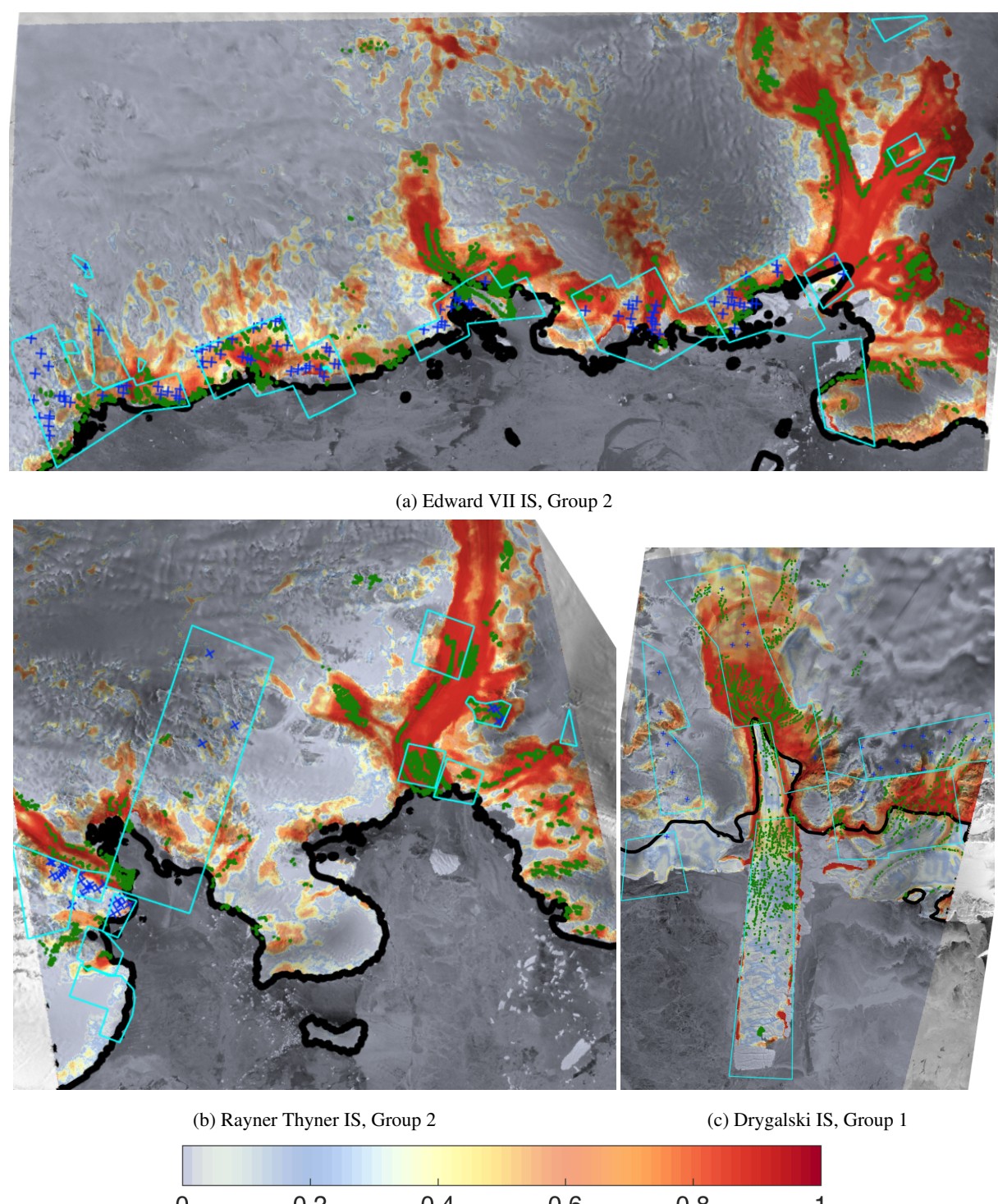

(a) Edward VII IS, Group 2

(b) Rayner Thyner IS, Group 2        (c) Drygalski IS, Group 1

0   0.2   0.4   0.6   0.8   1
<-- Less likely to fracture / More likely to fracture -->

**Figure 4.** Modelled probability of a fracture for Group 2: Rayner Thyner IS (a) and Edward VII IS (b) and Group 1: Drygalski IS (c) . Observed surface fractures are shown in green and observed non-fractured ice is marked with blue crosses. Cyan boxes represent regions where high resolution images were available. Black solid line shows the location of the grounding line.

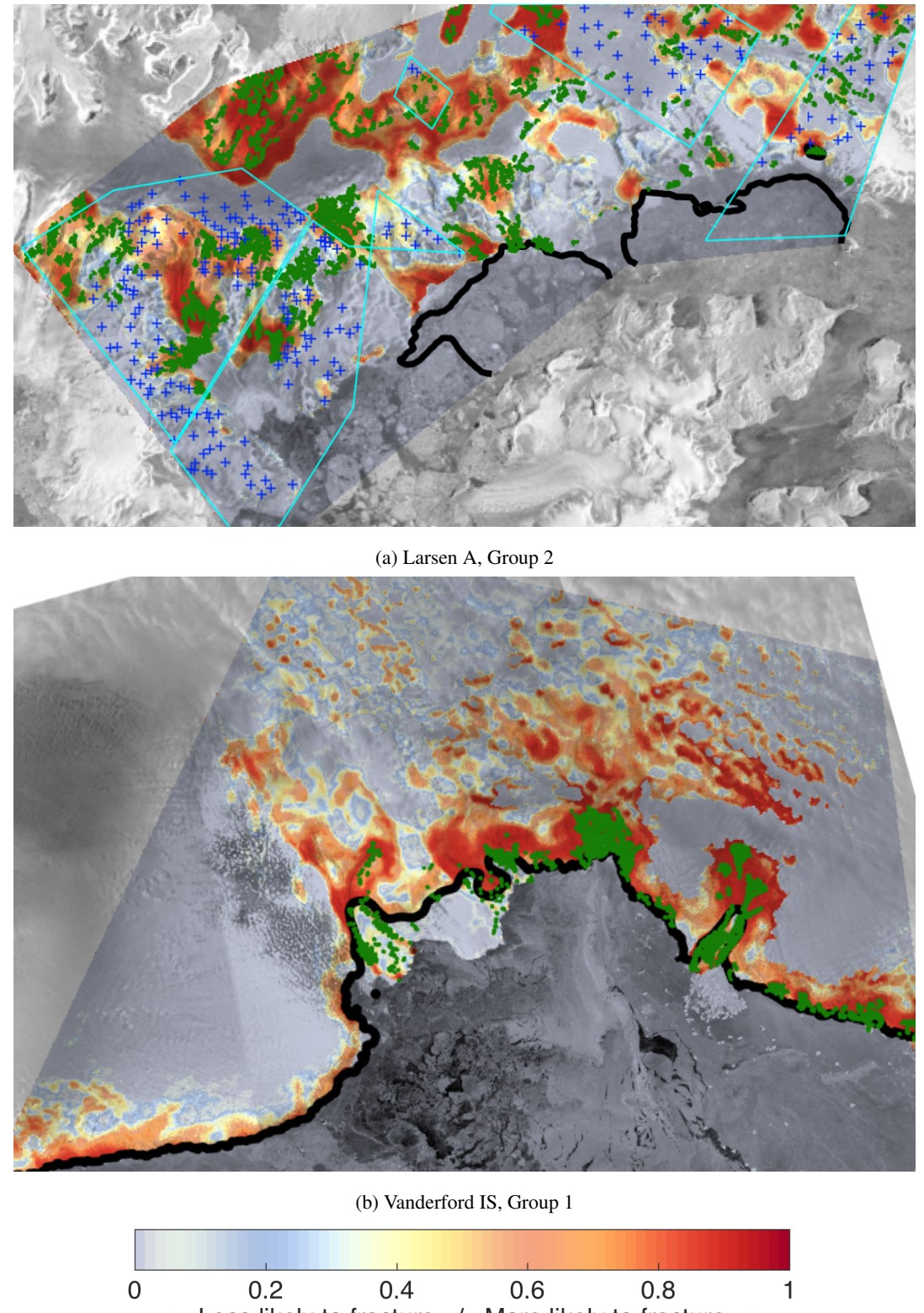

(a) Larsen A, Group 2

(b) Vanderford IS, Group 1

0   0.2   0.4   0.6   0.8   1
<-- Less likely to fracture / More likely to fracture -->

**Figure 5.** Modelled probability of a fracture for Group 2: Larsen A IS (a) and Group1: Vanderford IS (b). Labels are the same as Figure 4.

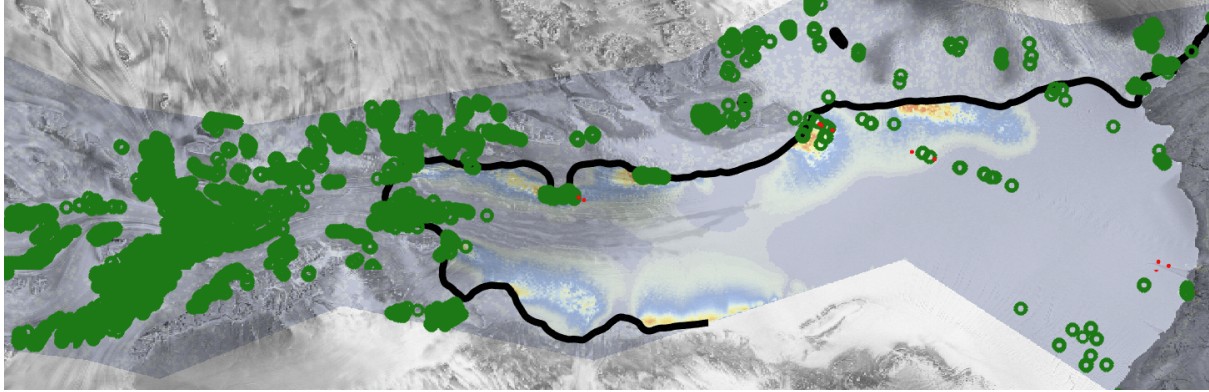

(a) LRA, Pine Island and Thwaites, Group 1

(b) LRA, Amery IS, Group 4

(c) Damage, Amery IS, Group 4

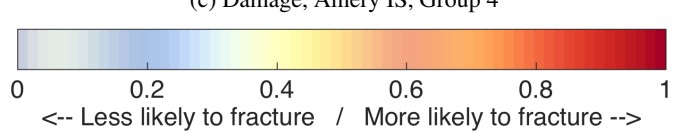

0          0.2          0.4          0.6          0.8          1
<-- Less likely to fracture   /   More likely to fracture -->

**Figure 6.** Modelled probability for Pine Island (Group 1) (a), modelled probability for Amery IS (Group 4) (b) and inferred damage for Amery (c). Labels the same as Figure 4. Red dots represent observed fractures that were considered to have advected downstream from areas identified as fracture formation zones by the damage model and, thus, were not used for evaluation of the damage-based results.

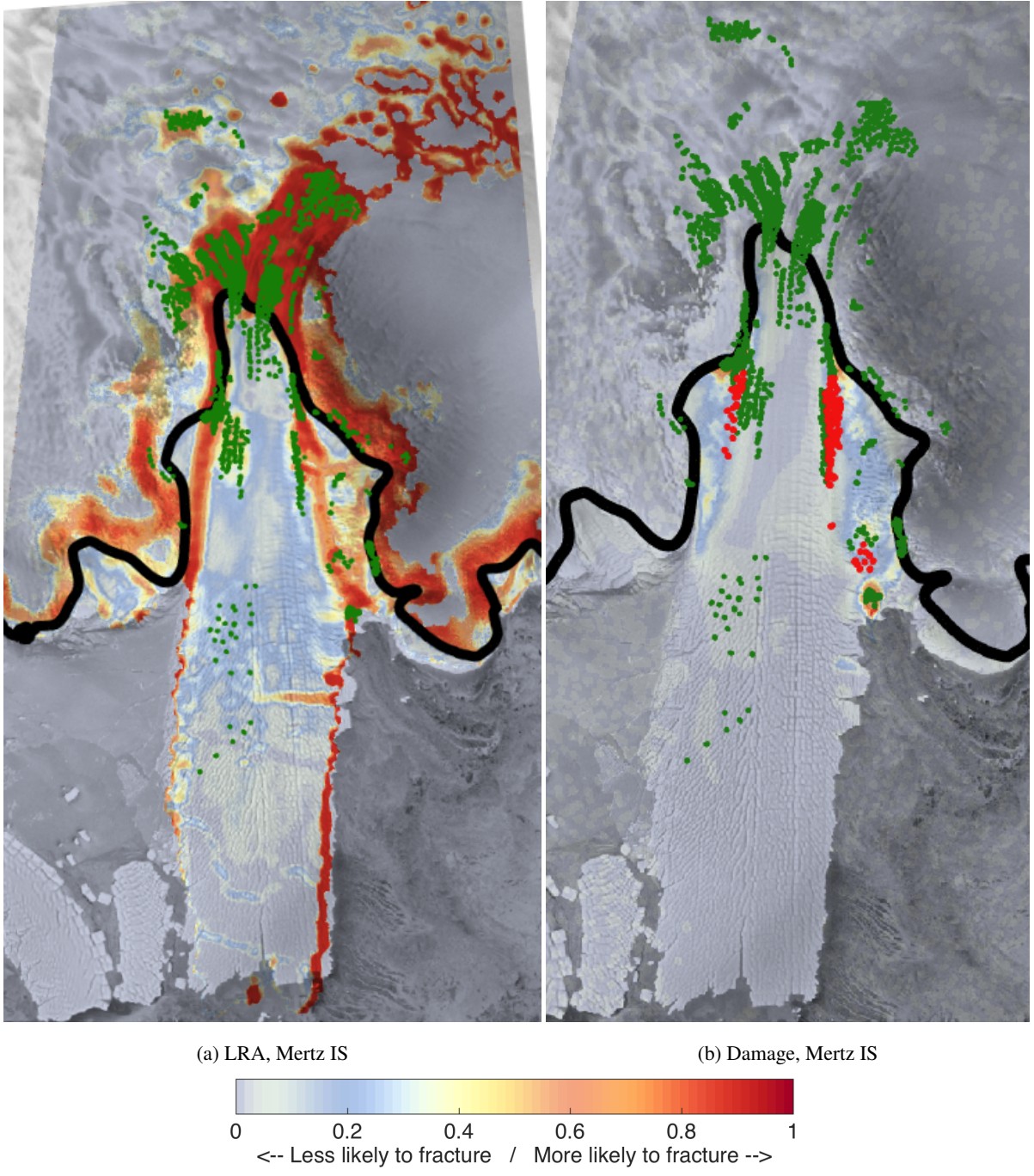

(a) LRA, Mertz IS                    (b) Damage, Mertz IS

0        0.2        0.4        0.6        0.8        1
<-- Less likely to fracture  /  More likely to fracture -->

**Figure 7.** Modelled probability (Figure a) and damage (Figure b) at Mertz IS (Group 1). Labels the same as Figure 4. Red dots represent observed fractures that were considered to have advected downstream from areas identified as fracture formation zones by the damage model and, thus, were not used for evaluation of the damage-based results.

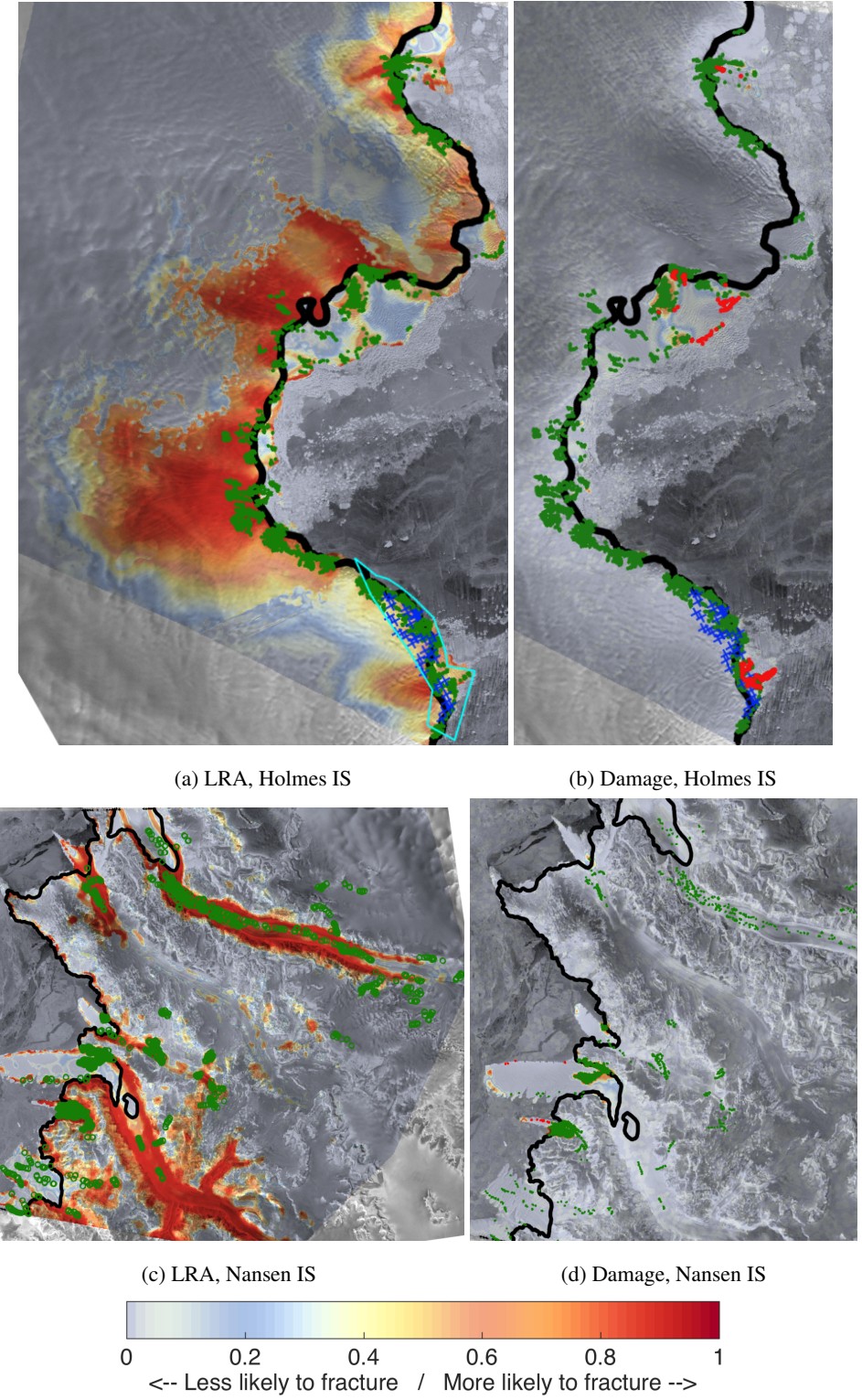

(a) LRA, Holmes IS

(b) Damage, Holmes IS

(c) LRA, Nansen IS

(d) Damage, Nansen IS

0    0.2    0.4    0.6    0.8    1

<-- Less likely to fracture   /   More likely to fracture -->

**Figure 8.** Modelled probability of a fracture vs. modelled damage for Holmes IS (Group 3) (a, b) and Nansen IS (Group 1) (c, d). Labels are the same as in Figure 4 and 7.

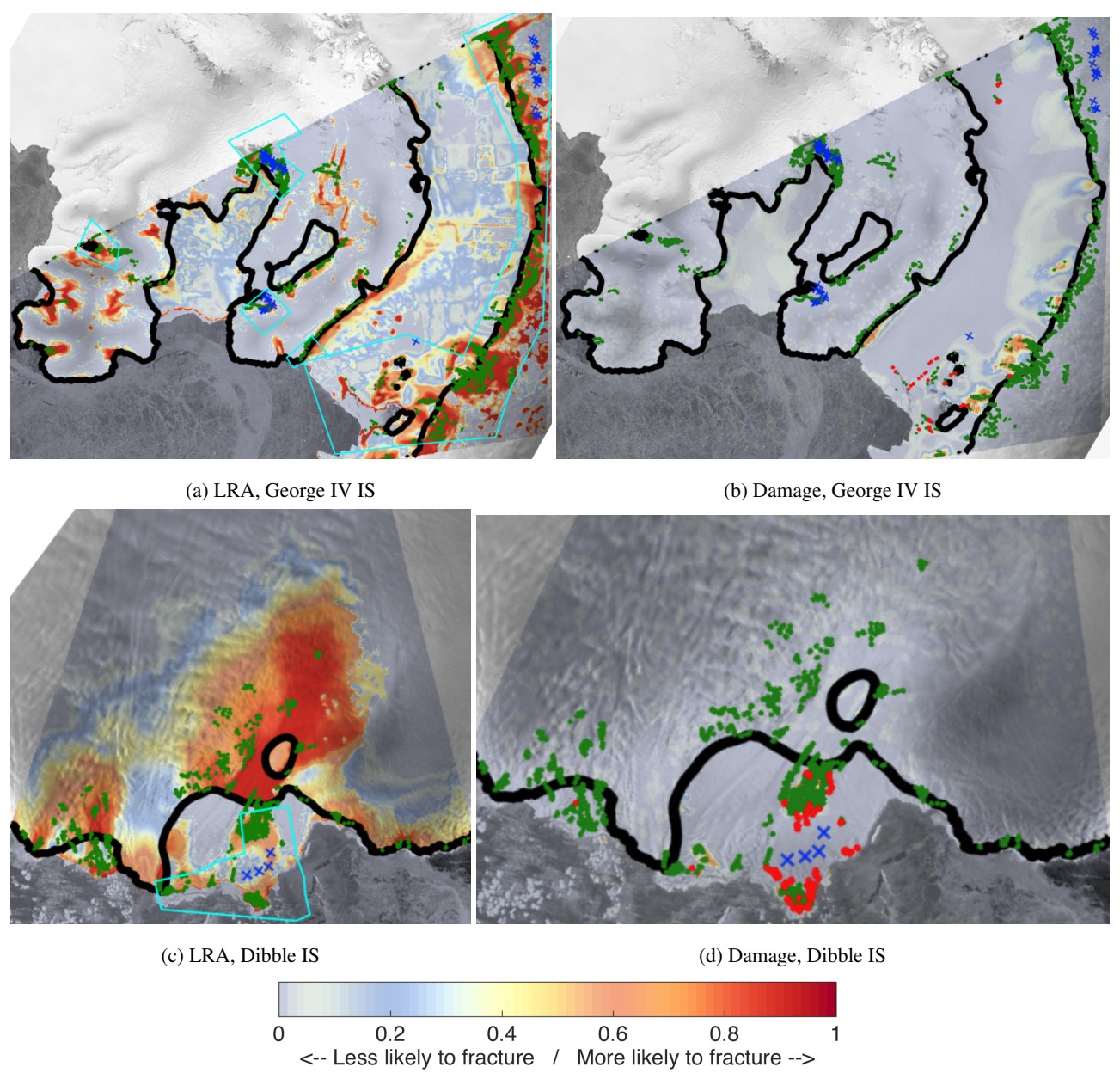

(a) LRA, George IV IS

(b) Damage, George IV IS

(c) LRA, Dibble IS

(d) Damage, Dibble IS

0    0.2    0.4    0.6    0.8    1
<-- Less likely to fracture  /  More likely to fracture -->

**Figure 9.** Modelled probability of a fracture vs. modelled damage for George IV IS (Group 4) (a, b) and Dibble IS (Group 3) (c, d). Labels are the same as in Figure 4 and 7.

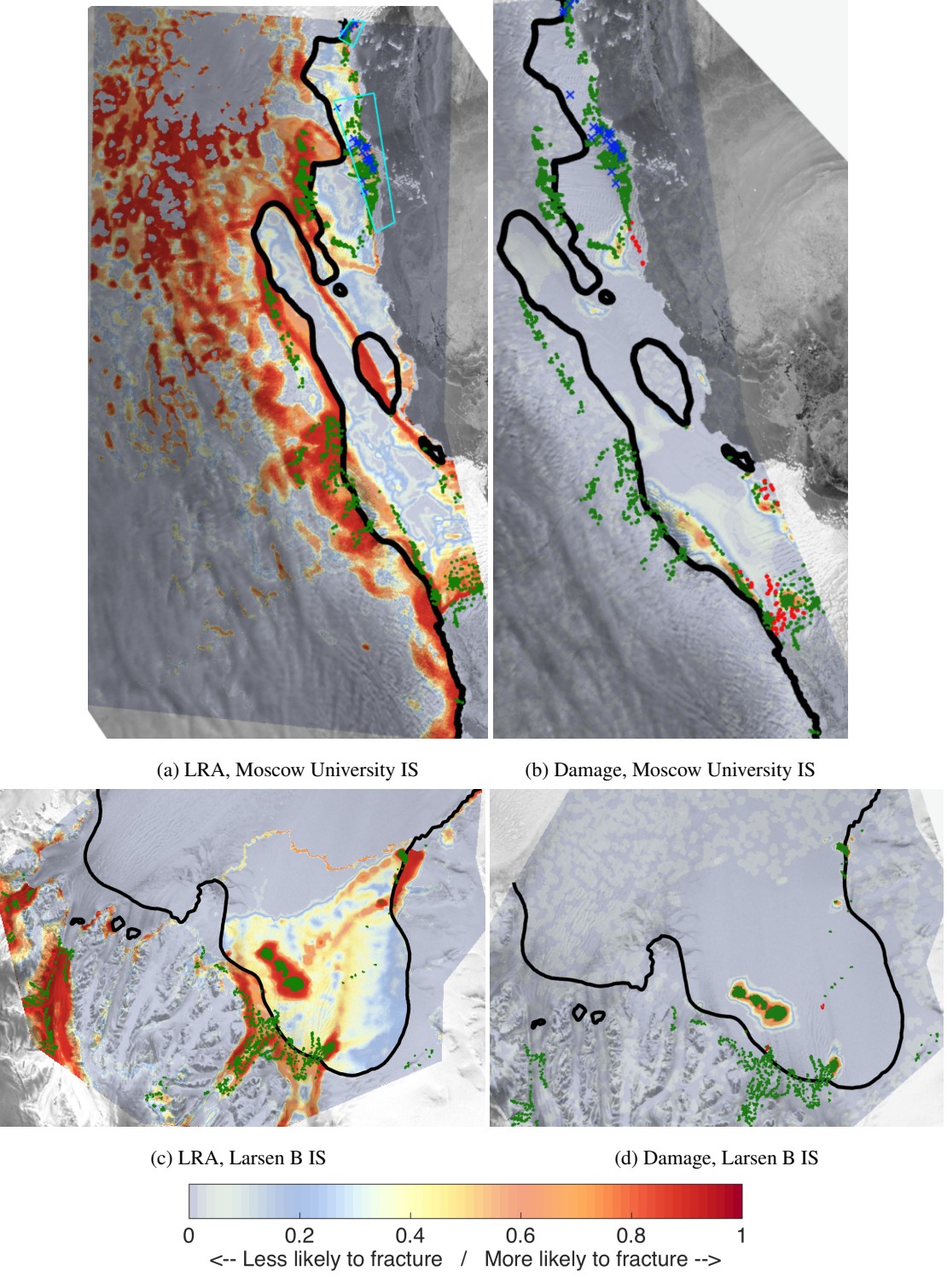

(a) LRA, Moscow University IS

(b) Damage, Moscow University IS

(c) LRA, Larsen B IS

(d) Damage, Larsen B IS

0    0.2    0.4    0.6    0.8    1
<-- Less likely to fracture  /  More likely to fracture -->

**Figure 10.** Modelled probability of a fracture vs. modelled damage for Moscow University IS (Group 1) (a, b) and Larsen B IS (Group 1) (c, d). Labels are the same as in Figure 4 and 7

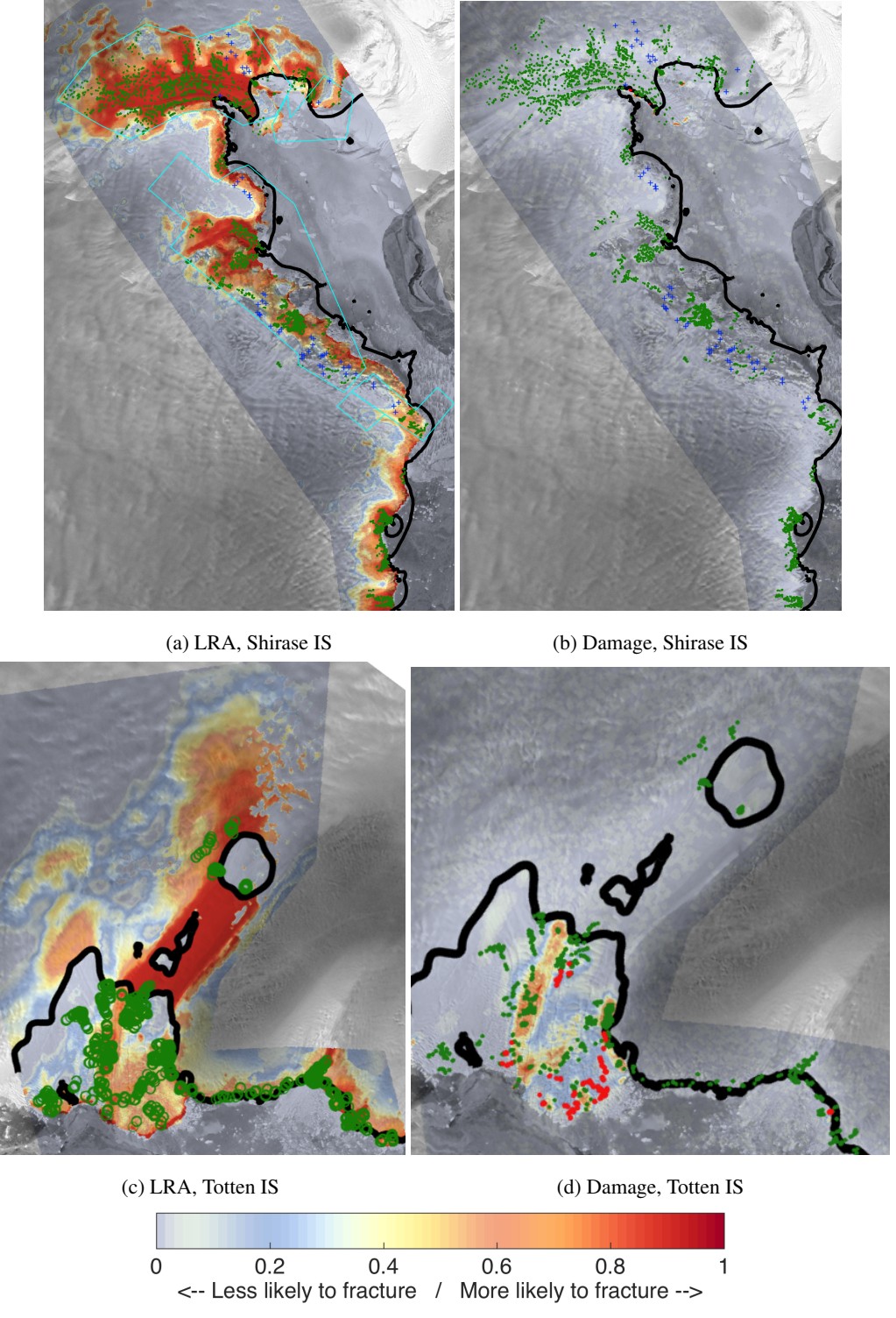

(a) LRA, Shirase IS

(b) Damage, Shirase IS

(c) LRA, Totten IS

(d) Damage, Totten IS

0    0.2    0.4    0.6    0.8    1
<-- Less likely to fracture  /  More likely to fracture -->

**Figure 11.** Modelled probability of a fracture vs. modelled damage for Shirase IS (Group 2) (a, b) and Totten IS (Group 3) (c, d). Labels are the same as in Figure 4 and 7 .

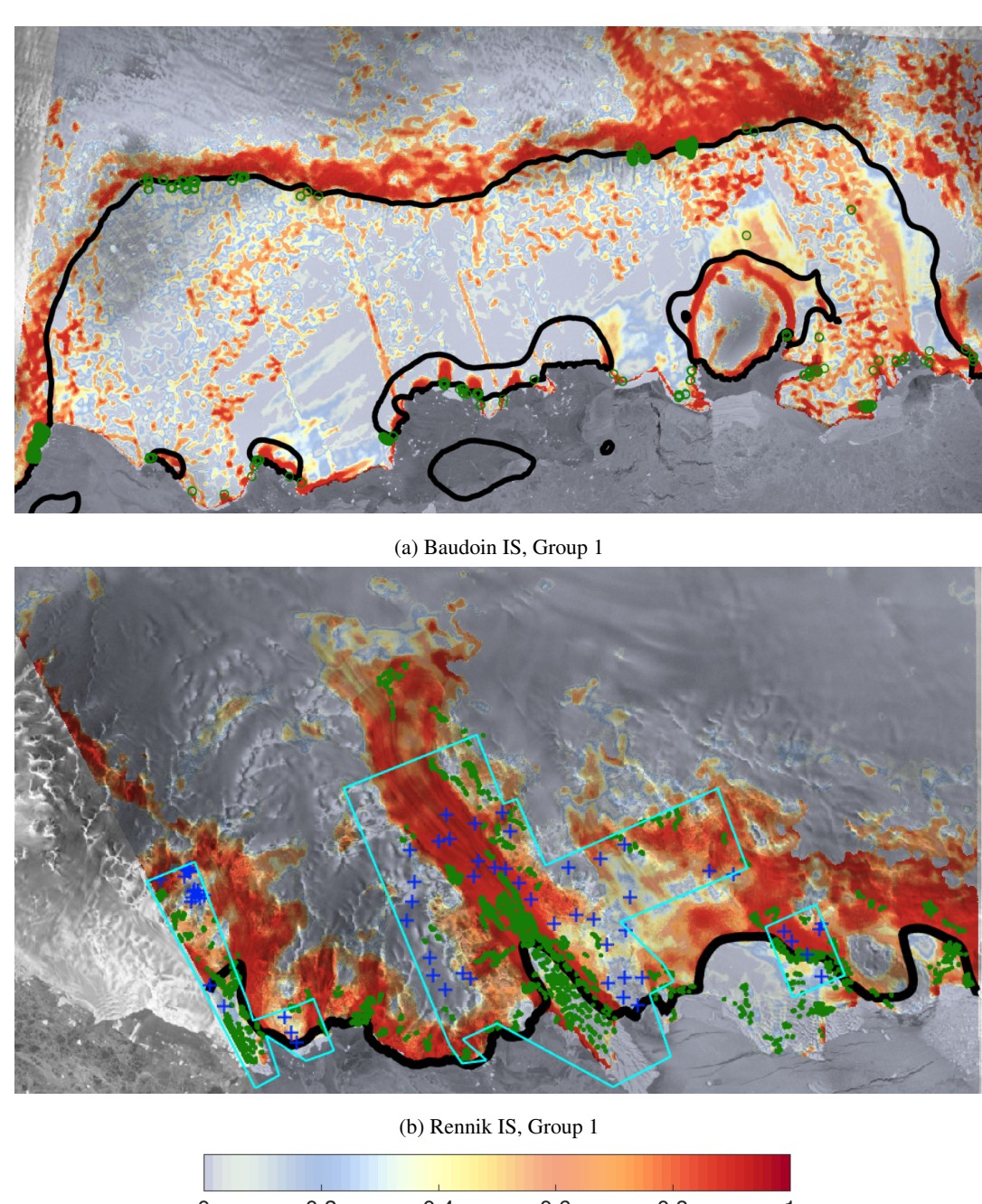

(a) Baudoin IS, Group 1

(b) Rennik IS, Group 1

0    0.2    0.4    0.6    0.8    1
<-- Less likely to fracture   /   More likely to fracture -->

**Figure S1.** Group 1: Modelled probability of a fracture for Baudoin IS (a) and Rennik IS (b). Observed surface fractures are shown in black and observed non-fractured ice is marked with orange circles. Red polygons represent regions where high resolution images were available.

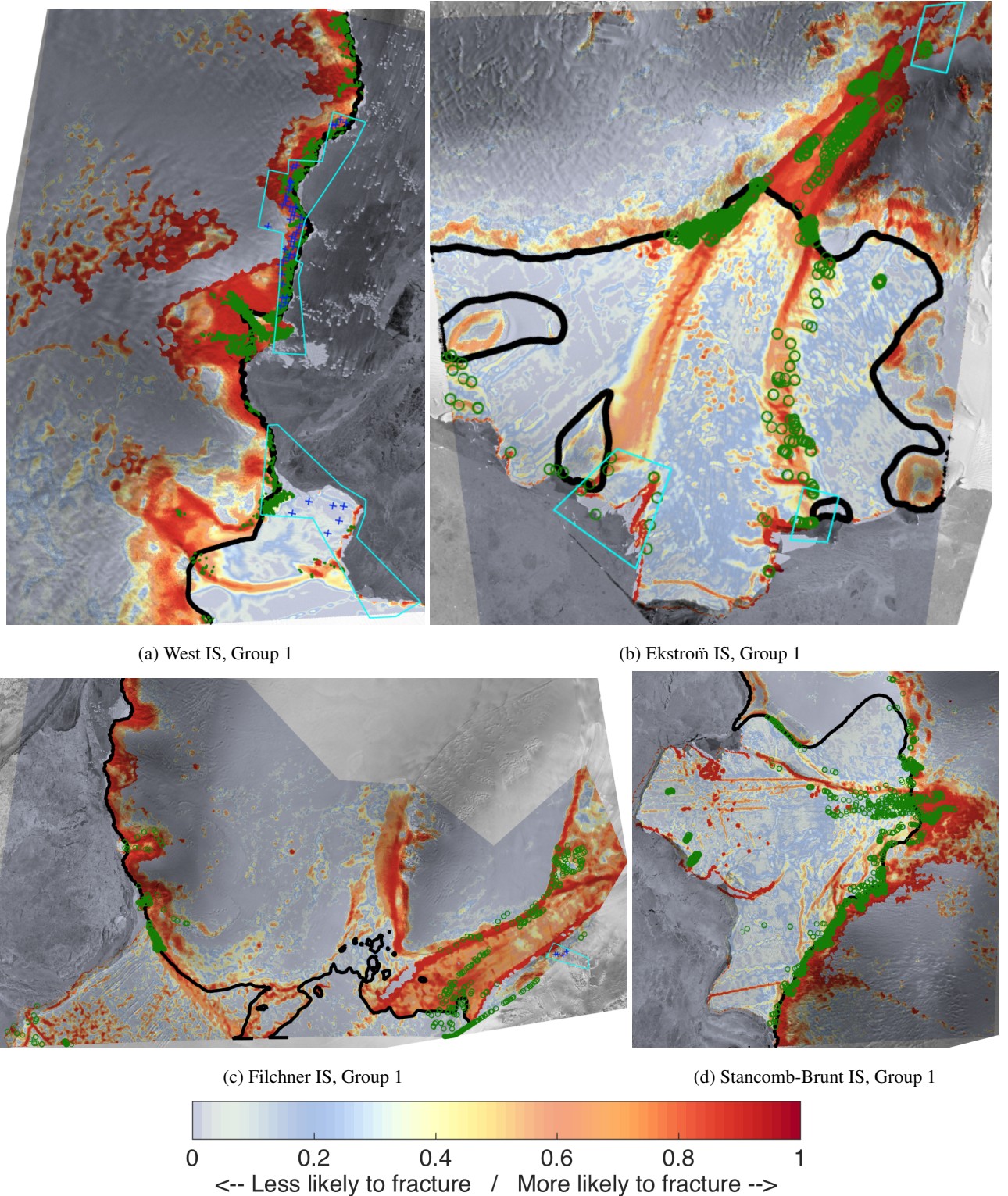

(a) West IS, Group 1

(b) Ekström IS, Group 1

(c) Filchner IS, Group 1

(d) Stancomb-Brunt IS, Group 1

0      0.2      0.4      0.6      0.8      1

<-- Less likely to fracture / More likely to fracture -->

**Figure S2.** Group 1: Modelled probability of a fracture for West IS (a), Ekström IS (b), Filchner IS (c) and Stancomb-Brunt IS (d). Labels the same as Figure S1.

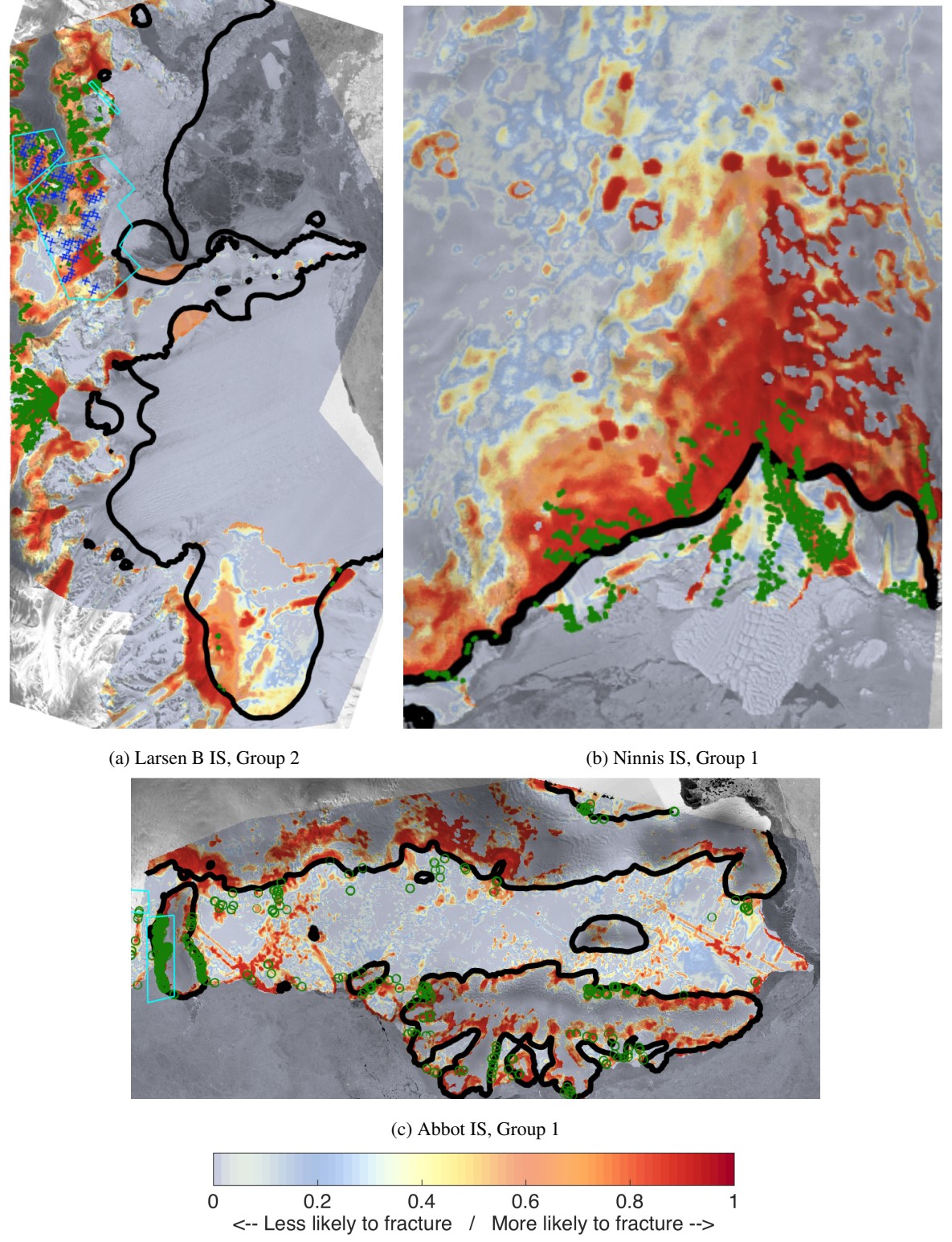

(a) Larsen B IS, Group 2

(b) Ninnis IS, Group 1

(c) Abbot IS, Group 1

0    0.2    0.4    0.6    0.8    1

<-- Less likely to fracture  /  More likely to fracture -->

**Figure S3.** Group 1 and 2: Modelled probability of a fracture for Larsen B IS (Nordenskjold coast) (a), Ninnis IS (b) and Abbot IS (c). Labels the same as Figure S1.

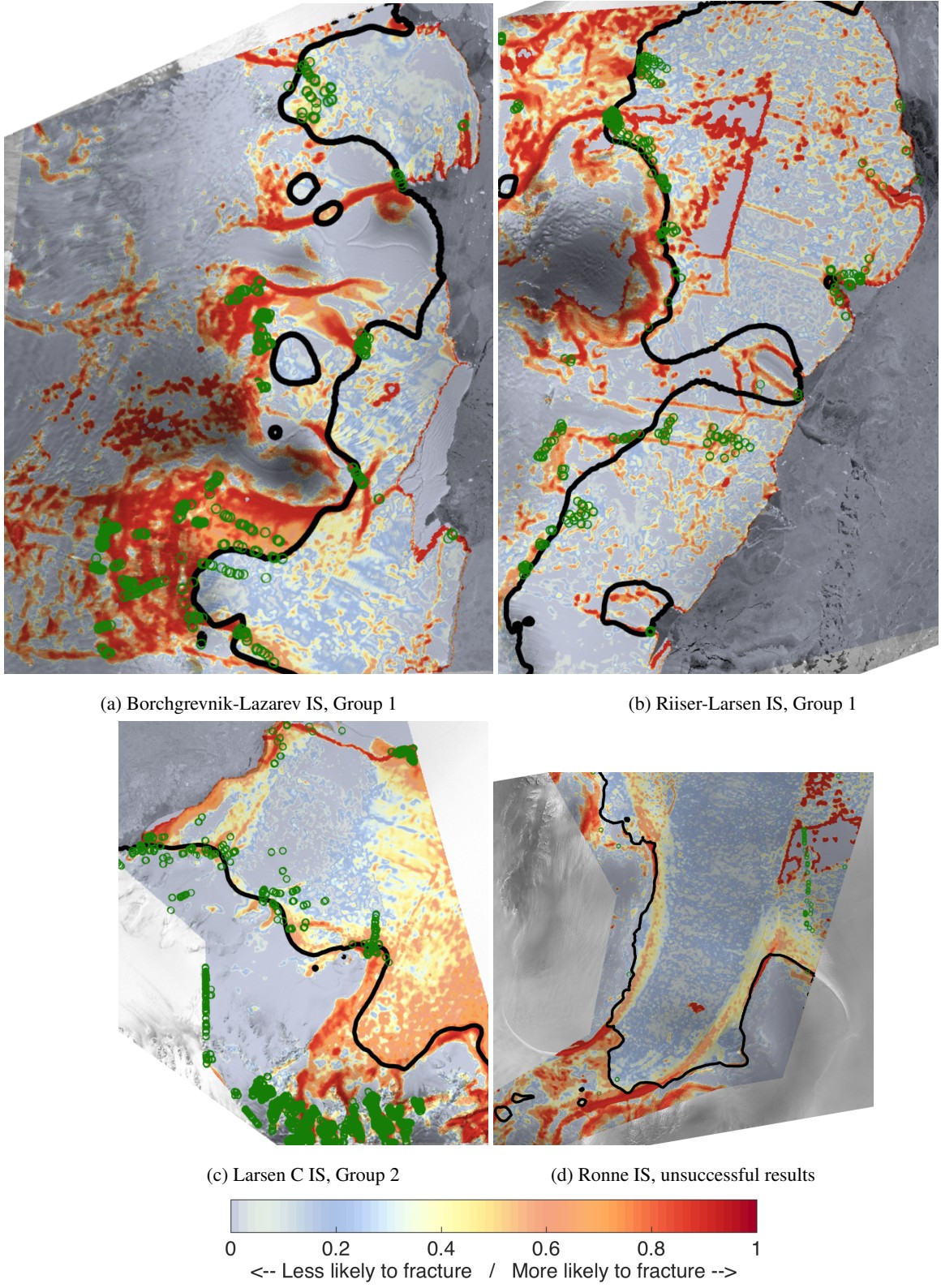

(a) Borchgrevnik-Lazarev IS, Group 1

(b) Riiser-Larsen IS, Group 1

(c) Larsen C IS, Group 2

(d) Ronne IS, unsuccessful results

0     0.2     0.4     0.6     0.8     1

<-- Less likely to fracture  /  More likely to fracture -->

**Figure S4.** Group 1 and 2: Modelled probability of a fracture for Borchgrevnik-Lazarev IS (a), Riiser-Larsen IS (b), Larsen C IS (c), Ronne IS, unsuccessful results (d). Labels the same as Figure S1.

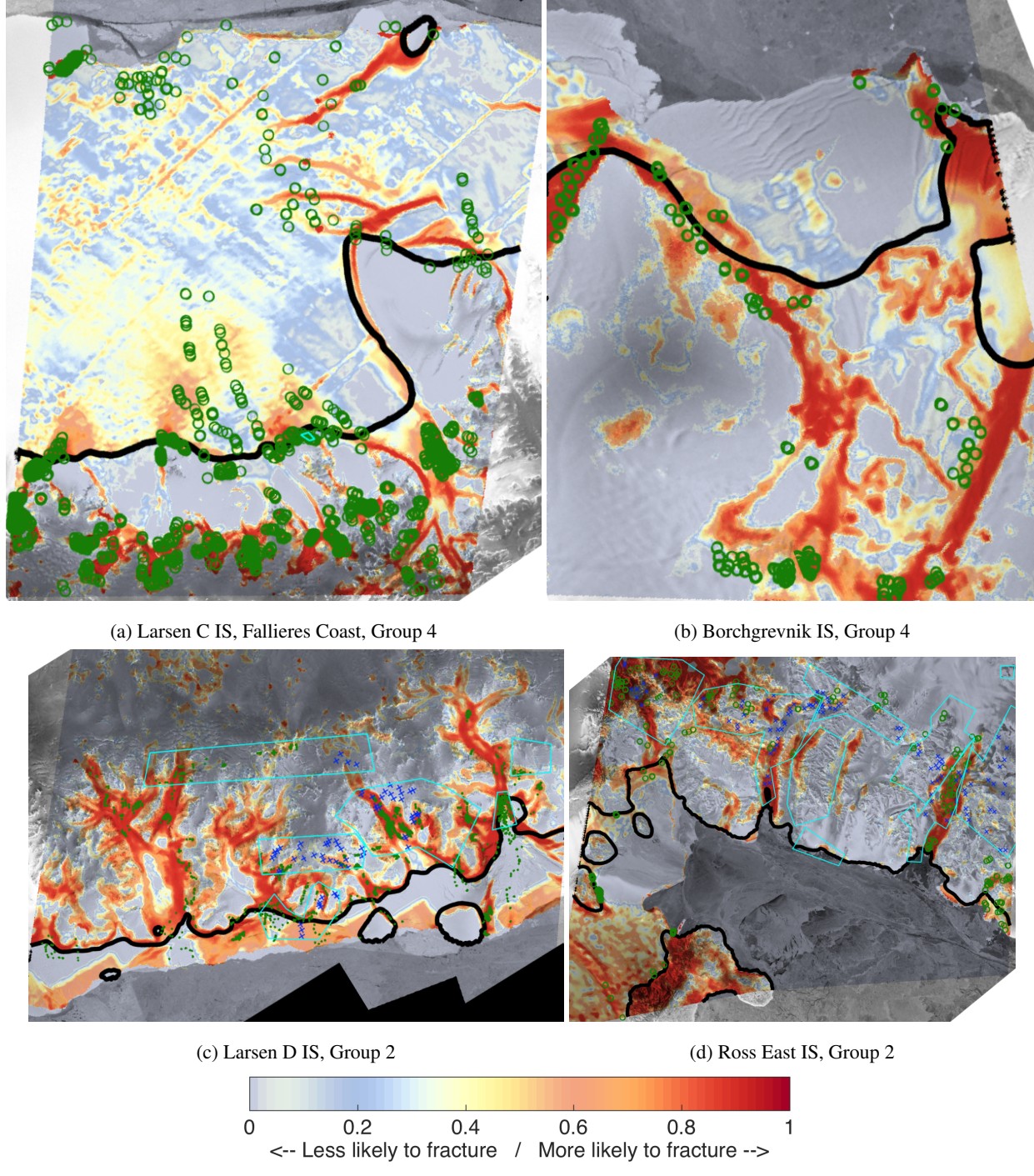

(a) Larsen C IS, Fallieres Coast, Group 4

(b) Borchgrevnik IS, Group 4

(c) Larsen D IS, Group 2

(d) Ross East IS, Group 2

| 0 | 0.2 | 0.4 | 0.6 | 0.8 | 1 |

<-- Less likely to fracture  /  More likely to fracture -->

**Figure S5.** Group 2 and 4: Modelled probability of a fracture for Larsen C IS (a), Borchgrevnik IS (b), Larsen D IS (c) and Ross East IS (d). Labels the same as Figure S1.

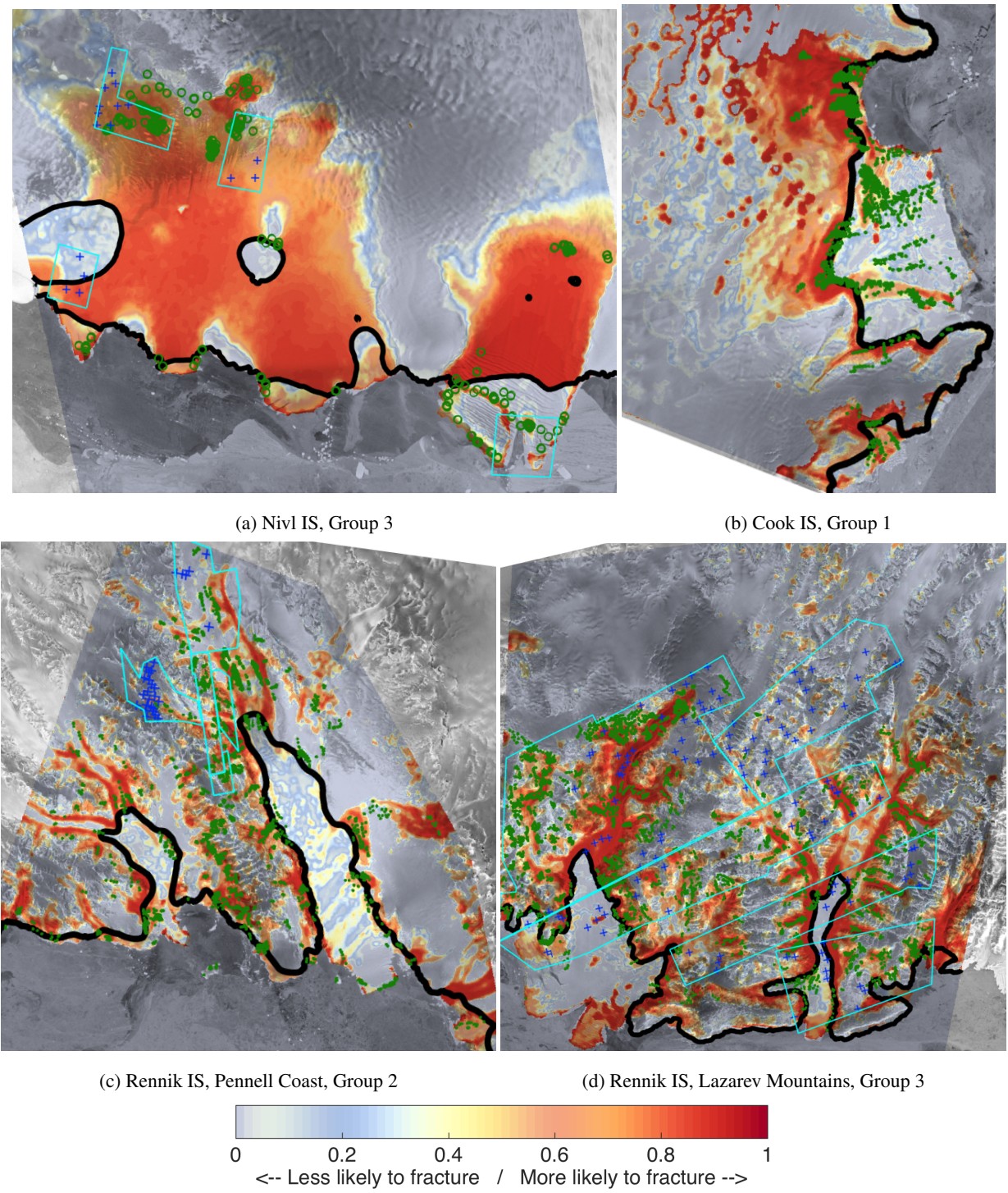

(a) Nivl IS, Group 3

(b) Cook IS, Group 1

(c) Rennik IS, Pennell Coast, Group 2

(d) Rennik IS, Lazarev Mountains, Group 3

0    0.2    0.4    0.6    0.8    1
<-- Less likely to fracture  /  More likely to fracture -->

**Figure S6.** Group 1, 2 and 3: Modelled probability of a fracture for Nivl IS (a), Cook IS (b), Rennik IS (c and d). Labels the same as Figure S1.

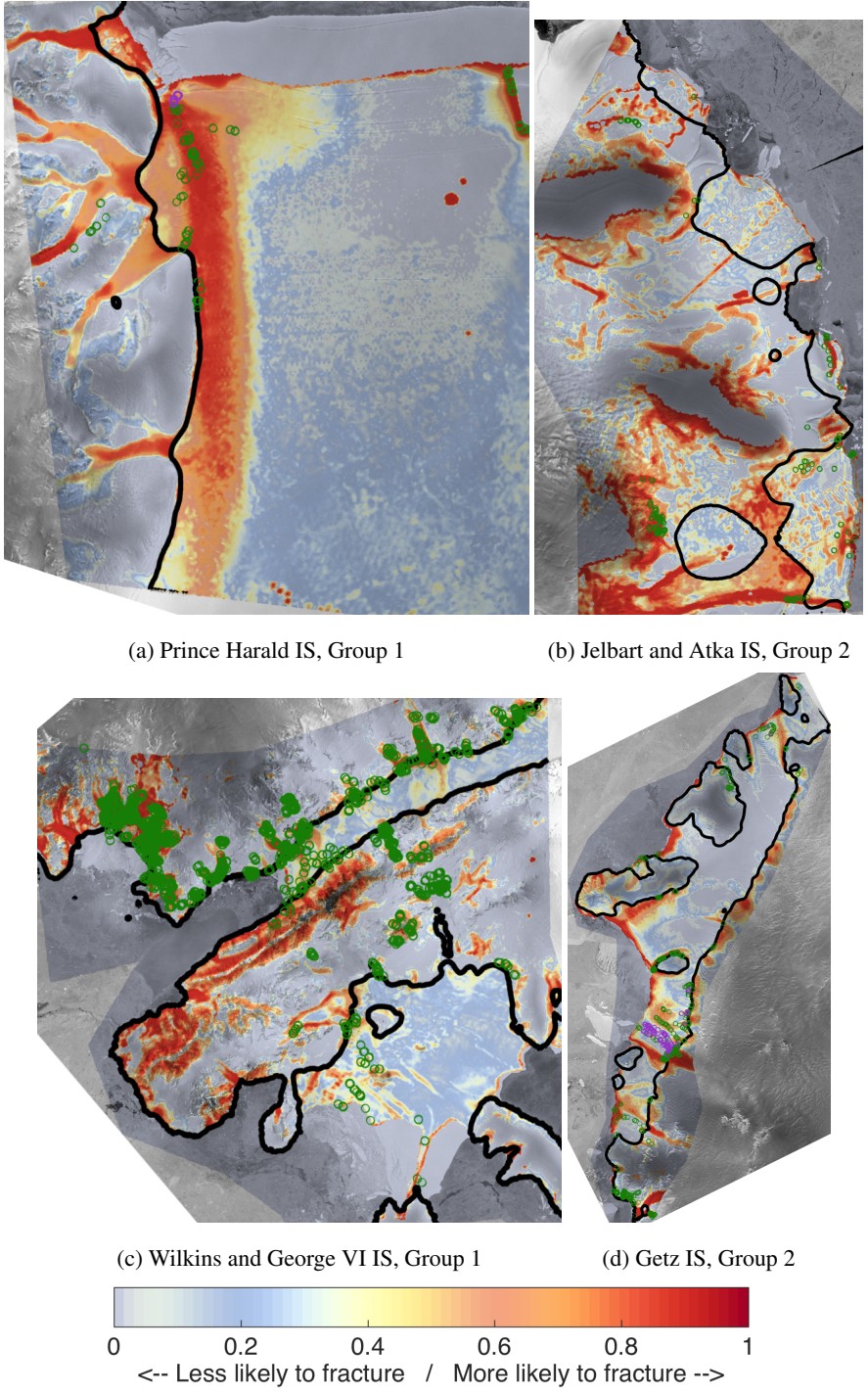

(a) Prince Harald IS, Group 1        (b) Jelbart and Atka IS, Group 2

(c) Wilkins and George VI IS, Group 1        (d) Getz IS, Group 2

0     0.2     0.4     0.6     0.8     1

<-- Less likely to fracture / More likely to fracture -->

**Figure S7.** Group 1 and 2: Modelled probability of a fracture for Prince Harald IS (a), Jelbart and Atka IS (b), Wilkins and George VI IS (c) and Getz IS (d). Labels the same as Figure S1.

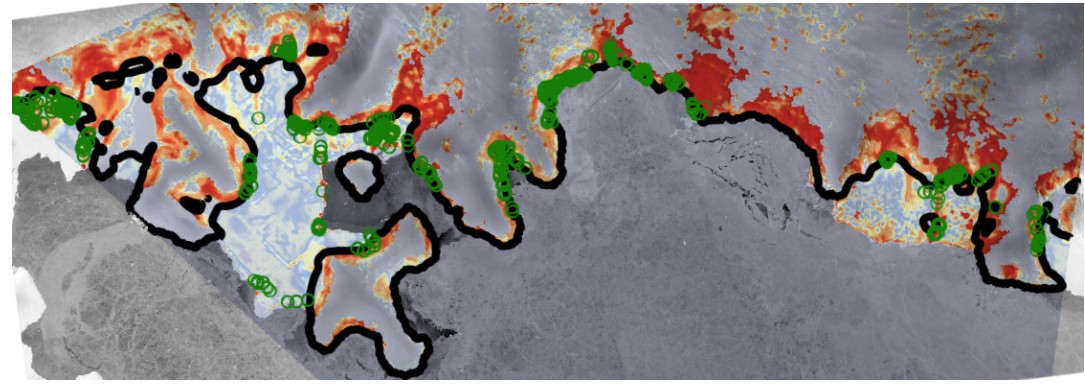

(a) Stange and Ferringo IS, Group 1

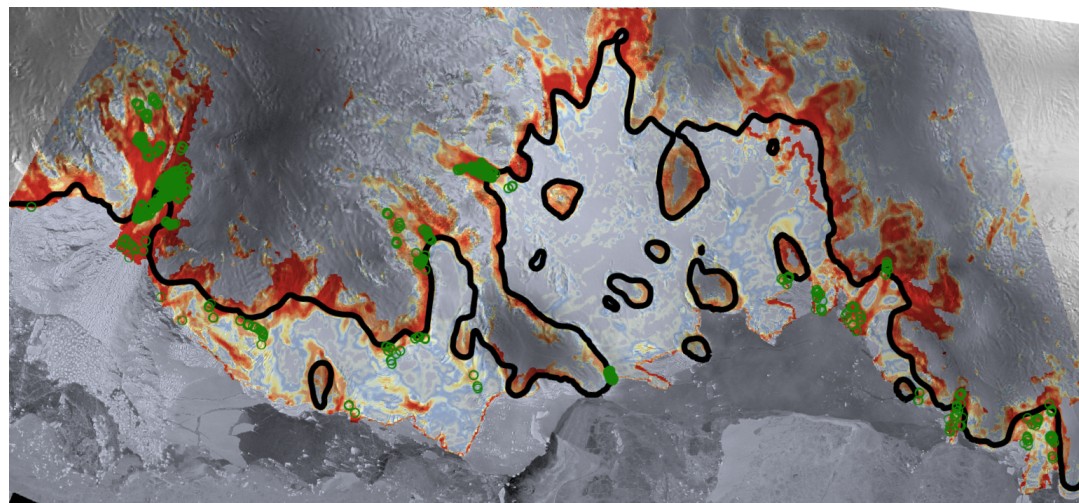

(b) Nickerson and Sulzberger IS, Group 1

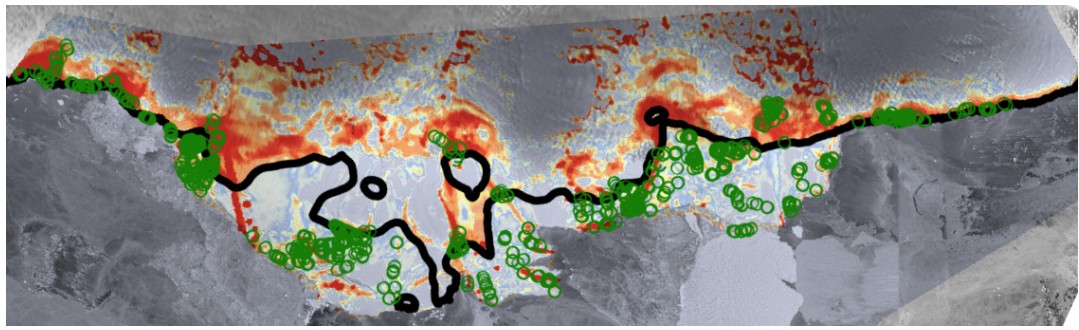

(c) West IS, Group 1

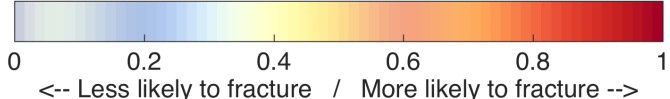

0      0.2      0.4      0.6      0.8      1

<-- Less likely to fracture  /  More likely to fracture -->

**Figure S8.** Group 1: Modelled probability of a fracture for Stange and Ferringo IS (a), Nickerson and Sulzberger IS (b) and West IS (c). Labels the same as Figure S1.

*Author contributions.* Veronika Emetc designed the study, developed the methodology, collected the data, performed the analysis, and wrote the manuscript. Paul Tregoning helped with models and revising the manuscript significantly. Mathieu Morlighem helped with the implementation of the analysis in ISSM, development of the code, implemented some features needed for the model and helped revising the manuscript. Chris Borstad suggested a way to improve the observational set using the damage-based results as well as helped with the damage description and modelling. Malcolm Sambridge developed the source function for the Bayesian analysis.

*Acknowledgements.* We would like to thank Teresa Neeman for her help with Logistic Regression Algorithm. We are also grateful to Anthony Purcell for his always helpful advices and his help revising the manuscript. Veronika Emetc received scholarship from ARC Discovery DP140103679. We would like to thank Julian Byrne for his help with setting up the ISSM software on Terrawulf. We are grateful to the reviewers for their careful and insightful comments on our paper.

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
