# Peer review of "A statistical fracture model for Antarctic ice shelves and glaciers"

_The Cryosphere, 2018_

## Referee Comment (RC1) · Anonymous Referee #1 · 18 Apr 2018

This paper uses a phenomenological approach to mapping crevasses in Antarctic ice-sheets and glaciers. The method uses a set of criterion that correlate with appearance of crevasses. The criteria and the approach is of Bayesian/AI type and lacks completely scientific rigour. However, the mapping is very extensive and match quite well, in most cases, with observed crevasses. The results may therefore be useful data for more rigorous approaches, and may simply because of that reason be published.

---

## Referee Comment (RC2) · J. Bassis (Referee) · 24 Apr 2018

**1   General Appreciation**

This manuscript describes a statistical method to predict the location of fractures in Antarctic ice shelves and glaciers. Overall, there is a welcomed novelty to the authors approach, in which they take observations of fracture and attempt to link these observations with dynamic variables predicted by ice sheet models. This type of analysis can be used to formulate an empirical model of fracture initiation and propagation or as part of a hypotheses testing program. This manuscript seeks to do a little bit of both of these, although the emphasis is on the former. People often have strong prejudices against the former, but this is largely a question of philosophy. I should acknowledge

that my sympathies lie towards the latter and some elements of my review may push the authors in this direction.

Overall, I like the idea of the manuscript, but I have a lot of comments. In particular, I had a very hard time following the methods and discussion and my review is going to focus on many of these elements. There were also quite a few typos or errors in the manuscript (sometimes it was hard to tell which) and I suspect that the manuscript will require a significant rewrite with the full attention of the senior authors to make the manuscript accessible to a wide glaciological audience.

**2 Major comments**

**2.1 Observed fractures vs inferred damage**

One of places where I'm confused is in the data sources that are ingested to compute where ice is fractured. The presentation in Section 2.2 led me to think that surface velocities are used to invert for damage and then damage is used as a proxy for locations where the ice is fractured. However, section 3 states that the authors use fractures when they are visible in satellite imagery. This is then confirmed in section 3.2 where it is stated that satellite images are used to determine when ice is fractured. As far as I can tell from reading the manuscript and figure captions, the authors used satellite imagery to identify surface fractures and these are shown as green dots in Figures 3 onward. The satellite imagery derived fractures were then ingested into the statistical framework and this is used to infer the probability that ice is fractured. My understanding is that damage is only used qualitatively to compare with the probabilities inferred. I really like the idea of using damage as an independent method to compare observed fractures with, but this should be emphasized early on. To make the method clear to readers, I suggest rewriting the methods and background section to introduce

both the damage based method and the satellite based method simultaneously. Or postpone the damage section entirely until the discussion section since it is only used qualitatively and does not factor into the analysis. Given the qualitative nature of the comparison, I also wonder if the details of the inversion can be omitted and replaced with a suitable reference.

**2.2 Location of surface fractures vs location of rifts and other fractures**

I also had a hard time interpreting the location of both damage and fractures identified in satellite imagery. I'm going to focus my comments here on the Amery Ice Shelf and Mertz glacier tongue because I know both regions well. Looking at the observed fractures (green circles in Figure 5), I see quite a few observed fractures on the grounded ice, but very few on the ice shelf. However, I know there are significant crevasses/rifts that originate near Gillock Island on the Amery that form a long crevasse/rift train. There are also several rifts near the front of the ice shelf, including the Loose Tooth, that don't have corresponding observations identified. As far as I can tell, the rifts that are most likely to become detachment boundaries are not clearly represented in the dataset used to infer the locations of fracture! There seems to be some objective criterion used that, unless I misunderstand, doesn't include what I would typically think of a crevasse. Similarly, you can see from Figure 6 that the entire Mertz Ice Tongue is heavily fractured and yet these fractures are not represented in the dataset used. The probability inference is only going to be as good as the data ingested so it is important to explain why most fractures on the ice shelf appear to be ignored.

**2.3 Location of damage vs location of rifts and other fractures**

Similarly, the inference for damage for most regions does not fill me with confidence given the fact that inferred damage occurs in regions where there is little evidence of

fracturing and misses regions of actively propagating rifts. To make things more per-plexing, there is very little agreement between the observed fractures and the damage that was inferred. What exactly is the damage supposed to tell us if it doesn't cor-respond to locations where there are fractures?In theory these two methods should provide independent confirmation of areas that are damaged. The limited overlap be-tween these regions makes me question if the ingested data is limiting the applicability of the results. Here I'm not sure what to suggest, but I do think the authors need to address the discrepancy between observed fractures, rifts visible in MODIS/MISR im-agery and inferred damage and what it does to the results presented. I do wonder if focusing first on a single region that could be studied in detail would be beneficial before attempting to merge many different regions.

**2.4   Choice of variables used as predictors: Part 1 strain or strain rate?**

I'm not sure that I understand the motivation for (or need) for many of the predictors ingested into the probabilistic framework . I should say that I like Tables 2, 4 and 5, which quickly summarizes the different variables considered and the dominant vari-ables. These are great. The text describing the motivation of many of the variables is, however, hard to follow. To start, the authors appear to be confused about the difference between strain and strain rate. Strain is related to the gradient of the dis-placement. Strain rate is related to the gradient of the velocity. These are not the same thing. The authors note multiple times that they are looking at strains and principle strains (e.g, page 7-8). It is, however, unclear how they can get strain: do you accumu-late strain rate over some interval of time? If so, what is the time interval? I think this could be a really interesting calculation, but after multiple readings I think the authors **might** really mean strain rate. This absolutely needs to be clarified.

**2.5 Choice of variables used as predictors: Part 2 what is physical and what is not?**

Some of the variables used as predictors are intuitive and have a long history of usage (often irrespective of whether they are supported by observations or not). Physical variables (in my opinion) include measures of the strain rate tensor and stress/deviatoric stress tensors. Most of the other variables included, especially the geometric variables should correlate with various measures of the stress and/or strain rate. This makes me wonder if these additional variables are needed to make up for deficiencies in the ISSM inferred values for things like stress. The authors also use some measure related to the gradient of the strain rate called strain change. Again, I'm not sure if they really mean strain or strain rate. But the gradient of strain rate, presumably converted to to some scalar measure, might be diagnostic of the presence of fractures rather than predictive. For example, fractures/rifts/crevasses lead to large gradients in the strain rate field across individual fractures. The fractures do not originate because of the change in strain rate. The change in strain rate is telling you that there are fractures present. I can understand why including this as a predictor would improve results, but the causality in this case is almost certainly in the wrong directly. I will also note that because ice is incompressible $\dot{\epsilon}_{zz} = -(\dot{\epsilon}_{xx} + \dot{\epsilon}_{yy})$ and Equation 6 is not obviously correct unless one somehow sets $\dot{\epsilon}_{zz} = 0$. Finally, given that the authors include the effective strain rate, why not also include the effective deviatoric stress invariant as a variable (or the Von Mises stress)?

**2.6 Choice of variables used as predictors: Part 3 a recommendation**

My suspicion is that all of the variables included in the statistical analysis were included because the authors found that they were needed to explain observations (but see my earlier question about the reliability of the observations). I wonder if it would be more physically useful to start with a simpler model that \*\*only\*\* considers one or two

variables. For example, can the authors prove that various measures of strain rate or stress by themselves are not sufficient to explain the observations? Two of the co-authors have made proposals (Borstad and Morlighem) that could be tested given an appropriate dataset. This alone would be a big step forward. Once, the authors demonstrate that stress/strain rate measures alone are not sufficient, then I think the authors could more easily motivate a more elaborate set of tests. But these could be motivated by regions where the statistical model fails. This would have the advantage of providing physical insight in addition to empirical predictions. (Again, note my bias here towards hypothesis testing.) For example, flexural stresses near the grounding line/pinning points are key features that could result in fracture formation and these processes are not included in ISSM. If this is the case I would expect that fractures in these locations would not be resolved. One could then include additional variables that could diagnose flexural stresses. The advantage of this approach to someone like myself, that is mechanically inclined, is that it tells me about the processes that are important and need to be included in models.

2.7   What about fracture advection?

There is also an issue that the authors hint at, but don't quite address which is that fractures advect after they form. A consequence is that places you observe fractures may be far from the places they are observed. Because stresses and strain rates have not been constant, this means that the state of stress when a fracture formed could have been very different than it is now. Moreover, the fracture could evolve based on the integral of strain rate/stress tensor invariants over the life time of fractures. Some of the fractures observed may be diagnostic of stress regimes hundreds or thousands of years ago and hence not that useful to the analysis.

[Figure]

**2.8 What about a yield strength?**

Most theories suggest that fractures form when some measure of the stress (or in some cases, strain rate tensor) exceed some material dependent parameter. It is not clear to me how this type of threshold behavior is incorporated into the statistical model. What happens when a larger (smaller) strain rate doesn't lead to more (less) fractures, but that there is an abrupt (or rapid) transition centered around some yield envelope?

**2.9 Tables and Figures need a bunch of work**

As I said, I like some of the Tables, but I don't see the point of Table 1.

Figure 1: I don't understand the colors or the content. For one, the legends have colors that don't correspond to the colors of the figure (e.g., Fig 1a has pink and green legend but bars are pink and brown).

Figures 3-10 need better captions and some attention. - What do the cyan boxes represent? - Can you include a small location box for each region? - What does it mean to show Group X for a particular region? - Figure 5 (a) appears to denote a place called Pain Isalnd. I am going to guess that is supposed to be Pine Island. - Figure 5b and c know show two different views of the Amery Ice Shelf with different color scales, but the caption tells us everything is identical to previous figures. I'm guessing that Figure 5c shows inferred damage. This needs to be in the caption. Figure 5c also seems to have some red dots that aren't described in the caption. - Why is damage only introduced in Figure 5 onwards? Why not earlier? What am I supposed to see in these figures? - In Figure 5b-c, the entire area around Gillock Island appears to have no observations or model results. This makes me wonder what was used as boundary conditions and how reliable the results can be given that many crevasses originate around Gillock Island. - Figure 6, now you tell us that red dots denote fractures that were filtered out due to damage upstream. What does this mean? Is this the same in Figure 5? This figure is

extremely frustrating because Figures 6a and 6b appear to show the same ice tongue, but the size and orientation are completely different making it impossible to compare. - Figure 8, again with the cyan boxes? What do those represent and why aren't they included in the captions? - Why not use the same color scale for probability as damage to make it easier to compare?

**2.10 Writing and style**

There were quite a few typos in the manuscript and these need to be fixed to ease the exposition. Given the number of typos I wasn't sure if some of issues I found were typos or errors (see strain vs strain rate). The manuscript needs a very careful scrubbing and editing to tighten the prose. This should be supervised by the senior authors of the study. The context around different approaches is not entirely correct in the introduction and background section. To my knowledge **no** method has been able to simulate the diversity of calving regimes observed. Damage mechanics is, in theory, able to simulate failure of grounded and floating ice. However, the approaches cited rely on small scale laboratory data, which may not apply to large scale glaciers. Moreover, viscoelastic damage mechanics is an approach that if often used to simulate the propagation of individual fractures. This can be prohibitive in large-scale models. Hence, the approach by Borstad et al. As far as I can tell, the approach by Borstad works really well for the Larsen ice shelves and is quite promising for ice shelves in general. I don't think this approach has been applied to grounded ice before. Similarly, the efforts by Levermann (eigen calving) seem like they work OK for floating ice. The Von Mises criterion (Morlighem) seems promising for grounded ice.

**3 Technical comments**

**3.1 How damage is defined and calculated?**

Damage is implicitly defined in equation 1, but the assumptions are a bit unclear. The stiffness parameter is a strong function of temperature. However, for ice shelves, we can often approximate the flow as plug flow and thus the stiffness parameter that is relevant is the depth averaged quantity:

$$\bar{B}_i = \frac{1}{H} \int_b^s B_T(z)(1 - D(z))\mathrm{d}z. \tag{1}$$

where $H$ is the ice thickness, $s$ is the surface elevation and $b$ is the bottom of the ice. Both temperature and damage will depend on the vertical coordinate $z$ and the integral cannot be done analytically. In the special case that damage or temperature is constant with depth, the integral can be done analytically. As far as I can tell, the authors are assuming that damage is independent of depth and thus they write:

$$\bar{B}_i = \frac{1}{H} \int_b^s B_T(z)(1 - D)\mathrm{d}z = (1 - D)\bar{B}_T. \tag{2}$$

and this leads to Equation 1. Given how little we know about damage in general and its depth dependence in specific this is perhaps a plausible assumption. However, the interpretation of constant damage with depth differs significantly from the observation of surface fractures in satellite images where it is unlikely that all fractures penetrate the entire ice thickness. This might explain why damage has little relation to observed crevasses and merits some comment.
**3.2 Damage inference**

There is also an issue with inferring damage based on the viscosity. The inferred value will depend sensitivity on ice temperature. Errors in assumed ice temperature will contaminate the damage calculation. That is inevitable, but should be acknowledged. What is exciting here is that the authors appear to have independent estimates of damage from satellite observations and this suggests that damage can be compared independently (subject to the many above caveats). I would personally like to see more of this, but that might be a different manuscript. Given my previous comments about the weird places damage is inferred, I do wonder if the damage calculated for some of the locations is fiercely contaminated by bad temperature estimates. Damage on the Amery Ice Shelf seems to be especially suspicious. However, because damage is always less than unity and, I assume errors in ice temperature are more gaussian distributed, one might be able to examine the frequency with which the model would prefer an ice viscosity that is stiffer than inferred from the temperature field alone (negative damage). If this is vanishingly rare than one would have significant confidence in the damage estimate. Perhaps that is what is going on in some places, like the Amery, where damage is inferred in unphysical locations?

**4 Minutia**

- Low friction can lead to larger tensile stresses, but won't this also lead to larger tensile strain rates? If strain rates and stresses are included in the model this seems redundant.

- Why does ice stiffness factor into the calculation? The fracture properties of ice have little sensitivity to ice temperature? I wonder here if this is getting at problems with estimating the temperature of ice in ISSM.
- Page 2, line 11: "There have been a number of approaches that successfully modelled rift formation on particular ice shelves," Really? I'm not sure that anyone has successfully modeled rift initiation and propagation.

- Page 2, line 21: this observations-> these observations

- Page 2, line 15 and down. This seems like discussion/abstract/results and without knowing more about the method is a bit confusing. I suggest moving this to later in the paper.

- Page 3: Discrete Element Models are also used to predict short-term calving events

- Page 3: "There are a number of other studies that proposed other calving laws (Pralong and Funk, 2005; Duddu and Waisman, 2012), but they might be not applicable in a generalised large-scale case." This is probably true, but is as much true as any of the other methods. These continuum damage mechanics methods can model the propagation of crevasses in the vertical and horizontal directions, but rely on calibrating to old and-perhaps-unreliable laboratory data. These methods can include hydrofracture and other modes of failure, but have largely been applied to grounded calving margins. The methods by Borstad in contrast, are calibrated to field data and have been applied to ice shelves, but not grounded calving margins. It isn't obvious to me how to include hydrofracture in the Borstad method.

- Page 4, paragraph near line 30: I don't think Duddu and Waisman applied their model to any specific model in Greenland. This was largely a prototype model that was applied using idealized geometries. The model, however, was calibrated to laboratory data and in theory this data should remain valid for any loading situation.
- Page 6, line 10: What units are the friction coefficient and what sliding law was used?

- Page 7, line 13: What do you mean velocity gradients? Strain rates are related to the gradient of the velocity. Do you mean that you also include vorticity as a predictor or do you mean the gradient of the strain rates? Also, how do you measure strains as opposed to strain rates?

- Page 7 line 26: missing space after Each

- Page 8, line 10: again how can you calculate strains from a viscous model? I can see how to get strain rates from ISSM, but strains seem to require an elastic component that is missing.

- Page 8, line "to model a gradual viscous process strains have to be taken into account" I don't understand this statement. What is a gradual viscous process and what does it have to do with strains. Viscous processes are usually a function of strain rates.

And this is where I stopped noting small quibbles with wording.

---

## Author Comment (AC1) · 12 Jun 2018

We would like to thank the referee for his comments and acknowledging that the results might be useful for future methods. Since no modifications were suggested we did not make any changes.
* * *

---

## Author Comment (AC2) · 12 Jun 2018

AC: We thank Jeremy Bassis for his careful and insightful review of our paper. His comments significantly improved the clarity of the manuscript.

RC: 1 General Appreciation This manuscript describes a statistical method to predict the location of fractures in Antarctic ice shelves and glaciers. Overall, there is a welcomed novelty to the author's approach, in which they take observations of fracture and attempt to link these observations with dynamic variables predicted by ice sheet models. This type of analysis can be used to formulate an empirical model of fracture initiation and propagation or as part of a hypotheses testing program. This manuscript seeks to do a little bit of both of these, although the emphasis is on the former. People often have strong prejudices against the former, but this is largely a question of philosophy. I should acknowledgethat my sympathies lie towards the latter and some elements of my review may push the authors in this direction. Overall, I like the idea of the manuscript, but I have a lot of comments. In particular, I had a very hard time following the methods and discussion and my review is going to focus on many of these elements. There were also quite a few typos or errors in the manuscript (sometimes it was hard to tell which) and I suspect that the manuscript will require a significant rewrite with the full attention of the senior authors to make the manuscript accessible to a wide glaciological audience.

AC: We revised significantly the sections "Ice Sheet System Model (ISSM) Model setup" and "Methods" following the reviewer's advice.

RC: . Major comments 2.1 Observed fractures vs inferred damage One of places where I'm confused is in the data sources that are ingested to compute where ice is fractured. The presentation in Section 2.2 led me to think that surface velocities are used to invert for damage and then damage is used as a proxy for locations where the ice is fractured.

AC: This is not entirely correct. We do not use damage as a proxy for locations where the ice is fractured. Damage is used only for qualitative comparison against our model and observed fractures.

RC: However, section 3 states that the authors use fractures when they are visible in satellite imagery.

AC: Correct. We do use observed fractures to construct our statistical model as well as a ground-truth for our modelled results. We do not use damage to construct our statistical model. We moved the whole section discussing damage-based method to the end of the methods section to make it clearer that the damage modelling does not form a part of our model.

RC: This is then confirmed in section 3.2 where it is stated that satellite images are used to determine when ice is fractured.

AC: Correct.

RC: As far as I can tell from reading the manuscript and figure captions, the authors used satellite imagery to identify surface fractures and these are shown as green dots in Figures 3 onward. The satellite imagery derived fractures were then ingested into the statistical framework and this is used to infer the probability that ice is fractured. My understanding is that damage is only used qualitatively to compare with the probabilities inferred.

AC: This is essentially correct. However, we do not compare damage with probabilities. We state in section 3.6 of the paper: "We do not compare our probability-based model with the damage model directly; rather, we evaluate their respective ability to predict the formation of fractures in ice." We added the following sentence to clarify our methodology: "We utilise the damage-based model as an independent method in order to compare it with the observations of fractures and to identify areas where it can and cannot accurately predict the presence of fractures." (we clarified the text in other places as well, see below)

RC: I really like the idea of using damage as an independent method to compare observed fractures with, but this should be emphasized early on.

AC: We have added a sentence in the end of the introduction section to make this clear.

RC: To make the method clear to readers, I suggest rewriting the methods and background section to introduce both the damage based method and the satellite based method simultaneously. Or postpone the damage section entirely until the discussion section since it is only used qualitatively and does not factor into the analysis.

AC: We chose the latter option. We moved the description of the damage method to the end of the methods section.

RC: Given the qualitative nature of the comparison, I also wonder if the details of the inversion can be omitted and replaced with a suitable reference.

AC: A previous reviewer suggested that we should include the equations, therefore we would like to keep them.

RC: 2.2 Location of surface fractures vs location of rifts and other fractures I also had a hard time interpreting the location of both damage and fractures identified in satellite imagery. I'm going to focus my comments here on the Amery Ice Shelf and Mertz glacier tongue because I know both regions well. Looking at the observed fractures (green circles in Figure 5), I see quite a few observed fractures on the grounded ice, but very few on the ice shelf.

AC: Some fracture observations might be left out of the set we used due to the fact that we identified the location of all the fractures manually, however missing a few fractures would not significantly affect the statistical model as a whole. We already explained this in the paper (section 3): "We did not need to select all the fractures on the ice sheet surface to build the statistical model but, in order to compare the results of our model with observations, we constructed extra data sets where we made a concerted effort to select all the visible fractures on the ice surface. It is possible that some fractures were missed due to the large spatial extent of the experiments."

RC: However, I know there are significant crevasses/rifts that originate near Gillock Island on the Amery that form a long crevasse/rift train. There are also several rifts near the front of the ice shelf, including the Loose Tooth, that don't have corresponding observations identified. As far as I can tell, the rifts that are most likely to become detachment boundaries are not clearly represented in the dataset used to infer the locations of fracture! There seems to be some objective criterion used that, unless I misunderstand, doesn't include what I would typically think of a crevasse. Similarly, you can see from Figure 6 that the entire Mertz Ice Tongue is heavily fractured and yet these fractures are not represented in the dataset used. The probability inference is

only going to be as good as the data ingested so it is important to explain why most fractures on the ice shelf appear to be ignored.

AC: This is an important point that we probably did not explain well. The definition of rifts and crevasses is different, and our paper attempts to model fracture processes, not rifting. We have now made this clear in the beginning of section 3.

RC: 2.3 Location of damage vs location of rifts and other fractures Similarly, the inference for damage for most regions does not fill me with confidence given the fact that inferred damage occurs in regions where there is little evidence of fracturing and misses regions of actively propagating rifts. To make things more perplexing, there is very little agreement between the observed fractures and the damage that was inferred. What exactly is the damage supposed to tell us if it doesn't correspond to locations where there are fractures?

AC: We agree. The focus of our paper is on the ability of our model to match the observed fractures, not on explaining why the damage approach can not always match the observations. RC: In theory these two methods should provide independent confirmation of areas that are damaged. The limited overlap between these regions makes me question if the ingested data is limiting the applicability of the results.

AC: Because we do not rely on the damage predictions to construct our model, the limitations of the damage approach do not affect our results. We stated this indirectly in section 3.6 where we indicate that our model construction depends only upon the probabilistic approach.

RC: Here I'm not sure what to suggest, but I do think the authors need to address the discrepancy between observed fractures, rifts visible in MODIS/MISR imagery and inferred damage and what it does to the results presented.

AC: The focus of our paper is on the performance of our own model, not of the damage modelling that has been published previously. The aim of this paper to develop an al-

ternative method that can predict fractures well. We do not want to change the focus of our paper. To reduce the apparent importance of the discussion of damage modelling, we have moved it to the end of section 3.

RC: I do wonder if focusing first on a single region that could be studied in detail would be beneficial before attempting to merge many different regions.

AC: The statistical approach does not work well if the data are taken from just a few glaciers, therefore working with many regions is absolutely necessary. The wider the range of data, the more robust the statistical model. Added to section 3: "This large number of ice shelf regions was chosen due to the fact that the statistical approach does not work well if we train the model on too few glaciers. Taking a wider range of input data (using different regions in Antarctica) is more important than having a larger number of similar data points. Variety of observations of different regimes improves the reliability of the statistical model, as the diversity in sampling provides a better estimation of correlation coefficients for the statistical model (called $\beta$ coefficients in LRA). Thus, by choosing multiple glaciers we can more accurately construct an approximate surface that separates fractured from non-fractured nodes (the plane is determined by $\beta$ coefficients)."

RC: 2.4 Choice of variables used as predictors: Part 1 strain or strain rate? I'm not sure that I understand the motivation for (or need) for many of the predictors ingested into the probabilistic framework. I should say that I like Tables 2, 4 and 5, which quickly summarizes the different variables considered and the dominant variables. These are great. The text describing the motivation of many of the variables is, however, hard to follow.

AC: We added an enumeration in the text to make this part clearer. We also revised the text to improve clarity.

RC: To start, the authors appear to be confused about the difference between strain and strain rate. Strain is related to the gradient of the displacement. Strain rate is

related to the gradient of the velocity. These are not the same thing. The authors note multiple times that they are looking at strains and principle strains (e.g, page 7-8). It is, however, unclear how they can get strain: do you accumulate strain rate over some interval of time? If so, what is the time interval? I think this could be a really interesting calculation, but after multiple readings I think the authors \*\*might\*\* really mean strain rate. This absolutely needs to be clarified.

AC: We indeed use strain rates, thank you for pointing this out. We modified the paper accordingly.

RC: 2.5 Choice of variables used as predictors: Part 2 what is physical and what is not? Some of the variables used as predictors are intuitive and have a long history of usage (often irrespective of whether they are supported by observations or not). Physical variables (in my opinion) include measures of the strain rate tensor and stress/deviatoric stress tensors. Most of the other variables included, especially the geometric variables should correlate with various measures of the stress and/or strain rate. This makes me wonder if these additional variables are needed to make up for deficiencies in the ISSM inferred values for things like stress.

AC: this may be true, but is not entirely relevant. Our statistical approach makes use of whatever indicators are available and relevant. If a particular parameter was superfluous then the LRA would, by construction, ignore it and that parameter would not have been included in our optimised set of predictors. This circumvents the issue raised here, however, we added an explanation of this in section 3.3.1.

RC: The authors also use some measure related to the gradient of the strain rate called strain change. Again, I'm not sure if they really mean strain or strain rate. we meant strain rate. This has been corrected.

RC: But the gradient of strain rate, presumably converted to some scalar measure, might be diagnostic of the presence of fractures rather than predictive. For example, fractures/rifts/crevasses lead to large gradients in the strain rate field across individual
fractures. The fractures do not originate because of the change in strain rate. The change in strain rate is telling you that there are fractures present. I can understand why including this as a predictor would improve results, but the causality in this case is almost certainly in the wrong directly.

AC: We agree with the reviewer that the change in strain rate tells us that there are fractures present. This is precisely why our new approach works. We use this as a predictor precisely because it tells us where we can expect fractures. Our method does not attempt to understand the process by which the fractures formed. Rather, our aim is to identify where they form. We would still discover new regions where crevasses form even if they was not observed there in the first place for two reasons. First, if there are no observed fractures but the strain rate is high it means that fractures are not visible but should be still there (maybe covered in snow or bad resolution images). Second, if strain rate is small we have other predictors in the model to tell us if there are fractures or not. Added to section 3.3 (iii): " It is important to note that the change in strain rate is not the cause but the result of presence of fractures. However, the aim of our study is to identify where fractures are present without attempting to fully describe the process by which they are formed. Using the strain rates as a predictor we would still discover new regions where crevasses form even if they were not observed there in the first place for two reasons. First, having no observed fractures but high strain rate means that fractures are not visible but should be still present (they can be covered in snow or not visible due to bad resolution of the satellite images). Second, if strain rate is small we have other predictors in the model to tell us if there are fractures or not."

RC: I will also note that because ice is incompressible and Equation 6 is not obviously correct unless one somehow sets $\varepsilon zz = 0$. Finally, given that the authors include the effective strain rate, why not also include the effective deviatoric stress invariant as a variable (or the Von Mises stress)?

AC: We corrected the equations

RC: 2.6 Choice of variables used as predictors: Part 3 a recommendation My suspicion is that all of the variables included in the statistical analysis were included because the authors found that they were needed to explain observations (but see my earlier question about the reliability of the observations). I wonder if it would be more physically useful to start with a simpler model that **only** considers one or two variables. For example, can the authors prove that various measures of strain rate or stress by themselves are not sufficient to explain the observations? Two of the co-authors have made proposals (Borstad and Morlighem) that could be tested given an appropriate dataset. This alone would be a big step forward. Once, the authors demonstrate that stress/strain rate measures alone are not sufficient, then I think the authors could more easily motivate a more elaborate set of tests. But these could be motivated by regions where the statistical model fails

AC: Following the reviewer's recommendation we ran additional experiments with only stress measures.

RC: This would have the advantage of providing physical insight in addition to empirical predictions. (Again, note my bias here towards hypothesis testing.) For example, flexural stresses near the grounding line/pinning points are key features that could result in fracture formation and these processes are not included in ISSM. If this is the case I would expect that fractures in these locations would not be resolved. One could then include additional variables that could diagnose flexural stresses. The advantage of this approach to someone like myself, that is mechanically inclined, is that it tells me about the processes that are important and need to be included in models.

AC: We added to the Methods a new subsection "Test runs with a small set of parameters" and included plots showing a number of test results obtained when only using the effective stress, when using principal deviatoric stresses and the last when using von Mises stress expression. We now show in this section that including only stress parameters does not yield a sufficiently accurate model.

[Figure]

RC: What about fracture advection?

There is also an issue that the authors hint at, but don't quite address which is that fractures advect after they form. A consequence is that places you observe fractures may be far from the places they are observed. Because stresses and strain rates have not been constant, this means that the state of stress when a fracture formed could have been very different than it is now. Moreover, the fracture could evolve based on the integral of strain rate/stress tensor invariants over the life time of fractures. Some of the fractures observed may be diagnostic of stress regimes hundreds or thousands of years ago and hence not that useful to the analysis.

AC: The direct modeling of advection of the fractures is beyond the scope of this research. We do not know just by looking at the images if the small crevasses we select are initiated or advected in that particular location, therefore we model surface crevasses without distinguishing if they are advected or initiated ones. However, the fact that what may be advected fractures are still visible provides information about the particular conditions at a location.

To clairfy this, we have added: "Our main goal is to determine the most likely location of fractures without focusing on their initial source, since we can not claim if the observations consist only of initiated crevasses. Although we do not directly model advection, the statistical model predicts the presence of fractures (both initiated and advected fractures without distinguishing one from another). The question that arises then is how do we know that the flow regime conditions that caused opening of the fracture are the conditions in the observed point and not the conditions upstream from the observed fracture (in case of advection)? However, even if an observed fracture was not formed at a particular location, but was advected with the ice flow it is still visible on the satellite image. The fact that fractures can be seen indicates that there are factors that act to permit the fractures to exist, whether they formed in that particular location or remained open after being advected from upstream (since another combination of factors could close the fracture).

RC: extremely frustrating because Figures 6a and 6b appear to show the same ice tongue, but the size and orientation are completely different making it impossible to compare.

AC: This was an accidental rotation of the figure, which we have now corrected.

RC: - Figure 8, again with the cyan boxes? What do those represent and why aren't they included in the captions? - Why not use the same color scale for probability as damage to make it easier to compare? a previous reviewer specifically asked for the information to be provided in a different colour. We have chosen to leave it as it is.

RC: 2.10 Writing and style There were quite a few typos in the manuscript and these need to be fixed to ease the exposition. Given the number of typos I wasn't sure if some of issues I found were typos or errors (see strain vs strain rate). The manuscript needs a very careful scrubbing and editing to tighten the prose. This should be supervised by the senior authors of the study.

AC: The manuscript has now been read carefully by all authors.

RC: The context around different approaches is not entirely correct in the introduction and background section. To my knowledge **no** method has been able to simulate the diversity of calving regimes observed. Damage mechanics is, in theory, able to simulate failure of grounded and floating ice. However, the approaches cited rely on small scale laboratory data, which may not apply to large scale glaciers. Moreover, viscoelastic damage mechanics is an approach that if often used to simulate the propagation of individual fractures. This can be prohibitive in large-scale models. Hence, the approach by Borstad et al. As far as I can tell, the approach by Borstad works really well for the Larsen ice shelves and is quite promising for ice shelves in general. I don't think this approach has been applied to grounded ice before. Similarly, the efforts by Levermann (eigen calving) seem like they work OK for floating ice. The Von Mises criterion (Morlighem) seems promising for grounded ice.
3 Technical comments 3.1 How damage is defined and calculated? Damage is implicitly defined in equation 1, but the assumptions are a bit unclear. The stiffness parameter is a strong function of temperature. However, for ice shelves, we can often approximate the flow as plug flow and thus the stiffness

parameter that is relevant is the depth averaged quantity:

where H is the ice thickness, s is the surface elevation and b is the bottom of the ice. Both temperature and damage will depend on the vertical coordinate z and the integral cannot be done analytically. In the special case that damage or temperature is constant with depth, the integral can be done analytically. As far as I can tell, the authors are assuming that damage is independent of depth and thus they write: BT (z)(1 − D)dz = (1 − D)BT . (2) and this leads to Equation 1. Given how little we know about damage in general and its depth dependence in specific this is perhaps a plausible assumption. However, the interpretation of constant damage with depth differs significantly from the observation of surface fractures in satellite images where it is unlikely that all fractures penetrate the entire ice thickness. This might explain why damage has little relation to observed crevasses and merits some comment.

AC: This is a good point, but again we iterate that the focus of our paper is not the damage results but the agreement of our new statistical approach with the observed fractures. Nonetheless, we added: "We use the method suggested by Borstad, 2016 where it is assumed that damage is independent of depth." to provide clear reference to the source of the damage model used.

RC: 3.2 Damage inference There is also an issue with inferring damage based on the viscosity. The inferred value will depend sensitivity on ice temperature. Errors in assumed ice temperature will contaminate the damage calculation. That is inevitable, but should be acknowledged.

AC: This is true, and we mentioned it the paper already: " Estimation of B_T is the source of the main uncertainty in damage calculations due to the lack of ice temperature data, which can be crucial in affecting the accuracy of the viscosity parameter". To make it clearer we added: "Thus, the errors in assumed temperature may affect the inferred value for damage."

RC: What is exciting here is that the authors appear to have independent estimates of damage from satellite observations and this suggests that damage can be compared independently (subject to the many above caveats). I would personally like to see more of this, but that might be a different manuscript. Given my previous comments about the weird places damage is inferred, I do wonder if the damage calculated for some of the locations is fiercely contaminated by bad temperature estimates. Damage on the Amery Ice Shelf seems to be especially suspicious. However, because damage is always less than unity and, I assume errors in ice temperature are more gaussian distributed, one might be able to examine the frequency with which the model would prefer an ice viscosity that is stiffer than inferred from the temperature field alone (negative damage). If this is vanishingly rare than one would have significant confidence in the damage estimate. Perhaps that is what is going on in some places, like the Amery, where damage is inferred in unphysical locations?

AC: This is interesting, but is not the focus of our paper. We have chosen not to add such information to our manuscript.

4 Minutia

RC: Low friction can lead to larger tensile stresses, but won't this also lead to larger tensile strain rates? If strain rates and stresses are included in the model this seems redundant.

AC: In all cases, each group includes either strain rates or friction, not both at the same time. We run an experiment replacing friction by strain rates and found that the prediction success for some glacier decreases by about 5%. We clarified this by adding information on additional experiments that we conducted: "Moreover, including both friction and strain rate is ambiguous since lower friction can lead to a larger strain

rate. However, by looking at the predictor data sets, we found that the optimal choice of parameters for each group includes either friction or strain rate, never both at the same time. We ran an experiment replacing friction by strain rates and found that the prediction success for some glacier decreases by about 5%. We therefore kept only friction as a predictor parameter.

RC: Why does ice stiffness factor into the calculation? The fracture properties of ice have little sensitivity to ice temperature? I wonder here if this is getting at problems with estimating the temperature of ice in ISSM.

AC: The reviewer is probably correct here. We don't include ice temperature in our model because the resolution of the values is not good. We found that using ice stiffness is a better indicator. We added: "In addition, we include the stiffness of ice as well as thickness due to their physical relation to fracture mechanics. When ice stiffness is high and ice crystals cannot creep fast enough, fracture might occur. Therefore, this parameter (obtained from the inversion of velocities implemented in ISSM) is added as a predictor. Adding temperature directly into the analysis did not improve the prediction results, which might be due to the uncertainties in the temperature estimation "

RC: Page 6, line 10: What units are the friction coefficient and what sliding law was used?

AC: The units is sqrt(s/m), however this section has been removed in our editing of the manuscript. This has been added: Budd sliding law (Budd et al. 1979)

RC: Page 7, line 13: What do you mean velocity gradients? Strain rates are related to the gradient of the velocity. Do you mean that you also include vorticity as a predictor or do you mean the gradient of the strain rates? Also, how do you measure strains as opposed to strain rates?

AC: We have modified the entire manuscript to refer to only strain rates, not strains. Also, we removed "velocity gradients", since these are not used.

RC: Page 7 line 26: missing space after Each

AC: Corrected

RC: Page 8, line 10: again how can you calculate strains from a viscous model? I can see how to get strain rates from ISSM, but strains seem to require an elastic component that is missing.

AC: It is strain rates. This has been modified everywhere in the paper.

RC: Page 8, line "to model a gradual viscous process strains have to be taken into account" I don't understand this statement. What is a gradual viscous process and what does it have to do with strains. Viscous processes are usually a function of strain rates. And this is where I stopped noting small quibbles with wording.

AC: Corrected to strain rates

---

## Editor Decision (ED1)

[revised manuscript text omitted]

This paragraph could be improved and reduced by avoinding first listing the model and then going into the description of each one.

| | | | |
|---|---|---|---|
| **T** Nombre : 3 | Auteur : ogagliardini | Sujet : Texte surligné | Date : 25/08/2018 19:24:55 |

give a paper reference not the name of a specific model

| | | | |
|---|---|---|---|
| **T** Nombre : 4 | Auteur : ogagliardini | Sujet : Texte surligné | Date : 25/08/2018 19:25:48 |

Elmer/Ice

| | | | |
|---|---|---|---|
| **T** Nombre : 5 | Auteur : ogagliardini | Sujet : Texte surligné | Date : 25/08/2018 19:26:37 |

.

| | | | |
|---|---|---|---|
| **T** Nombre : 6 | Auteur : ogagliardini | Sujet : Texte surligné | Date : 25/08/2018 19:27:27 |

already defined above

| | | | |
|---|---|---|---|
| **T** Nombre : 7 | Auteur : ogagliardini | Sujet : Texte surligné | Date : 25/08/2018 22:44:53 |

Which method are you referencing to? Not clear from the previous sentence.

| | | | |
|---|---|---|---|
| **T** Nombre : 8 | Auteur : ogagliardini | Sujet : Texte surligné | Date : 25/08/2018 19:29:44 |

Krug et al. (2014) built an alternative scheme by combining LEFM and CDM and found that they could match the observed evolution of a tidewater glacier in Greenland. This method is more complex compared to earlier approaches as it allows for both viscous and elastic behaviour and is able to reproduce development of small crevasses over a long period of time. The ELMER/Ice model (Gagliardini et al., 2013) combines CDM and LEFM to model calving, but this method has only been applied to Greenland (Krug et al., 2014). There are a number of studies that have proposed other calving laws (Pralong and Funk, 2005; Duddu and Waisman, 2012). These methods can include hydrofracture and other modes of failure, but have largely been applied to grounded calving margins, in contrast to the methods by Borstad et al. (2016) that are calibrated to remote sensing data and have been applied to ice shelves, but not grounded calving margins. Moreover, most of the mentioned methods might be not applicable in a generalised large-scale case.

**3  Observational datasets**

**3.1  Ice Sheet System Model (ISSM) setup**

Our statistical model is built upon knowledge of the velocities, stresses, strain rates, back stresses as well as friction coefficient and viscosity of the ice. We use ISSM (Larour et al., 2012) to derive estimates of these predictors for our statistical model. ISSM is a fully dynamic model that includes both two dimensional (2-D) and 3-D stress balance approximations. Our experiments rely on the shelfy stream approximation (SSA) as it is computationally cheap and suitable for modelling floating ice shelves and grounded ice streams undergoing widespread basal sliding.

All our experiments were performed for 45 ice shelf regions in Antarctica (see Figure 3a), each including both ice shelves and the grounded ice around 100 kilometres upstream from the grounding line (hereafter referred to as ice shelf regions or ice shelf/glacier). We ran one simulation to create a stress balance solution per region (ice shelf/glacier), which allowed us to obtain the predictor parameters required for the calculation of the probability of fracturing. We used SeaRISE air temperature, snow accumulation and geothermal heat flux (Le Brocq et al., 2010) as climate forcing data. The information about the ice temperature for grounded ice is calculate as the depth-average temperature from Liefferinge and Pattyn (2013); Pattyn (2010) (21 vertical levels). The steady-state depth averaged ice temperature on floating ice shelves (mainly used for the calculation of damage) was calculated using surface, basal temperatures and basal melting rate according to Holland and Jenkins (1999). To calculated ice temperature we corrected the surface temperature with a lapse rate and imposed it on the ice surface. Basal melting rates on ice shelves were taken from (Depoorter et al., 2013).

The data for the geometry of the ice shelves and surrounded grounded ice (bedrock topography, ice thickness and glacier surface) were interpolated from Bedmap2 data (Fretwell et al., 2013) at 1 km spatial resolution. Basal friction under grounded ice and rheology for floating ice were calculated from an inversion of velocities (Khazendar et al., 2007), where the observations of the horizontal ice velocities were taken from InSAR (450-metres resolution) (Rignot et al., 2011b, a) and the sliding law is the Budd sliding law (Budd et al., 1979). In the inversions we used regularisation to penalise sharp gradients of the cost function, calibrated using an L-curve analysis (Morlighem et al., 2013). We set boundary conditions as follows: the upper surface is considered stress-free and friction is applied at the ice-bedrock interface. At the inflow boundary we applied Dirichlet

**Page : 4**

| | | | |
|---|---|---|---|
| **T** Nombre : 1 | Auteur : ogagliardini | Sujet : Texte surligné | Date : 25/08/2018 19:30:33 |

should be above?

| | | | |
|---|---|---|---|
| **T** Nombre : 2 | Auteur : ogagliardini | Sujet : Texte surligné | Date : 25/08/2018 19:32:03 |

might not be applicable

| | | | |
|---|---|---|---|
| **T** Nombre : 3 | Auteur : ogagliardini | Sujet : Texte surligné | Date : 25/08/2018 19:33:56 |

is that the correct name for this section?

| | | | |
|---|---|---|---|
| **T** Nombre : 4 | Auteur : ogagliardini | Sujet : Texte surligné | Date : 25/08/2018 19:34:10 |

what is the BC on this fictitious boundary

| | | | |
|---|---|---|---|
| **T** Nombre : 5 | Auteur : ogagliardini | Sujet : Texte surligné | Date : 25/08/2018 19:35:29 |

| | | | |
|---|---|---|---|
| **T** Nombre : 6 | Auteur : ogagliardini | Sujet : Texte surligné | Date : 25/08/2018 19:36:54 |

applying which velociy ? Observation?

[revised manuscript text omitted]

where j is ?

| T | Nombre : 2 | Auteur : ogagliardini | Sujet : Texte surligné | Date : 25/08/2018 19:53:44 |
|---|---|---|---|---|

:

| T | Nombre : 3 | Auteur : ogagliardini | Sujet : Texte surligné | Date : 25/08/2018 19:54:23 |
|---|---|---|---|---|

calculate each predictor parameter as well as a brief description as to why each parameter may have an impact on the location of fractures:

(i) Principal values of the deviatoric stress and effective stress:

Following the Shallow ice approximation, the devatoric stress is:

$$\sigma' = \begin{bmatrix} \sigma'_{xx} & \sigma'_{xy} & 0 \\ \sigma'_{xy} & \sigma'_{yy} & 0 \\ 0 & 0 & -\sigma'_{xx} - \sigma'_{yy} \end{bmatrix} \tag{4}$$

The devatoric stress values have a direct effect on the opening and closing of crevasses; the sign of the first principal stress component determines whether it is compressive (negative) or tensile (positive). Effective deviatoric stress is calculated as:

$$\sigma'_e = \sqrt{\sigma'^2_{xx} + \sigma'^2_{yy} + \sigma'^2_{xy} + \sigma'_{xx}\sigma'_{yy}}, \tag{5}$$

where $\sigma'_{ij}$ are the deviatoric stress components.

Von Mises stress is calculated as:

$$\sigma_{vm} = \sqrt{\frac{3}{2} \sum_{i,j} \sigma'_{ij}\sigma'_{ij}} = \sqrt{3} B \dot{\varepsilon}_e^{1/n} \tag{6}$$

where $B$ and $n$ are the creep parameter and the creep exponent, respectively.

(ii) Effective strain rate:

The effective strain rate $\dot{\varepsilon}_e$ is included in our analysis because it is known that crevasse initiation is linked to strain rates (Campbell et al., 2013). If the strain rate in the horizontal plane is sufficiently high, crevasses can propagate to greater depth (Benn and Evans, 2010). In addition, stresses can trigger brittle fracturing but, to take into account a gradual viscoelastic effect that can lead to fracture formation, strain rates are included in our model.

The principal strain rates are calculated as eigenvalues of the matrix:

$$\dot{\varepsilon} = \begin{bmatrix} \frac{\partial u}{\partial x} & \frac{1}{2}\left(\frac{\partial u}{\partial y} + \frac{\partial v}{\partial x}\right) & 0 \\ \frac{1}{2}\left(\frac{\partial u}{\partial y} + \frac{\partial v}{\partial x}\right) & \frac{\partial v}{\partial y} & 0 \\ 0 & 0 & -\frac{\partial u}{\partial x} - \frac{\partial v}{\partial y} \end{bmatrix} \tag{7}$$

Nombre : 1    Auteur : ogagliardini  Sujet : Texte surligné          Date : 25/08/2018 19:57:46
should be below Eq. (4)

Nombre : 2    Auteur : ogagliardini  Sujet : Texte surligné          Date : 25/08/2018 19:58:21
and what is \epsilon_e?

where $u$ and $v$ are horizontal components of the surface velocity.

Using again the shallow ice approximation, vertical shear is neglected and the [1] effective pressure is approximated as:

$$\dot{\varepsilon}_e = \sqrt{\dot{\varepsilon}_{xx}^2 + \dot{\varepsilon}_{yy}^2 + \dot{\varepsilon}_{xy}^2 + \dot{\varepsilon}_{xx}\dot{\varepsilon}_{yy}}, \tag{8}$$

where $\dot{\varepsilon}_{ij}$ are the strain rate components (since in 2D we neglect $\dot{\varepsilon}_{xz}$ and $\dot{\varepsilon}_{yz}$ and using incompressibility $\dot{\varepsilon}_{zz} = -\dot{\varepsilon}_{yy} - \dot{\varepsilon}_{xx}$).

(iii) Horizontal strain rate gradient:

The change in strain rate sometimes is not the cause but the consequence of the presence of fractures. However, the aim of our study is to identify where fractures are present without attempting to fully describe the process by which they are formed. Thus, we use the change in strain rate as a predictor precisely because it tells us where we can expect to find fractures. This predictor allows us to discover new regions where crevasses are present even if they were not seen in the imagery. A lack of observed fractures but high strain rate means [2] that fractures may not be visible but should still be present (e.g. if fractures are covered in snow or not visible due to bad resolution of the satellite images). Furthermore, changing geometry or boundary conditions can cause changes in strain rate, and also cause fractures (e.g. a glacier flowing over a convex slope or icefall: the change in bed slope causes a change in strain rate and also causes fractures, [3] and it's not the fractures that cause the change in strain rate in this case.

(iv) Friction:

Low friction at the base of glaciers will lead to a higher sensitivity to membrane stresses, which can lead to more crevassing in tensile mode. [4] We obtain this parameter from the inversion of surface velocities in ISSM.

(v) Stiffness of ice and [5] ice [6] thickness:

In addition, we include the viscosity parameter B in Glen's flow law as well as ice thickness [7] due to their physical relation to fracture mechanics. When ice stiffness increases and ice crystals cannot creep fast enough, fracture may occur. Therefore, this parameter (obtained from the inversion of velocities implemented in ISSM) is added as a predictor. Adding temperature directly into the analysis did not improve the prediction results, which might be due to the uncertainties in the temperature estimation.

(vi) Proximity to glacier edges:

Generally the lateral friction along the glacier boundary is not considered in ice sheet models when stress is calculated. The stress field alone can predict transverse, longitudinal and radial splaying crevasses. They are all formed due to opening stress and are normally considered in existing damage modelling methods. However, the prediction of crevasses near the edges of glaciers requires a parameterisation of the lateral drag. Thus, we include the proximity to edges of glaciers and to nunataks as a predictor in our model.

**Nombre : 1**     Auteur : ogagliardini   Sujet : Texte surligné      Date : 25/08/2018 19:59:35

effective strain-rate

**Nombre : 2**     Auteur : ogagliardini   Sujet : Texte surligné      Date : 25/08/2018 20:01:53

or not. It can also be badly estimated from error in the velocity field?

**Nombre : 3**     Auteur : ogagliardini   Sujet : Texte surligné      Date : 25/08/2018 20:03:50

I don't agree with this, and it is in contradiction with what is written just before. Yes, at the end, it is the strain rate that will cause the fracture to open, because of a particular bedrock geometry

**Nombre : 4**     Auteur : ogagliardini   Sujet : Texte surligné      Date : 25/08/2018 20:05:07

the same should be tell for all parameters.

My feeling is that stress and strain rate parameters are obtained from the same observation, eg surface velocity? This should be clearly mentioned.

**Nombre : 5**     Auteur : ogagliardini   Sujet : Texte surligné      Date : 25/08/2018 20:07:42

why ice thickness here? It is not mentioned in the following paragraph

**Nombre : 6**     Auteur : ogagliardini   Sujet : Texte surligné      Date : 25/08/2018 20:06:32

**Nombre : 7**     Auteur : ogagliardini   Sujet : Texte surligné      Date : 25/08/2018 20:08:15

which I cannot see. Should be explained

(vii) Distance to the ice front and the grounding line:

[1]We can see in the satellite images that more fractures are present at a certain distance from the ice front as well as near the grounding line. We found that the relation between the presence of fractures and distance to the ice front as well as the distance to the grounding line is non-linear (Figure 3b). For most ice shelves/glaciers we can see more fractures 3-5 km as well as 10-13 km away from the front and a slightly smaller number of fractures closer than 3 km to the front or between 5 and 10 km. Therefore, instead of using $d_{IF}$ and $d_{GL}$ (distance to the ice front and the grounding line in km) as predictor variables, we construct dummy variables: $DM_{IF}$ and $DM_{GL}$, respectively, which represent two-column arrays in the following form:

$$DM_{IF} = \begin{cases} (1,1), & \text{when } 3\text{km} \leq d_{IF} < 5\text{km} \\ & \text{or} \quad 10\text{km} \leq d_{IF} < 13\text{km} \\ (1,0), & \text{when } 5\text{km} \leq d_{IF} < 10\text{km} \\ (0,1), & \text{when } d_{IF} < 3\text{km} \\ (0,0), & \text{else} \end{cases} \quad (9)$$

$$DM_{GL} = \begin{cases} (1,1), & \text{when } 5\text{km} \leq d_{GL} < 15\text{km} \\ (1,0), & \text{when } d_{GL} < 5\text{km} \\ (0,1), & \text{when } 15\text{km} \leq d_{GL} < 20\text{km} \\ (0,0), & \text{else} \end{cases} \quad (10)$$

(viii) Bed and surface slopes:

[2]There are a number of parameters such as surface velocity, surface slope and a curvature of a glacier channel that are included by other studies in the calculation of the stress field (Larour et al., 2012), but for our method we look at each component separately:

Thus, bed and surface slopes are included in the [3]model since shear stress increases on a steeper slope and can lead to fracturing (e.g. ice fall is an extreme case).

(ix) Surface gradient change:

We include this predictor in the analysis due to the fact that fracturing can be caused by [4]an increase in stress due to an abrupt change in surface elevation.

(x) Curvature:

**T** Nombre : 1          Auteur : ogagliardini  Sujet : Texte surligné          Date : 25/08/2018 20:12:07

you might capture them in the stress field with a FS model. It should be discussed that the fact that you don't capture them directly with stress is because of the SSA assumption.

**T** Nombre : 2          Auteur : ogagliardini  Sujet : Texte surligné          Date : 25/08/2018 20:12:58

I don't understand what you mean here

**T** Nombre : 3          Auteur : ogagliardini  Sujet : Texte surligné          Date : 25/08/2018 20:13:42

and why it doesn't appear in your stress field?

**T** Nombre : 4          Auteur : ogagliardini  Sujet : Texte surligné          Date : 25/08/2018 20:14:13

then is should be seen in the stress field?

[revised manuscript text omitted]

**T** Nombre : 1          Auteur : ogagliardini  Sujet : Texte surligné          Date : 25/08/2018 20:34:11

the second stage of what? Is that in link with the two obove mentioned method?

**T** Nombre : 2          Auteur : ogagliardini  Sujet : Texte surligné          Date : 25/08/2018 20:33:26

**T** Nombre : 3          Auteur : ogagliardini  Sujet : Texte surligné          Date : 25/08/2018 20:34:44

of crevasses, not damage

**T** Nombre : 4          Auteur : ogagliardini  Sujet : Texte surligné          Date : 25/08/2018 20:36:23

are you sure that these two references are not inverted?

**T** Nombre : 5          Auteur : ogagliardini  Sujet : Texte surligné          Date : 25/08/2018 20:36:05

Is that the main limitation?

Estimation of $B_T$ is the source of the main uncertainty in damage calculations due to the lack of ice temperature data, which can be crucial in affecting the accuracy of the viscosity parameter (Bassis and Ma, 2015). Thus, the errors in assumed temperature may affect the inferred value for damage.

Fractures that have been advected can be identified by damage but this is not always the case, due to the fact that the inverse method for calculating damage will only find damage where there are fractures that give rise to velocity gradients. Damage will capture some fractures that were formed upstream and advected to a region with different stress conditions only if the fracture enhances the flow and creates a local velocity gradient. Thus, we first calculate flow lines for each observed fracture[1] If upstream from the fracture the damage is larger than 50% we assume that the observed fracture was formed upstream, that the damage calculation may be correct and that the observed fracture was formed upstream. If there is no damage initiated at the point or damage upstream from the observed fracture we assign the observation point as not captured by the damage method and consider this as a failure of damage to identify the fracture (which can be due to the fact that the fracture in observation point does not cause a local gradient in strain rate) .

Physics-based methods, such as Linear Elastic Fracture Mechanics (LEFM) and Continuum Damage Mechanics (CDM), are necessary when modelling fractures in Antarctica. We do not intend to substitute these methods; rather, we seek a method that can improve on some aspects and cases when physics-based models do not predict well the formation of fractures. In particular it is possible that some fractures are initiated upstream from the grounding line rather than on floating ice. It is therefore important to be able to predict the formation of fractures in both cases. Damage is calculated only on floating ice based on model inversions using the Ice Sheet System Model (ISSM) (Larour et al., 2012) because it is not possible to distinguish between basal friction and damage on grounded ice, as they have similar effects on the ice velocity. Thus, the main motivation of this study is not to replace the damage approach, which in fact provides a strong physical background for ice sheet modelling, but to find an alternative method that can be applied to both ice shelves and grounded ice, can work for a large set of glaciers/ice shelves and does not depend on temperature observations and threshold parameters.

**5 Results**

We applied the LRA method combined with the random walk method to 45 ice shelf regions that include both ice shelves and surrounding grounded ice (the corresponding names and locations can be found in Table 6 and [2] Figure 3a, respectively) and found a best-fitting model for 44 of them. The fracturing of the remaining ice shelf cannot be described using the predictors we have, producing unacceptably large or small probabilities.

[revised manuscript text omitted]

**Nombre : 1**     Auteur : ogagliardini   Sujet : Texte surligné      Date : 25/08/2018 20:46:47

don't understand the list of glaciers here and its link to the sentence. Explain.

**Nombre : 2**     Auteur : ogagliardini   Sujet : Texte surligné      Date : 25/08/2018 20:47:50

in Tables 3, 4 and 5

**Nombre : 3**     Auteur : ogagliardini   Sujet : Texte surligné      Date : 25/08/2018 20:55:41

It was correct in the previous version, should be Fig. Refer to https://www.the-cryosphere.net/for_authors/manuscript_preparation.html

**5.1 Group 1**

This was the largest group of glaciers and the best-fit model includes as many as 10 predictors for grounded ice and seven predictors for floating ice. The analysis of the estimated coefficients in LRA showed that predictors with the highest weights in our model for this group of glaciers were: effective strain rate, proximity to glacier edges and nunataks as well as the surface elevation gradient. We present the modelled probability of fractures in Figures 5b and 4c as well as comparison with the damage-based results in Figures 7a and 10c.

The main pattern of surface fractures is well represented for this group. On grounded ice the success of identifying fractures is larger than 88% with a quarter of glaciers at almost 100%. The failure related to over-estimation of fractures is 27%. On floating ice the success amounted to 55% and the failure was equal to 15% on average. For Vanderford IS (see Figure 5b) the overall pattern is well represented, even though high resolution images were not available for this glacier. The over-estimation error is mainly related to the region that is far from the ice front and has a relatively high accumulation rate, possibly obscuring the fractures in the imagery. On floating ice the probability of fracturing is relatively smaller, mainly showing a higher chance of fracturing closer to the groundling line. Conversely, Drygalski Ice Shelf has a larger number of high resolution areas and, as a possible result, less over-estimation of fracturing (see Figure 4c). We can see that the "definitely non-fractured nodes" (selected in blue ice areas) are successfully represented in our model. For this glacier, none of the observed non-fractured nodes was assigned to have a high probability of fracturing, with the modelled probability being as low as 0.1. Moreover, in the regions with a large number of observed fractures, the probability is as high as 0.9 and it is slightly lower in the areas with a smaller number of observed fractures (between 0.6 and 0.8). Observed fractures not captured by our model were not captured by the damage-based model either.

The modelling results for the Cook ice shelf are shown in Figure 7b. There are distinct fractures visible towards the front and in the central part of the ice shelf that are not captured by either approach, which we interpret as showing that most of the fractures are formed further upstream near the groundling line. In general, the probability and the damage-based models show good agreement on the floating ice near the grounding line. However, damage does not reach 50% in the majority of the locations. Moreover, in many locations where rifts are visible it shows 0 damage.

The modelling results for Larsen B IS are illustrated on Figure 10c. It is clear that the nodes where damage is high have a high probability of fracturing due to the fact that we added damage as one of the predictor parameters to this glacier. It can be also seen that there are two lines of high probability of fracturing that coincide with the location of the large rifts that can be seen on the satellite images.

The results for Nansen IS (Figure 8c) as well as for Pine Island (Figure 6a) agree well with observations even though the data from these two glaciers were not included in the calibration data set used to construct the LRA model. For Pine Island, we observe fractures in the central part of the shelf that were not captured by the model, but our model predicted high probabilities of fracture upstream where the ice is grounded. Thus, these fractures are likely advected from the grounding line out onto the floating ice shelf.

| | Nombre : 1 | Auteur : ogagliardini | Sujet : Texte surligné | Date : 25/08/2018 20:56:22 |
|---|---|---|---|---|

Figs.

| | Nombre : 2 | Auteur : ogagliardini | Sujet : Texte surligné | Date : 25/08/2018 20:57:13 |
|---|---|---|---|---|

give its number also?

**6.5  Discussion**

We found that, in general, the most important predictor factors to model surface fractures on grounded ice for all analysed glaciers were the surface velocity and the change of the surface gradient, which is in agreement with the theory of possible mechanism of fracture formation (Colgan et al., 2016). Interestingly, the required parameters on floating ice were different from grounded ice, with effective strain rate and principal stress being the most important. Previous analysis based on damage accounts for effective stresses, thickness and viscosity, but does not include such predictors as proximity to glacier edges, nunataks and the grounding line as well as the curvature of a channel, which helped to improve the modelling of fractures on most ice shelves in our analysis. [1] Our results can be used to identify potential regions with snow covered crevasses that may pose hazards for navigation in Antarctica. Many researchers use ground penetrating radar to find hidden crevasses, but it is a real time assessment method that requires both financial and human resources and, therefore, can not cover all the areas in Antarctica. Our approach can be done remotely and at low cost, in advance of field campaigns.

We do not claim that all the predictors that were chosen in the final set for each group represent the exact fracture mechanisms for each glacier. For some ice shelf regions, sets containing different predictors can lead to results close to the best-fitting model. However, for some cases, such as Amery and Totten Ice Shelves the number of good-fitting models is very limited. For example, including the effective strain rate and proximity to the ice front in the analysis we can achieve a better fit to the observations. Therefore, we conclude that some factors have a very strong effect on fracturing, while others are only minor for some glaciers. Ultimately, we seek only to be able to develop a model that can identify correctly the geographical location of fractures, not necessarily explain why they are there.

[revised manuscript text omitted]

20    We found that the Ronne IS has the lowest elevation change as well as the principal stresses components. We do not have enough samples to cover values that are non-typical for the majority of glaciers, which may explain why we could not find a good-fit model for this ice shelf, neither with LRA nor using the Bayesian analysis. Thus, we conclude that our probabilistic model is not appropriate in this case.

**7    Conclusions**

25    Most previous large-scale modelling of surface fractures has focused on applying zero stress, Linear Elastic Fracture Mechanics, Continuum Damage Mechanics. We have shown that, using the suggested nominal parameters, damage-based approach does not fully reproduce the location of fracturing for any ice shelf region. In this study, we constructed a probability-based method to model surface fractures and generated improved predictions of fractures when physics-based models did not predict well the location of surface fractures. From this different perspective, we can construct an alternative method to predict the

30    location of fractured zones not only on floating ice but also on grounded ice.

We found that the Logistic Regression Analysis, combined with other statistical methods, can significantly improve the prediction of fractured zones for the Antarctic ice shelves/glaciers and can lead to the identification of up to 99% of observed surface crevasses for some ice shelf regions with an average of 70% for all ice shelf regions. Our approach has a number of

Nombre : 1          Auteur : ogagliardini   Sujet : Texte surligné          Date : 25/08/2018 21:12:01
this should be mentioned before

**8 Tables**

**Table 1.** Development of calving parameterisations

| Year | Reference | Method |
| --- | --- | --- |
| 1955 | Crevasse penetration depth using tensile stress and overburden pressure | Nye (1955) |
| 1973 | Crevasse penetration depth of a single crevasse | Weertman (1973) |
| 1976 | Crevasse penetration depth estimation using LEFM | Smith (1976) |
| 1993 | Strain related fracture formation | Vaughan (1993) |
| 1997 | Sea level dependent calving | Motyka (1997) |
| 1998 | Linear Elastic Fracture Mechanics | Van der Veen (1998a, b) |
| 2003 | Damage mechanics for a single crevasse | Pralong et al. (2003) |
| 2005 | Damage mechanics for a single crevasse | Pralong and Funk (2005) |
| 2007 | Crevasse depth | Benn et al. (2007a, b) |
| 2010 | Crevasse depth | Nick et al. (2010) |
| 2010 | Crevasse depth | Otero et al. (2010) |
| 2012 | Damage mechanics applied to a crevasse field | Borstad et al. (2012) |
| 2012 | Kinetic 1st order calving | Levermann et al. (2012) |
| 2012 | CDM | Duddu and Waisman (2012) |
| 2013 | CDM | Duddu and Waisman (2013) |
| 2013 | Discrete element models | Bassis and Jacobs (2013) |
| 2013 | Particle-based simulation | Astrom et al. (2013) |
| 2013 | Crevasse depth criterion | Nick et al. (2013) |
| 2014 | Crevasse depth criterion | Cook et al. (2014) |
| 2014 | CDM | Albrecht and Levermann (2014) |
| 2014 | Combining CDM and LEFM | Krug et al. (2014) |
| 2016 | [1]on Mises tensile stress | |

**Page : 27**

**Table 2.** Predictor factors (predictors)

| Type | Predictor | Description |
|---|---|---|
| **Geometry** | Ice thickness | Bedmap2 data for Antarctica at 1 km spatial resolution |
| | Maximum bed slope | Bedrock and ice surface slopes are [1] calculated using a nodal function |
| | Maximum surface slope | |
| | Proximity to the ice front | $DM_{IF}$, calculated using Eq. 9 |
| | Proximity to grounding line | $DM_{GL}$, calculated using Eq. 10 |
| | Proximity to glacier edges and nunataks | |
| | Curvature | Curvature of the glacier channel $\alpha$, calculated in each node based on the direction and rate of the flow velocities (see Eq. 11) |
| **Flow parameters** | back stress | Buttressing effect on ice streams calculated in ISSM from inversion |
| | Velocity | InSAR ice flow velocity |
| | Rheology predictor (viscosity) | $B$, Glen's flow predictor, calculated from inversion of velocities (only for floating ice) |
| | Effective Strain rate | The effective strain rate is calculated using Eq. 8 with observed velocities as an input |
| | Principal stress (1 and 2) | [2] eigenvalues $\lambda$ (normal stresses) in Eq. 4 |
| | Principal strain rate (1 and 2) | Eigenvalues $\mu$ (see Eq. 7) |
| | Strain rate gradient | Maximum strain rate change in a 400-600 metres vicinity |

**Table 3.** Formed groups of ice shelf regions

| | |
|---|---|
| **Group1** | 9, 10, 11, 12, 13, 15, 18, 20, 21, 23, 25, 26, 30, 31, 32, 34, 4, 29 |
| **Group2** | 3, 6, 7, 8, 19, 27, 28, 33, 35 |
| **Group3** | 14, 17, 22, 24 |
| **Group4** | 1, 2, 5, 16 |

Glaciers/ice shelves for which we could not find a good-fitting probability are marked with red.

Nombre : 1      Auteur : ogagliardini    Sujet : Texte surligné      Date : 25/08/2018 23:01:42

which dataset?

Nombre : 2      Auteur : ogagliardini    Sujet : Texte surligné      Date : 25/08/2018 23:02:04

from what?

**Table 4.** Predictors for grounded ice regions in each formed group

| | Effective stress | Effective strain rate | Principal 1 strain rate | Principal 2 strain rate | Principal 1 stress | Principal 2 stress | Surface slope | Bed slope | Strain change | Curvature | Rheology B | Thickness | at the ice front | at the grounding line | near edges | surface change | Velocity |
|---|---|---|---|---|---|---|---|---|---|---|---|---|---|---|---|---|---|
| Group1: | | ✓ | ✓ | | | | ✓ | | | ✓ | ✓ | | | ✓ | ✓ | ✓ | ✓ |
| Group2: | ✓ | | | | ✓ | | | | ✓ | ✓ | ✓ | | | | | ✓ | ✓ |
| Group3: | | | | ✓ | | | | | | | ✓ | | | | ✓ | | ✓ |
| Group4: | | ✓ | | | | | | | ✓ | ✓ | | | | | | ✓ | ✓ |

Tick-mark stands for an addition of a predictor to the model for grounded ice.

**Table 5.** Predictors for floating ice in each formed group

| | Effective stress | Back stress | Effective strain rate | Principal 1 strain rate | Principal 2 strain rate | Principal 1 stress | Principal 2 stress | Surface slope | Strain rate change | Curvature | Rheology B | Thickness | at the ice front | at the grounding line | surface change | Velocity |
|---|---|---|---|---|---|---|---|---|---|---|---|---|---|---|---|---|
| Group1: | ✓ | | | | ✓ | ✓ | | ✓ | ✓ | | | | | | | ✓ |
| Group2: | | | ✓ | | | ✓ | | | | | ✓ | ✓ | | ✓ | ✓ | |
| Group3: | | | ✓ | | | | ✓ | | | | ✓ | ✓ | | | ✓ | |
| Group4: | ✓ | ✓ | ✓ | | | | | ✓ | ✓ | | | | | ✓ | ✓ | ✓ |

Tick-mark stands for an addition of a predictor to the model for floating ice.

Nombre : 1        Auteur : ogagliardini   Sujet : Texte surligné        Date : 25/08/2018 22:55:41

why only putting a mark and not the derived beta value for each predictor? It would give information on which predictor are the most important in each group.

What is the beta threshold value to have a mark or not?

**Table 6.** A list of analysed ice shelf regions

| Glacier | Group | Corresponding IS name | Region |
|---------|-------|----------------------|--------|
| 1 | 4 | George IV | Palmer land, AP |
| 2 | 4 | Larsen C | Fallieres Coast, AP |
| 3 | 2 | Larsen D | Black Coast, AP |
| 1 | 1 | Orville Coast side of the Ronne IS | WA |
| 5 | 4 | Amery | EA |
| 6 | 2 | Edward VII | Mawson Coast, EA |
| 7 | 2 | Rayner Thyner | EA |
| 8 | 2 | Shirase | Prince Harald Coast, EA |
| 9 | 1 | Stancomb-Brunt | Caird Coast, EA |
| 10 | 2 | Riiser-Larsen | Princess Martha Coast, EA |
| 11 | 3 | Fimbul IS | EA |
| 12 | 1 | Abbot | Eights Coast, WA |
| 13 | 2 | Baudoin | Princess Ragnhild Coast, EA |
| 14 | 3 | Nivl | Princess Astrid Coast, EA |
| 15 | 1 | Borchgrevnik and Lazarev | Princess Astrid Coast, EA |
| 16 | 4 | Borchgrevnik | Princess Raghild Coast, EA |
| 17 | 3 | Dibble IS | Clarie Coast, EA |
| 18 | 1 | Mertz IS | EA |
| 19 | 2 | Rennik | Pennell Coast, EA |
| 20 | 1 | Cook | George V Coast, EA |
| 21 | 1 | Ninnis | George V Coast, EA |
| 22 | 3 | Holmes | Banzare Coast, EA |
| 23 | 1 | Moscow University | Sabrina Coast, EA |
| 24 | 3 | Totten IS | EA |
| 25 | 2 | Vanderford IS | EA |
| 26 | 1 | West IS | Queen Mary Coast, EA |
| 27 | 2 | Larsen C | Oscar II Coast, AP |
| 28 | 2 | Larsen B | Nordenskjold Coast, AP |
| 29 | 2 | Larsen A | Davis Coast, AP |
| 30 | 3 | Tracy-Tremenchus | Knox Coast, EA |
| 31 | 1 | Drygalski | Scott Coast, EA |
| 32 | 2 | Mariner | Borchgrevnik Coast EA |
| 33 | 3 | Rennik | Lazarev Mountains, Oates Coast, EA |

AP - Antarctic Peninsula, EA - East Antarctica, WA - West Antarctica, IS -ice shelf

**T** Nombre : 1          Auteur : ogagliardini   Sujet : Texte surligné               Date : 25/08/2018 21:16:27

Is it discussed in the text that the analysis for Ronne IS is restricted to this area?
From the text, the reader is expecting a line in this table for Ronne IS with no group assigned.

**Table 7.** A list of analysed ice shelf regions

| Glacier | Group | Corresponding IS name | Region |
|---------|-------|----------------------|--------|
| 34 | 1 | Filchner | Coast Land, WA |
| 35 | 2 | Ross East | Hut Point Peninsula, EA |
| 36 | 1 | Wilkins and George VI | Rumill Coast, AP |
| 37 | 1 | Stange and Ferringo IS | Bryan Coast, AP |
| 38 | 1 | Pine Island and Thwaites | Walgreen Coast, WA |
| 39 | 2 | Getz | Hobbs and Bakutis Coast, WA |
| 40 | 1 | Nickerson and Sulzberger | Ruppers Coast, WA |
| 41 | 1 | West | Leopold and Astrid Coast, EA |
| 42 | 1 | Jelbart and Atka | Princess Martha Coast, WA |
| 43 | 1 | Nansen | Borchgrevnik Coast, EA |
| 44 | 1 | Prince Harald | Prince Harald Coast, EA |
| 45 | 1 | Larsen B | Oscar II Coast, AP |

AP - Antarctic Peninsula, EA - East Antarctica, WA - West Antarctica, IS -ice shelf

Nombre : 1        Auteur : ogagliardini  Sujet : Texte surligné        Date : 25/08/2018 21:16:49
Should be still Table 6

(a) Grounded ice, final set vs test set 1

(b) Floating ice, final set vs test set 1

(c) Grounded ice, final set vs test set 2

(d) Floatint ice, final set vs test set 2

(e) Grounded ice, final set vs test set 3

(f) Floating ice, final set vs test set 3

**Figure 1.** Comparison between the success of identifying fractures using LRA (purple for grounded ice, green for floating ice) and using Test set 1: effective deviatoric stress, Test set 2: principal deviatoric stress 1 and 2, Test set 3: von Mises stress (blue for grounded ice, yellow for floating ice). Left column represents grounded ice (a,c,e) and right show the results for floating ice (b,d,f).

**Page : 32**

**T** Nombre : 1      Auteur : ogagliardini   Sujet : Texte surligné      Date : 25/08/2018 21:17:09
the legend are too small.

**T** Nombre : 1      Auteur : ogagliardini   Sujet : Texte surligné      Date : 25/08/2018 21:17:09
the legend are too small.

[Figure]

(a) Location

(b) Distance to the ice front

**Figure 3.** The location of each of the 45 ice shelf regions is shown in panel a. The number of observed fractures versus distance from the ice front (b).

Nombre : 1      Auteur : ogagliardini Sujet : Texte surligné    Date : 25/08/2018 21:19:02
What is the reason to have these two panels in the same figure?

[Figure]

(a) LRA, Shirase IS        (b) Damage, Shirase IS

(c) LRA, Totten IS        (d) Damage, Totten IS

0    0.2    0.4    0.6    0.8    1

<-- Less likely to fracture   /   More likely to fracture -->

**Figure 11.** Modelled probability of a fracture vs. modelled damage for Shirase IS (Group 2) (a, b) and Totten IS (Group 3) (c, d). Labels are the same as in figure 4 and 7 .

as in Figs. 4 and 7.

[Figure]

(a) Baudoin IS, Group 1

(b) Rennik IS, Group 1

0    0.2    0.4    0.6    0.8    1
<-- Less likely to fracture   /   More likely to fracture -->

**Figure S1.** Group 1: Modelled probability of a fracture for Baudoin IS (a) and Rennik IS (b). Observed surface fractures are shown in black and observed non-fractured ice is marked with orange circles. Red polygons represent regions where high resolution images were available.

**Page : 44**

[Figure]

I was not able to find a reference to these S1 to S8 figures in the text. Check that all figures are cited in the text.

(a) Stange and Ferringo IS, Group 1

(b) Nickerson and Sulzberger IS, Group 1

(c) West IS, Group 1

0    0.2    0.4    0.6    0.8    1
<-- Less likely to fracture   /   More likely to fracture -->

**Figure S8.** Group 1: Modelled probability of a fracture for Stange and Ferringo IS (a), Nickerson and Sulzberger IS (b) and West IS (c). Labels the same as Figure S1.

Fig. S1.

Fig. S1.

---

## Author Response (AR2)

**Response to the suggestions for revision:**

RC: This manuscript is a revised version of a statistical method to predict the location of fractures in Antarctic ice shelves and glaciers. My initial review had quite a few comments and I think the authors have largely addressed these comments. I still found quite a few typos, although I found fewer in the later parts the manuscript. I don't know if that is because I became less diligent in checking or there were just a few spots early on that were problematic. The authors should thoroughly go through and attempt to minimize ferret out any additional mistakes.

RC: I still have several comments, although most of these are technical and get into the minutia of the text.

Overarching comments:

Results: I would like a bit more comment on the differences between the damage approach and the probability of fracture. If I were to fully trust the inferred probability of fracture, then I would be forced to conclude that the damage inversion is rather unreliable. But the damage method is not only picking up on surface crevasses and might be sensitive to depth of crevasses, amongst other things. (It is very disturbing that the damage method is not picking up on known locations of rifts in ice shelves.) Moreover, the inference might not be as reliable in all regions. This isn't something that needs to be resolved, but could be addressed in more detail.

AC: It is correct that damage is not capable of identifying every rift and fracture, however, it does not represent a failure in damage method, rather the fact that it is not designed to do so.
 The inversion only infers damage in areas where fractures (crevasses or rifts) are being ACTIVELY formed and thus creating a jump in strain rate/velocity. Many rifts are formed at one point in time and then only intermittently propagate. If the velocity observations don't show a discrete jump across a fracture, then there is nothing for the inversion to pick up in terms of damage. This is the definition of the inferred damage, and it is not meant to locate every fracture. It only finds fractures that are actively enhancing the flow.

We have tried in this paper not to be overly critical of the damage method, but the reviewer has picked up on the fact that the damage method fails to predict crevasses in known locations of fractures. We would prefer not to make stronger statements on this issue. In fact, since it was understood by the reviewer, it is probably not necessary to do so. However, we added in the Conclusions the following statements:
 "The probabilistic results suggest that using our method we can identify fractures and rifts that the damage method is not designed to identify (especially in predicting locations of old rifts in ice shelves).
To "Calculation of Damage" section we added:

" It is important to keep in mind that the inversion only infers damage in areas where fractures (crevasses or rifts) are being actively formed and, thus, creating a jump in

strain rate/velocity. Many rifts are formed at one point in time and then only intermittently propagate. If the velocity observations do not show a discrete jump across a fracture, then there is nothing for the inversion to pick up in terms of damage. It only finds fractures that are actively enhancing the flow and it is not meant to locate every fracture."

AC: The uncertainty related to temperature we had already (section "Calculation of Damage)

AC: We added to the "Conclusion":

"The probabilistic results suggest that our statistics-based methods are more reliably in identifying fractures and rifts in the locations were the damage method does not predict them (which is not related to a failure in the damage method, but the fact that it is not constructed to do so). There are also uncertainties in the damage-based method related to the surface temperatures in Antarctica, which may be poorly represented with available observations."

RC: Figures: I would have liked to see the same colour scale used for damage and probability as both of these range from 0 to unity to make it easier to compare. I can understand if the authors don't want to do this because damage and fracture probability aren't the same, but I think it would be easier to see how well the model is reproducing the fracture probability.

AC: We have changed all the figures describing damage results in the manuscript.

Technical comments:

RC: Page 2, line 20: fractures can be more frequent than every 50 m

AC: Changed to: " can occur more often than every 50 m".

RC: Page line 23, built-->build

AC: Corrected

Page 2 line 26: prediction-->predictive

AC: Corrected

RC: Page 2 line 29: missing "the" before damage? Something is grammatically off. Also, you have not introduced the damage method yet so this doesn't make a lot of sense.

AC: Changed to: "whereas in a damage-based method that we used for a qualitative comparison the average success rate for floating ice was equal to 34\%"

RC: Page 2 line 16 extra parenthesis in references

AC: Corrected

RC: Page 2, line 25 extra period.

AC: Corrected

RC: Page 2, line 26-30: this sentence appears to be exactly the same as in the abstract.

AC: The sentence in the text says: "We compare the ability of our model to match observations of fractures from satellite imagery versus the prediction ability of the damage-based method of Borstad et al. (2013). From the modelling of 45 ice shelves/glaciers, we found that we can predict the location of fractures that match the observations with a success rate from ~45% to 99% with an average success rate of 84% (Figure 2a) for grounded ice and 61% for floating ice (Figure 2b) (we found that the average success rate when applying damage method to floating ice 30 to be equal to 34%). "

AC: In the abstract it is: We can predict the location of observed fractures with an average success rate of 84% for grounded ice and 61% for floating ice and mean over-estimation error of 26% and 20%, respectively.
The sentences are not the same. The abstract repeats only the information about the average success rate. We have modified the sentence in the manuscript to:

"From the modelling of 45 ice shelves/glaciers, we found that we can predict the location of fractures that match the observations with a success rate ranging from ~45% to 99% (Figure 2a) for grounded ice and ~30% to 90% for floating ice (Figure 2b). The average success rate when applying damage method to floating ice was equal to 34% in contrast to 61% achieved when applying our LRA method."

RC: Page 4, line 20: What do you do to infer ice temperature

AC: " The information about the ice temperature for grounded ice is calculate as the depth-average temperature from Liefferinge and Pattyn (2013); Pattyn (2010) (21 vertical levels). The steady-state depth averaged ice temperature on floating ice shelves (mainly used for the calculation of damage) was calculated using surface, basal temperatures and basal melting rate according to Holland and Jenkins (1999). "

25 To calculated ice temperature we corrected the surface temperature with a lapse rate and imposed it on the ice surface. Basal melting rates on ice shelves were taken from (Depoorter et al., 2013).

RC: Page 5, missing space between swell and open parentheses.

AC: Corrected.

RC: Page 5: I don't know that there is any evidence to support the hypotheses that tidal deformation is a strong driver of basal fractures or rifts. It might, but the strength of this statement is a bit excessive given the fact that no references are provided to support it.

AC: Removed.

RC: Page 6 line 15: How are the discrete fracture locations observed turned into a probability distribution? This seems to be described later. Is this related to the area that they occupy? Also, note that you can have deep or shallow surface crevasses. The spacing of crevasses is often related to their depth and hence a field of widely spaced crevasses might consist of a smaller number of crevasses that penetrate much deeper into the ice than an equivalent set of closely spaced shallow crevasses. This may factor into the comparison with damage.

AC: Good points. We added a comment to this effect to the line 15: "It is important to mention that the spacing of crevasses is often linked to their depth. A single crevasse can penetrate much deeper than a crevasse in a set of closely spaced crevasses. However, in this study we do not focus on estimating either depth or spacing of crevasses."

RC: Page 7, line 25: I see that here you recognize the problem of crevasse depths. You might think about moving this statement up to respond to my previous comment.

AC: We have added a sentence describing it to Page 6.

RC: Equation 5 is a bit confusing. In equation 4, you define the deviatoric stress using the symbol $\sigma'$ and components with the notation $tau_{xx}$, $tau_{xy}$, $tau_{yy}$. In equation 5, components of the deviatoric stress are instead denotes using $\sigma_{ij}'$. Be consistent with your notation and usage.

AC: Corrected.

RC: Equation 6: I'm really confused by your notation here. The Von Miss stress is proportional to the second deviatoric stress invariant, but this isn't apparent from the set of equations.

AC: Corrected

RC: Equations 8: I would have expected the effective strain rate to look similar to the effective stress. You can write both of these in different forms, but it would be helpful for readers to see the symmetry in the equations.

AC: The equation is correct when we use the shallow ice approximation. We added some clarification in the text.

RC: Equation 7-8: Now, because ice is incompressible there is also a vertical strain rate ($\partial w/\partial z$, say) and this term is non-zero. It is unclear why this has been omitted from Equation 7 and Equation 8. In fact, my guess is that this term is included when ISSM calculates the effective strain rate . . .

AC: Corrected.

RC: Page 10, line: I don't really understand why back stress is included. Aside from the fact that it is ambiguous in a two-dimensional model, the effect of back stress should be entirely captured by the stress and strain metrics already used.

AC: Removed.

[revised manuscript text omitted]

---

## Author Response (AR3)

**Page : 2**

Nombre :  should read: (e.g. Nick et al., 2010; Cook et al., 2014; Krug…),

AC: Corrected everywhere.

Nombre : 2 Auteur

AC: Corrected.

Nombre : 3

AC: Corrected.

**Page : 3**

 Nombre : 1

AC: Corrected.

Nombre : 2  This paragraph could be improved and reduced by avoinding first listing the  model and then going into the description of each one.

AC: Modified.

Nombre : 3 give a paper reference not the name of a specific model

AC: Added.

Nombre : 4 Elmer/Ice

AC: Corrected.

Nombre : 5 .

AC: Corrected.

Nombre : 6 already defined above

AC: Corrected.

Nombre : 7 Which method are you referencing to? Not clear from the previous sentence.

AC: Modified the sentence to make this clear.

Nombre : 8

AC: Corrected.

**Page : 4**

Nombre : 1 should be above?

AC: Corrected.

Nombre : 2 might not be applicable

AC: Corrected.

Nombre : 3 is that the correct name for this section?

AC: Changed to: " Construction of the input data to the statistical model"

Nombre : 4 what is the BC on this fictitious boundary

AC: Added: "Neuman boundary conditions at the ice front and Dirichlet boundary conditions at the boundary of the ice shelf region".

Nombre : 5 Auteur :

AC: Corrected.

Nombre : 6 applying which velociy ? Observation?

AC: Corrected.

**Page : 5**

Nombre : 1 output of which simulations. I only see ISSM simulations in what is above?

AC: The simulations for the 45 ice shelf regions. Added this to the paragraph.

**Page : 6**

Nombre : 1 why a new definition of LRA here?

AC: Corrected the definition.

**Page : 7**

Nombre : 1 again

AC: Corrected.

Nombre : 2 before the ref

AC: Corrected.

**Page : 8**

Nombre : 1 where j is ?

AC: Removed the word 'for'.

Nombre : 2 :

AC: Corrected.

Nombre : 3

AC: Corrected.

**Page : 9**

Nombre : 1 should be below Eq. (4)

AC: Corrected.

Nombre : 2 and what is \epsilon_e?

AC: Added.

**Page : 10**

Nombre : 1 effective strain-rate

AC: Corrected

Nombre : 2 : or not. It can also be badly estimated from error in the velocity field?

AC: Yes, added this sentence to page 1o.

Nombre : 3 : I don't agree with this, and it is in contradiction with what is written just before. Yes, at the end, it is the strain rate that will cause the fracture to open, because of a particular bedrock geometry

AC: Both processes can take place: fractures causing a change in strain rate or strain rate causing a fracture to open because of particular geometry. The first was added in the manuscript because the previous reviewer pointed it out. We have modified

the paragraph to make it more clear.

Nombre : 4 the same should be tell for all parameters.

AC: All the descriptions of the parameters were modified accordingly.

My feeling is that stress and strain rate parameters are obtained from the same observation, eg surface velocity? This should be clearly mentioned.

AC: It is correct, added to the beginning of the section.

Nombre : 5 : why ice thickness here? It is not mentioned in the following paragraph

AC: Added as another separate item

Nombre : 6

AC: Corrected

Nombre : 7 which I cannot see. Should be explained

We split the paragraph and added the description for ice thickness:

"Ice thickness was included due to the fact that fracture formation in thicker glaciers/ice shelves might differ from fracturing in thin glaciers/ice shelves.

This parameter was taken directly from Bedmap2 observations."

**Page : 11**

Nombre : 1 you might capture them in the stress field with a FS model. It should be discussed that the fact that you don't capture them directly with stress is because of the SSA assumption.

AC: The statement related to Nombre 1 has nothing to do with SSA or any modelling at all. It is simply a statement of fact to point out to the reviewers where the number of visible fractures are highest.

Nombre : 2 I don't understand what you mean here

AC: Changed to: " Calculation of the stress field already considers surface velocity, surface slope and a curvature of a glacier channel \citep{larour2012continental}, but for our method we estimate separately the effect of each parameter on fracturing:"

Nombre : 3 and why it doesn't appear in your stress field?

AC: It is included. It is mentioned above.

Nombre : 4 then is should be seen in the stress field?

AC: It is in the stress field, as mentioned above.

Added to make it more clear: "because including only stress was not sufficient to explain fracture formation (described in Section \ref{sec:stress}):"

**Page : 12**

Nombre : 1 is it x_i^\star or x_i which is used in Eq. (1)? This should be mentioned.

AC: Added.

Nombre : 2 more than the ice sheet model, it is the simplification of the Stokes equation that might explain this.

AC: Added.

Nombre : 3 20%

AC: Corrected

Nombre : 4 50%

AC: Corrected

Nombre : 5 this is not a stress measure, but a velocity measurement converted in stress through a model relying on SSA.

AC: Changed to " This shows that stress derived from velocity observations through a model relying on SSA".

Nombre : 6

AC: Corrected

**Page : 15**

Nombre : 1 Auteur : there should be a i in the riht member?

AC: No it should not be i. It was written already that i is the number of the predictor, j is the summation index for kj.

**Page : 16**

Nombre : 1 the second part of the sentence is not restricticted to inversion?

AC: Changed to: "methods for calculation of damage"

**Page : 17**

Nombre : 1 the second stage of what? Is that in link with the two obove mentioned method?

AC: Modified to:
"A method for modelling of propagation of damage was suggested by \citet{krug2014combining, albrecht2014fracture} using an advection scheme and a source function." It is related to previously mentioned: " There are two different methods for calculation of damage: methods applied to invert for damage and methods used to model damage propagation in ice sheet models."

Nombre : 2

AC: Corrected.

Nombre : 3 of crevasses, not damage

AC: It is damage that we describe in this section, not crevasses. However, we meant advection and not propagation. Changed to: "advection of damage".

Nombre : 4 are you sure that these two references are not inverted?

AC: Corrected.

Nombre : 5 Texte surligné Is that the main limitation?

AC: No, it is not the main limitation, just one of the limitations. Changed to: " Another limitation of these models is that they do not yet account for factors such as ice fabric and impurities"

**Page : 18**

Nombre : 1 is the repetition in this sentence necessary?

AC: Corrected.

Nombre : 2 Auteur  Fig. check this everywhere in the manuscript and refer to the TC instructions.

AC: Corrected everywhere.

**Page : 19**

Nombre : 1 don't understand the list of glaciers here and its link to the sentence. Explain.

AC: Changed to:
This took place for six glaciers/ice shelves with the numbers 10, 13, 15, 11, 30, 32 (see Table \ref{table: groups}).

Nombre : 2 in Tables 3, 4 and 5

AC: Corrected

Nombre : 3  It was correct in the previous version, should be Fig. Refer to https://www.the-cryosphere.net/for_authors/manuscript_preparation.html

AC: Corrected

**Page : 20**

Nombre : 1 Figs.

AC: Corrected.

Nombre : 2 give its number also?

AC: Added.

**Page : 24**

Nombre : 1  is this paragraph at the right place. At most place, radar is used for security, as crevasses evolve from on year to an other, so that I am not convince your method could completely replace the instantaneous filed measurement.

AC:  Moved to conclusion section. Modified: " This statistics-based method can help to expand our current knowledge of the crevasses as well as improve mapping of potential hazards. Our results can be used  to identify potential regions with snow covered crevasses that may pose hazards for navigation in Antarctica and thus, complement field campaigns and in-situ observations."

**Page : 25**

Nombre : 1 this should be mentioned before

AC: Moved it up one paragraph

**Page : 27**

Nombre : 1 missing reference?

AC: Corrected.

**Page : 28**

Nombre : 1 which dataset?

AC: Added: (Using Bedmap2 dataset)

Nombre : 2 from what?

AC: Added: (Using InSAR velocity observations)

**Page : 29**

Nombre : 1  why only putting a mark and not the derived beta value for each predictor? It would give information on which predictor are the most important in each group.

What is the beta threshold value to have a mark or not?

AC: In our method beta is a coefficient in the probability function. The table is not related the beta value, but it simply shows whether a parameter was or was not included in the analysis. There are no exact value that determined whether a parameter is included or not because the selection was based on a number of algorithms  that are explicitly described in Section 4.3 and trial tests. Beta that the editor refers to is a weight coefficient that is calculated within LRA model only for the parameters that are included in the model.
However, we think that the of the editor suggestion to provide beta values can add more detail to the manuscript, therefore we added values of beta to the table instead of the ticks to provide an additional information on the weight of each of the included parameters in calculation of probabilities.

We modified the tables and added to the description: " It is important to mention that $\beta$ does not define wether a parameter is included in the model or not. The $\beta$ coefficients are calculated only for the parameters that were already

included in the model as they are a part of the model itself, it's the weight defining the calculation of the probabilities of fracturing (see Section \ref{sec:LRA}).

**Page : 30**

Nombre : 1 Is it discussed in the text that the analysis for Ronne IS is restricted to this area? From the text, the reader is expecting a line in this table for Ronne IS with no group assigned.

AC: Corrected.

**Page : 31**

Nombre : 1 Should be still Table 6

AC: Corrected.

**Page : 32**

Nombre : 1 the legend are too small.

AC: Corrected.

**Page : 34**

Nombre : 1 What is the reason to have these two panels in the same figure?

AC: Broke it into two figures.

**Page : 42**

Nombre : 1 as in Figs. 4 and 7.

AC: Corrected.

**Page : 44**

Nombre : 1  I was not able to find a reference to these S1 to S8 figures in the text. Check that all figures are cited in the text.

AC: We added to the results section: Plots for other glaciers/ice shelves are shown in Figs. \ref{fig:SupG1}-\ref{fig:SupG4} in the Supplementary Material.

**Page : 51**

Nombre : 1 Fig. S1.

AC: Corrected.

[revised manuscript text omitted]

---

## Author Response (AR4)

AC: We have added the data to http://www.usap-dc.org/view/dataset/601117
(https://doi.org/10.15784/601117).
The section data availability is added to the manuscript accordingly.

- page 1, line 25: of Larsen C ice shelf](Mercer, 1978 -> of Larsen C ice shelf)(Mercer, 1978

AC: Changed bracket.

- page 2, lines 25-34: this part is more already a conclusion presenting the main results. The introduction section should end with a presentation of the different sections that will follow. Also, regarding the comparison of the LRA results and the damage results, there is a bias that should be discussed. Your method is build on the observations you are using to evaluate the method whereas the damage method is not using these observations, and is doing a much more complicated job?

AC:Changed to:

" In Section  \ref{sec:1} we describe previous studies focused on fracture and calving modelling. In Section \ref{sec:2} we provide information about the datasets used to construct our model. In Section \label{sec:Methods} we construct our probability model. In Section \ref{sec:3} we group glaciers/ice shelves based on common characteristics, then in Section \ref{disc} we describe each group. Our best results were achieved combining LRA with Bayesian as well as Jensen-Shannon Divergence theory described in Section \ref{sec:Methods}. "

- page 4, line 17: this is not completely clear if this first dataset is included in the others? And also, my understanding was that the dataset only contains the crevasses location, but seems not the case from this sentence?

AC: Here we are talking not about the crevasses data set, but the information about the flow parameters in the locations of observations. They are not the same data sets. We added the following to make it clear:
" To construct the model we needed two types of data sets: a data set with combined observations of fractures and a data set with the flow/geometry information in each fracture. To run the model for a particular ice shelf region only information about the flow/geometry is required.
We constructed 46 data sets containing information about flow regime/geometry (different from the observational data set containing information about the location of fractures)"

- page 4, line 21: (described in Section 4.2. The 450-metre resolution horizontal ice velocities were taken from InSAR (Rignot et al., 2011b, a)). -> (described in Section 4.2). The 450-metre resolution horizontal ice velocities were taken from InSAR (Rignot et al., 2011b, a).

AC: Removed extra bracket.

- page 4, line 27: two-dimensional (2-D) and three-dimensional (3-D) ...

AC: Corrected.

- page 5, line 23: mélange is more a mixture of icebergs and sea ice than snow and sea ice

AC: Corrected

- page 6, line 34: to the fact if there -> to the fact that if there

AC:  Corrected.

- Eq. (1): should be x^*_{ij} in this equation?

AC: Yes, corrected.

- after Eq. (5), is it a new sentence (then a dot after (5)), or not (then a "and" before Von Mises).

AC: Corrected to a new sentence.

- page 12, line 4: fracture formation (described in Section 4.2.1): -> fracture formation (described in Section 4.2.1).

AC: Corrected.

- page 12, line 8: the definition of the nodal function should arrive before the previous sentence. Also, this sentence is a bit technical and could be simplified? e.g. The calculation of these (not this) parameters is performed by computing the derivative of the bed and surface using the finite element nodal function in ISSM (I don't think you need to define them).

AC: Corrected to the suggested sentence. and added: "(a linear function calculated for each node using the information about the horizontal coordinates)"

- page 13, line 7: only effective deviatoric stress -> only the effective deviatoric stress

AC: Corrected.

- page 13, line 10: deviatoric or Cauchy principal stresses?

AC: Corrected to deviatoric.

- page 17, line 23: a bit vague and it is really needed? This section about damage is mixing bibliography on previous works (that might have been presented in the introduction) and what is really done here, and all is a bit mixed so that it is not easy to follow. I suggest to move part of this material in the introduction to only keep what you have really used to infer the damage here.

AC: We have moved the introduction to damage methods to the Section Background.

- page 24, line 19: what do you mean by surface gradient? surface slope or strain-rate? (surface slope doesn't seem to be so important from Tables 4 and 5).

AC: It is not the surface slope we refer to. We have clarified this by adding the following to the manuscript:
" the surface change (maximum difference between the surface elevation within 500 metres radius)"

- page 25, line 9: fractures in or evaluation -> fractures in our evaluation

AC: Corrected.

- page 26, line 4: I don't understand what you mean by "zero stress". Also: Mechanics, Continuum Damage Mechanics -> Mechanics and Continuum Damage Mechanics.

AC: To make it clearer we added:

"Zero Stress Models (propagation of crevasses to the depth where the overburden pressure and the tensile stress are equal \citep{nye1955comments})"

- Figure 7: Is there red dots only in panel c? This is difficult to see red dots over the damage or LRA coulour scale.

AC: Yes, the red dots are in panel c) only. They are the removed crevasses in damage method only, they are not applied to LRA and are not shown.

- caption of Fig. 8 should be: Labels are the same as for Fig. 7. and the last sentence removed.

AC: Corrected.

[revised manuscript text omitted]

---

## Author Response (AR5)

**Editor Decision: Publish subject to minor revisions (review by editor)** (07 Sep 2018) by Olivier Gagliardini
Comments to the Author:
Dear Veronika,

Thanks for this new version and your reply to my comments.

I nevertheless think there is still missing information about the way the 46 datasets are constructed. Page 5, it is mentioned that the first dataset includes combined information from 35 regions, and the 45 others information for each individual region. Is the first dataset including all the information of the 35 regional datasets (i.e. is it just the sum of the 35 individual dataset?)? Some words should be added about the choice of these 35 regions? How they were chosen? Randomly?

AC: There is no duplication of use of observations in the calibration and validation processes. We use a subset of the observations on 35 glaciers to calibrate the model (i.e to estimate the beta coefficients in the LRA), which is roughly 5% of all observations on each glacier. We deliberately left 10 glaciers out of the calibration process so that we had independent observations against which to compare our model results. We don't use all the observations of the 35 glaciers when we construct the model so that we can then compare our model results against the remaining observations on the 35 glaciers plus the full set of observations on the other 10 glaciers not used in the calibration process. The 35 glaciers were chosen at random from the 45 glaciers, which were themselves chosen because high resolution imagery was available for them.

We have modified Section 3.2 to make it clearer:

"We form the evaluation data set to test how well our new approach predicts fractures for each ice shelf region individually. The evaluation set for each glacier/ice shelf is much larger than the number of fractures selected from each of the regions for calibration as we did not need every observed fracture to construct the model (as previously mentioned it was the variety and not the number of data points that was required for a successful construction of our model)."

We also added to Section 3:

"To construct the model we needed two types of data: observations of fractures and the flow/geometry information in each fracture. To run the model for a particular ice shelf region only information about the flow/geometry is required. We chose 45 ice shelves/glaciers where high resolution imagery was available.

For calibration of the model we constructed a data set containing information about flow regime/geometry and the locations of some observed fracture/non-fracture (described in Section \ref{seq:fractures}) collected from 35 ice shelf regions. We use these observations to derive the \beta values in the LRA. The 35 ice shelves/glaciers were chosen at random out of the 45 analyzed regions for which we had observed fractures.

To validate the model results we use an independent set of observed fractures

on these 35 ice shelves/glaciers. Thus, there is no duplication of use of observations in the calibration and validation processes. Overall, there are twenty times more observations of fractures used for validation of the model than in calibrating the model. We include in the validation process fracture observations on an additional 10 ice shelves/glaciers, which provide completely independent observations against which to compare our model results. "

How the results are influenced by the choice of the 35 regions included in the first dataset?

AC: The selection of the 35 glaciers has no effect on the calibration of the model. We tested using all the observations on each glacier and just using a subset and this did not have any significant effect. We have added this to the manuscript.

The 35 regions chosen to calibrate the model should be tagged in Table 6?

AC: It was already stated in the text that the "first" 35 glaciers were the ones used in the calibration process. We have now added a "*" superscript to each of the 35 glaciers in the Table to make this clearer.

This bring a second question, that was already in my last comments but that you didn't answered : "Also, regarding the comparison of the LRA results and the damage results, there is a bias that should be discussed. Your method is build on the observations you are using to evaluate the method whereas the damage method is not using these observations, and is doing a much more complicated job?". This is even more true if the method is evaluated against data 35/45=78% of which have been used to calibrate the model?

AC: There is no duplication of use of observations in the calibration and validation processes and, therefore, there is no bias. We have now made this clearer in the manuscript.

Regarding my remarks on the wording "surface gradient", that you have changed in "surface change": you should check in all the manuscript that the same wording is used for all the predictors all along the manuscript. For example, the name (and number! 14 in Table 2 and 16 in Tables 4 and 5) of predictors in the first column of Table 2 should be the same as those in the first line of Tables 4 and 5 (even is some of them are only relevant for grounded part or floating part, they should be tagged as irrelevant). Typically, "surface change" is not listed in Table 2 as a predictor and this name appear for the first time in the discussion page 24, where it is given as one of the most important (and its definition is only given in the discussion, why?).

AC: Corrected the tables and the word gradient.

All the predictors (16 as in Tables 4 and 5 or 19, as stated at the beginning of 4.2) should be defined in Table 2 and in the text in Sect. 4.2. Also, the order of the predictors in Tables 2, 4 and 5 should be the same for an easiest comparison.

AC: Corrected the tables accordingly. There were 18 predictors used, because as discussed in the manuscript, we removed friction from the analysis. To clarify we add:

" the set of factors  for each group are presented in Tables \ref{table: groups}, \ref{table: predictors} and \ref{table: predictors_shelf}, respectively (18 predictors, as friction was discarded)."

Names should be exactly the same: are "strain-rate gradient" in Table 2, "strain change" in Table 4 and "strain-rate change" in Table 5 the same predictor?

AC: Yes, corrected.

[revised manuscript text omitted]